# A Critical Analysis of Document Out-of-Distribution Detection

**Jiuxiang Gu**[1*]    **Yifei Ming**[2*†]    **Yi Zhou**[3]    **Jason Kuen**[1]
**Vlad I. Morariu**[1]    **Handong Zhao**[1]    **Ruiyi Zhang**[1]    **Nikolaos Barmpalios**[1]
**Anqi Liu**[3]    **Yixuan Li**[2]    **Tong Sun**[1]    **Ani Nenkova**[1]
[1]Adobe Research    [2]University of Wisconsin-Madison    [3]Johns Hopkins University
[1]{jigu,kuen,morariu,hazhao,barmpali,ruizhang,tsun,nenkova}@adobe.com
[2]{alvinming,sharonli}@cs.wisc.edu    [3]yzhou188@jhu.edu    [3]aliu@cs.jhu.edu

## Abstract

Large-scale pre-training is widely used in recent document understanding tasks. During deployment, one may expect that models should trigger a conservative fallback policy when encountering out-of-distribution (OOD) samples, which highlights the importance of OOD detection. However, most existing OOD detection methods focus on single-modal inputs such as images or texts. While documents are multi-modal in nature, it is underexplored if and how multi-modal information in documents can be exploited for OOD detection. In this work, we first provide a systematic and in-depth analysis on OOD detection for document understanding models. We study the effects of model modality, pre-training, and fine-tuning across various types of OOD inputs. In particular, we find that spatial information is critical for document OOD detection. To better exploit spatial information, we propose a spatial-aware adapter, which serves as a parameter-efficient add-on module to adapt transformer-based language models to the document domain. Extensive experiments show that adding the spatial-aware adapter significantly improves the OOD detection performance compared to directly using the language model and achieves superior performance compared to competitive baselines.

## 1 Introduction

The recent success of large-scale pre-training has propelled the widespread deployment of deep learning models in the document domain, where model predictions are used to help humans make decisions in various applications such as tax form processing and medical reports analysis. However, models are typically pre-trained on data collected from the web but deployed in an environment with distributional shifts (Cui et al., 2021). For instance, the outbreak of COVID-19 has led to continually

---

* Equal contribution
† Work done during the internship at Adobe Research

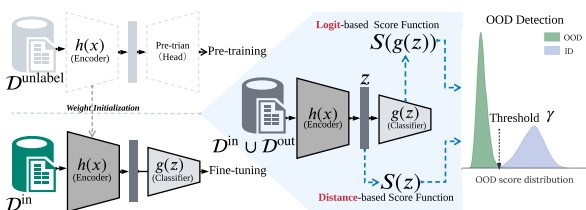

Figure 1: Illustration of OOD detection for document classification. The pre-training and fine-tuning pipelines are shown on the top left and bottom left, respectively. Right: During inference time, an OOD score can be derived based on logits $g(x)$ or feature embeddings $z := h(x)$. A document input $x$ is identified as OOD if its OOD score is below some threshold $\gamma$.

changing data distributions in machine-assisted medical document analysis systems (Velavan and Meyer, 2020). This motivates the need for reliable document understanding models against out-of-distribution (OOD) inputs.

The goal of OOD detection is to categorize in-distribution (ID) samples into one of the known categories and detect inputs that do not belong to any known classes at test time (Bendale and Boult, 2016). A plethora of OOD detection methods has been proposed for single-modal (image or text) inputs (Ge et al., 2017; Nalisnick et al., 2019; Oza and Patel, 2019; Tack et al., 2020; Hsu et al., 2020; Arora et al., 2021; Zhou et al., 2021; Xiao et al., 2020; Xu et al., 2021a; Li et al., 2021b; Shen et al., 2021; Jin et al., 2022; Zhou et al., 2022; Ming et al., 2022b,c; Podolskiy et al., 2021; Ren et al., 2023). Recent works (Fort et al., 2021; Esmaeilpour et al., 2022; Ming et al., 2022a; Ming and Li, 2023; Bitterwolf et al., 2023) also demonstrate promising OOD detection performance based on large-scale models pre-trained on text-image pairs, as pre-training enables models to learn powerful and transferable feature representations (Radford et al., 2021). However, it remains largely unexplored if existing findings in the OOD detection literature for images or texts can be naturally extended to the document

domain.

Multiple unique challenges exist for document OOD detection. Unlike natural images, texts, or image-text pairs, no captions can describe a document and images in documents rarely contain natural objects. Moreover, the spatial relationship of text blocks further differentiates multimodal learning in documents from multimodal learning in the vision-language domain (Lu et al., 2019; Li et al., 2020). In addition, while recent pre-training methods have demonstrated remarkable performance in downstream document understanding tasks (Xu et al., 2020, 2021b; Li et al., 2021a; Gu et al., 2022; Hong et al., 2022; Huang et al., 2022; Li et al., 2022; Wang et al., 2022a), existing pre-training datasets for documents are limited and lack diversity. This is in sharp contrast to common pre-training datasets for natural images. It remains underexplored whether existing OOD detection methods are reliable in the document domain and how pre-training impacts OOD reliability.

In this work, we first present a comprehensive study to better understand OOD detection in the document domain through the following questions: *(1)* What is the role of document pre-training? How do pre-training datasets and tasks affect OOD detection performance? *(2)* Are existing OOD detection methods developed for natural images and texts transferrable to documents? *(3)* How does modality (textual, visual, and especially *spatial* information) affect OOD performance? In particular, we find that spatial information is critical for improving OOD reliability. Moreover, we propose a new spatial-aware adapter, a small learned module that can be inserted within a pre-trained language model such as RoBERTa (Liu et al., 2019). Our module is computationally efficient and significantly improves both ID classification and OOD detection performance (Sec. 5.2). Our contributions are summarized as follows:

- We provide an extensive and in-depth study to investigate the impacts of pre-training, fine-tuning, model-modality, and OOD scoring functions on a broad spectrum of document OOD detection tasks. Our codebase will be open-sourced to facilitate future research.

- We present unique insights on document OOD detection. For example, we observe that distance-based OOD scores are consistently advantageous over logit-based scores, which is underexplored

in the recent OOD detection literature on vision-language pre-trained models.

- We further propose a spatial-aware adapter module for transformer-based language models, facilitating easy adaptation of pre-trained language models to the document domain. Extensive experiments confirm the effectiveness of our module across diverse types of OOD data.

## 2 Preliminaries and Related Works

### 2.1 Document Models and Pre-Training

Large-scale pre-trained models gradually gain popularity in the document domain due to their success in producing generic representations from large-scale unlabeled corpora in vision and natural language processing (NLP) tasks (Devlin et al., 2018; Lu et al., 2019; Su et al., 2019; Schiappa et al., 2022). As documents contain both visual and textual information distributed spatially in semantic regions, document-specific models and pre-training objectives are often necessary, which are distinct from vision or language domains.

We summarize common model structures for document pre-training in Fig. 2a. Specifically, LayoutLM (Xu et al., 2020) takes a sequence of Optical Character Recognition (OCR) (Smith, 2007) words and word bounding boxes as inputs. It extends BERT to learn contextualized word representations for document images through multitask learning. LayoutLMv2 (Xu et al., 2021b) improves on the prior work with new pre-training tasks to model the interaction among texts, layouts, and images. DocFormer (Appalaraju et al., 2021) adopts a CNN model to extract image grid features, fusing the spatial information as an inductive bias for the self-attention module. LayoutLMv3 (Huang et al., 2022) further enhances visual and spatial characteristics with masked image modeling and word-patch alignment tasks. Another line of work focuses on various granularities of documents, such as region-level text/image blocks. Examples of such models include SelfDoc (Li et al., 2021a), UDoc (Gu et al., 2021), and MGDoc (Wang et al., 2022b), which are pre-trained with a cross-modal encoder to capture the relationship between visual and textual features. These models incorporate spatial information by fusing position embeddings at the output layer of their encoders, instead of the input layer. Additionally, OCR-free models (Kim et al., 2022; Tang et al., 2023) tackle document understanding as a se-

quence generation problem, unifying multiple tasks through an image-to-sequence generation network.

While these pre-trained models demonstrate promising performance on downstream applications, their robustness to different types of OOD data, the influence of pre-training and fine-tuning, and the value of different modalities (*e.g.* spatial, textual, and visual) for document OOD detection remain largely unexplored.

## 2.2 Out-of-Distribution Detection

OOD detection has been extensively studied for open-world multi-class classification with natural image and text inputs, where the goal is to derive an OOD score that separates OOD from ID samples. A plethora of methods are proposed for deep neural networks, where the OOD scoring function is typically derived based on logits (without softmax scaling) (Hendrycks et al., 2022), softmax outputs (Liang et al., 2018; Hsu et al., 2020; Huang and Li, 2021; Sun et al., 2021), gradients (Huang et al., 2021), and feature embeddings (Tack et al., 2020; Fort et al., 2021; Ming et al., 2023). Despite their impressive performance on natural images and texts, it is underexplored if the results are transferrable to the document domain. A recent work (Larson et al., 2022) studied OOD detection for documents but only explored a limited number of models and OOD detection methods. The impacts of pre-training, fine-tuning, and spatial information remain unknown. In this work, we aim to provide a comprehensive and finer-grained analysis to shed light on the key factors for OOD robustness in the document domain.

**Notations.** Following prior works on OOD detection with large-scale pre-trained models (Ming et al., 2022a; Ming and Li, 2023), the task of OOD detection is defined with respect to the downstream dataset, instead of the pre-training data which is often hard to characterize. In document classification, we use $\mathcal{X}^{\text{in}}$ and $\mathcal{Y}^{\text{in}} = \{1, \ldots, K\}$ to denote the input and label space, respectively. Let $\mathcal{D}^{\text{in}} = \{(\boldsymbol{x}_i^{\text{in}}, y_i^{\text{in}})\}_{i=1}^{N}$ be the ID dataset, where $\boldsymbol{x} \in \mathcal{X}^{\text{in}}$, and $y^{\text{in}} \in \mathcal{Y}^{\text{in}}$. Let $\mathcal{D}^{\text{out}} = \{(\boldsymbol{x}_i^{\text{out}}, y_i^{\text{out}})\}_{i=1}^{M}$ denote an OOD test set where $y^{\text{out}} \in \mathcal{Y}^{\text{out}}$, and $\mathcal{Y}^{\text{out}} \cap \mathcal{Y}^{\text{in}} = \emptyset$. We express the neural network model $f := g \circ h$ as a composition of a feature extractor $h : \mathcal{X} \to \mathbb{R}^d$ and a classifier $g : \mathbb{R}^d \to \mathbb{R}^K$, which maps the feature embedding of an input to $K$ real-valued numbers known as logits. During inference time, given an input $\boldsymbol{x}$, OOD detection can be formulated as:

$$G_\gamma(\boldsymbol{x}; h, g) = \begin{cases} \text{ID} & S(\boldsymbol{x}; h, g) \geq \gamma \\ \text{OOD} & S(\boldsymbol{x}; h, g) < \gamma \end{cases},$$

where $S(\cdot)$ is a scoring function that measures OOD uncertainty. In practice, the threshold q$\gamma$ is often chosen so that a high fraction of ID data (*e.g.*, 95%) is above the threshold.

**OOD detection scores.** We focus on two major categories of computationally efficient OOD detection methods[1]: logit-based methods derive OOD scores from the logit layer of the model, while distance-based methods directly leverage feature embeddings, as shown in Fig. 1. We describe a few popular methods for each category as follows.

- **Logit-based:** Maximum Softmax Probability (MSP) score (Hendrycks and Gimpel, 2017) $S_{\text{MSP}} = \max_{i \in [K]} e^{f_i(\boldsymbol{x})} / \sum_{j=1}^{K} e^{f_j(\boldsymbol{x})}$ naturally arises as a classic baseline as models often output lower softmax probabilities for OOD data; Energy score (Liu et al., 2020): $S_{\text{Energy}} = \log \sum_{i \in [K]} e^{f_i(\boldsymbol{x})}$ utilizes the Helmholtz free energy of the data and theoretically aligns with the logarithm of the ID density; the simple MaxLogit score (Hendrycks et al., 2022): $S_{\text{Maxlogit}} = \max_{i \in [K]} f_i(\boldsymbol{x})$ has demonstrated promising performance on large-scale natural image datasets. We select the above scores due to their simplicity and computational efficiency. In addition, recent studies demonstrate that such simple scores are particularly effective with large-scale pre-trained models in vision (Fort et al., 2021) and vision-language domains (Ming et al., 2022a; Bitterwolf et al., 2023). We complement previous studies and investigate their effectiveness for documents.

- **Distance-based:** Distance-based methods directly leverage feature embeddings $\mathbf{z} = h(\mathbf{x})$ based on the idea that OOD inputs are relatively far away from ID clusters in the feature space, compared to ID inputs. Distance-based methods can be characterized as parametric and non-parametric. Parametric methods such as Mahalanobis score (Lee et al., 2018; Sehwag et al., 2021) assume ID embeddings follow class-conditional Gaussian distributions and use the Mahalanobis distance as the distance metric. On the other hand, non-parametric methods such as KNN+ (Sun et al., 2022) use cosine similarity as the distance metric.

---

[1] We also investigate gradient-based methods such as Grad-Norm (Huang et al., 2021) in Appendix C.

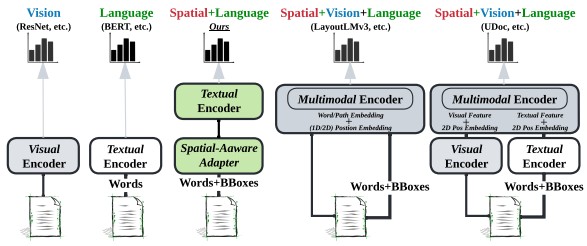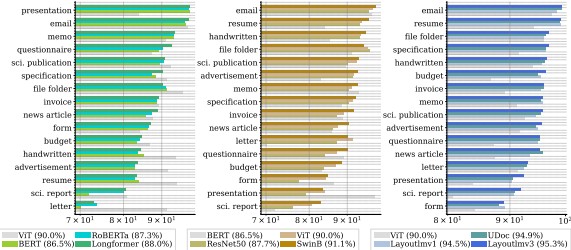

(a) Illustration of common structures for document pre-training and classification.

(b) A detailed comparison of per-category accuracy on the RVL-CDIP test set.

Figure 2: (**Left**) Illustration of models for document pre-training and classification, with our proposed spatial-aware models in green blocks. Modality information is also shown atop each architecture. (**Right**) Evaluating fine-tuning performance for document classification of pre-trained models. Models are grouped into several categories (from left to right): language-only, vision-only, and multi-modal. For comparison, the performance of corresponding models in other groups is shown in gray. The average accuracy for each model is indicated in the parenthesis.

**Evaluation metrics.** To evaluate OOD detection performance, we adopt the following commonly used metrics: the Area Under the Receiver Operating Characteristic (AUROC), False Positive Rate at 95% Recall (FPR95), and the multi-class classification accuracy (ID Acc).

## 3 Experimental Setup

**Models.** Fig. 2a summarizes common structures for document pre-training and classification models[2]. While documents typically come in the form of images (Harley et al., 2015), an OCR system can be used to extract words and their coordinates from the input image. Therefore, models can use single-modal or multi-modal information. We categorize these models according to the input modalities into the following groups: (1) models using only visual features, (2) models using solely textual features, (3) models incorporating both visual and textual features, and (4) models integrating additional spatial (especially layout) information. Further details can be found in Appendix A.

- **Vision-only:** Document classification can be viewed as a standard image classification problem. We consider ResNet-50 (He et al., 2016) and ViT (Fort et al., 2021) as exemplar document image classification models. We adopt two common pre-training settings: (1) only pre-trained on ImageNet (Deng et al., 2009) and (2) further pre-trained on IIT-CDIP (Lewis et al., 2006) with masked image modeling (MIM)[3]. After pre-training, we append a classifier for fine-tuning.

- **Text-only:** Alternatively, we can view document classification as text classification since documents often contain text blocks. To this end, we use RoBERTa (Liu et al., 2019) and Longformer (Beltagy et al., 2020) as the backbones. RoBERTa can handle up to 512 input tokens while Longformer can handle up to 4,096 input tokens. We pre-train the language models with masked language modeling (MLM) on IIT-CDIP extracted text corpus.

- **Text+Layout:** Layout information plays a crucial role in the document domain, as shown in Fig. 3. To investigate the effect of layout information, we adopt LayoutLM as the backbone. We will show that spatial-aware models demonstrate promising OOD detection performance. However, such specialized models can be computationally expensive. Therefore, we propose a new spatial-aware adapter, a small learned module that can be inserted within a pre-trained language model such as RoBERTa and transforms it into a spatial-aware model, which is computationally efficient and competitive for both ID classification and OOD detection (Sec. 5.2).

- **Vision+Text+Layout:** For comprehensiveness, we consider LayoutLMv3 and UDoc, which are large and computationally intensive. Both models are pre-trained on the full IIT-CDIP for fairness. These models utilize different input granularities and modalities, including textual, visual, and spatial information for document tasks.

---

[2]Apart from document classification, in the Appendix B, we also investigate OOD detection for two entity-level tasks: document entity recognition and document object detection.

[3]Note that the document classification dataset we used in

this paper, RVL-CDIP (Harley et al., 2015), is a subset of IIT-CDIP. Hence, unless otherwise specified, the IIT-CDIP pre-training data used in this paper excludes RVL-CDIP.

**Constructing ID and OOD datasets.** We construct ID datasets from RVL-CDIP (Harley et al., 2015), where 12 out of 16 classes are selected as ID classes. Dataset details are in Appendix A. We consider two OOD scenarios: in-domain and out-domain, based on the content (*e.g.*, words, background) and layout characteristics.

- **In-domain OOD:** To determine the OOD categories, we analyzed the performance of recent document classification models on the RVL-CDIP test set. Fig. 2b shows the per-category test accuracy of various models. Naturally, for the classes the models perform poorly on, we may expect the models to detect such inputs as OOD instead of assigning a specific ID class with low confidence. We observe that the 4 categories (*letter*, *form*, *scientific report*, and *presentation*) result in the worst performance across most of the models with different modalities. We use these as OOD categories and construct the OOD datasets accordingly. The ID dataset is constructed from the remaining 12 categories, which we refer to as *in-domain* OOD datasets, as they are also sourced from RVL-CDIP.

- **Out-domain OOD:** In the open-world setting, test inputs can have significantly different color schemes and layouts compared to ID samples. To mimic such scenarios, we use two public datasets as *out-domain* OOD test sets: NJU-Fudan Paper-Poster Dataset (Qiang et al., 2019) and CORD (Park et al., 2019). NJU-Fudan Paper-Poster Dataset contains scientific posters in digital PDF format[4]. CORD is a receipt understanding dataset with significantly different inputs compared to RVL-CDIP. As shown in Fig. 3, receipt images can be challenging and require models to handle not only textual but also visual and spatial information.

We further support our domain selection using OTDD (Alvarez-Melis and Fusi, 2020), a flexible geometric method for comparing probability distributions, which enables us to compare any two datasets regardless of their label sets. We observe a clear gap between in-domain and out-domain data, which aligns with our data selection. Further details can be found in Appendix A.1.

---

[4]Extracted using https://github.com/pymupdf/PyMuPDF

# 4 Analyzing OOD Reliability for Documents

## 4.1 OOD Detection Without Fine-Tuning

In this section, we begin by examining the influence of pre-training datasets on zero-shot OOD detection. For each model, we adopt the same pre-training objective while adjusting the amount of pre-training data. Specifically, we increase the data diversity by appending 10, 20, 40, and 100% of randomly sampled data from IIT-CDIP dataset (around 11M) and pre-train each model. After pre-training, we measure the OOD detection performance with KNN+ score based on feature embeddings.

We observe that: (1) for out-domain OOD data (Fig. 4a, right), increasing the amount of pre-training data can significantly improve the zero-shot OOD detection performance (w.o. fine-tuning) for models across different modalities. Our hypothesis is that pre-training with diverse data is beneficial for coarse-grained OOD detection, such as inputs from different domains (*e.g.*, color schemes). (2) For in-domain OOD inputs, even increasing the amount of pre-training data by over 40% provides negligible improvements (Fig. 4a, left). This suggests the necessity of fine-tuning for improving in-domain OOD detection performance (Fig. 6).

We further explore a more restricted setting for zero-shot OOD detection where potential OOD categories are removed from the pre-training dataset IIT-CDIP. First, we use LayoutLM fine-tuned on RVL-CDIP to predict labels for all documents in IIT-CDIP. Fig. 4b summarizes the distribution of the predicted classes on IIT-CDIP. Next, we remove the "OOD" categories from IIT-CDIP and pre-train two models (RoBERTa and LayoutLM) with 10, 20, 40, and 100% of randomly sampled data from the filtered IIT-CDIP (dubbed III-CDIP$^-$), respectively. The zero-shot OOD performance for in-domain and out-domain OOD is shown in Fig. 4c[5]. For RoBERTa, we observe similar trends as in Fig. 4a, where increasing the amount of pre-training data improves zero-shot OOD detection performance for *out-domain* data. However, the zero-shot performance of LayoutLM benefits from a larger pre-training dataset. In particular, given the same amount of pre-training data, LayoutLM consistently outperforms RoBERTa for both in-domain and out-domain OOD detection, which suggests that *spatial information* can be essential

---

[5]Note that we do not show 0% in Fig. 4c since we pre-train LayoutLM from scratch.

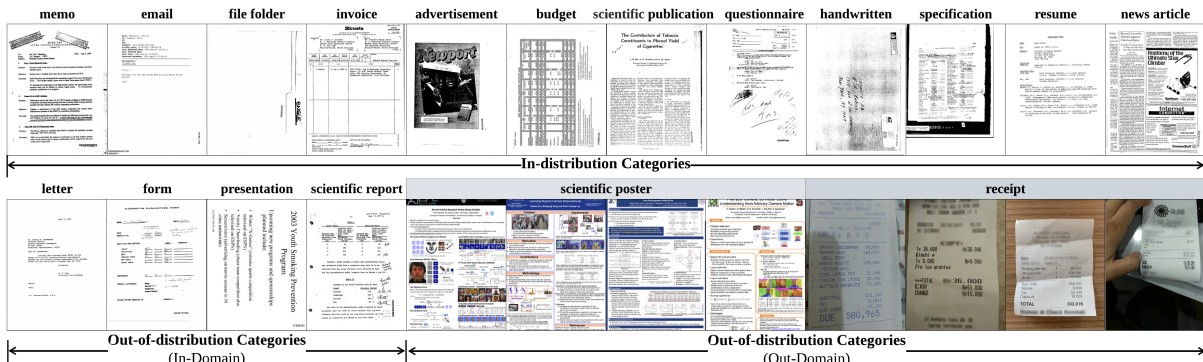

Figure 3: (**Top**) Examples of ID inputs sampled from RVL-CDIP (top). (**Bottom**) In-domain OOD from RVL-CDIP, and out-domain OOD from *Scientific Poster* and *Receipts*.

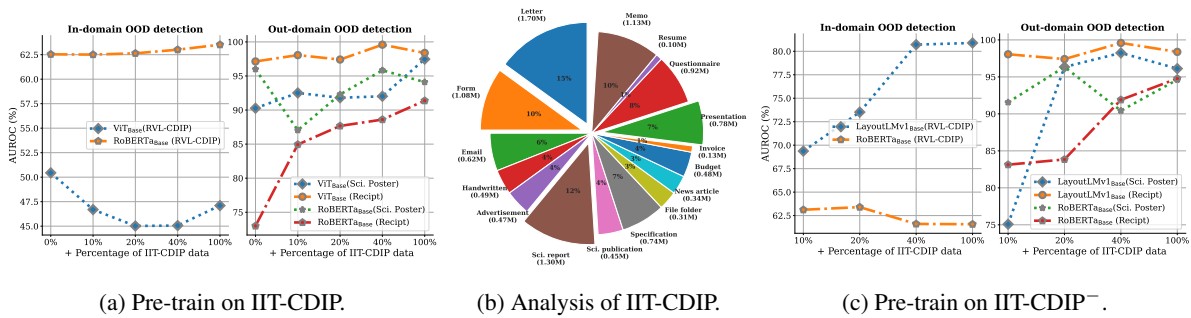

(a) Pre-train on IIT-CDIP.  (b) Analysis of IIT-CDIP.  (c) Pre-train on IIT-CDIP⁻.

Figure 4: The impact of pre-training data on zero-shot OOD detection performance. IIT-CDIP⁻ denotes the filtered pre-training data after removing the "*OOD*" categories.

for boosting the OOD reliability in the document domain. Motivated by the above observations, we dive deeper and analyze spatial-aware models next.

While pre-trained models exhibit the capability to differentiate data from various domains as a result of being trained on a diverse range of data. We observe that achieving more precise separation for in-domain OOD inputs remains difficult. Given this observation, we further analyze the impacts of fine-tuning for OOD detection with fixed pre-training datasets in the next section. By combining pre-trained models with a simple classifier and fine-tuning on RVL-CDIP (ID), we find that fine-tuning is advantageous in enhancing the OOD detection performance for both types of OOD samples.

### 4.2 The Impact of Fine-Tuning on Document OOD Detection

Recent document models are often pre-trained on a large-scale dataset and adapted to the target task via fine-tuning. To better understand the role of fine-tuning, we explore the following questions: 1) *How does fine-tuning impact OOD reliability for in-domain and out-domain OOD inputs? 2) How does model modality impact the performance?*

We consider a wide range of models pre-trained on pure-text/image data (*e.g.*, ImageNet and Wikipedia) described in Appendix A.3. During fine-tuning, we combine pre-trained models with a simple classifier and fine-tune on RVL-CDIP (ID). For models before and after fine-tuning, we extract the final feature embeddings and use a distance-based method KNN+ (Sun et al., 2022) for OOD detection. The results are shown in Fig. 6. We observe the following trends. First, fine-tuning largely improves OOD detection performance for both in-domain and out-domain OOD data. The same trend holds broadly across models with different modalities. Second, the improvement of fine-tuning is less significant for out-domain OOD data. For example, on Receipt (out-domain OOD), the AUROC for pre-trained ViT model is 97.13, whereas fine-tuning only improves by 0.79%. This suggests that pre-trained models do have the potential to separate data from different domains due to the diversity of data used for pre-training, while it remains hard for pre-trained models to perform finer-grained separation for in-domain OOD inputs. Therefore, fine-tuning is beneficial for improving OOD detection performance for both types of OOD

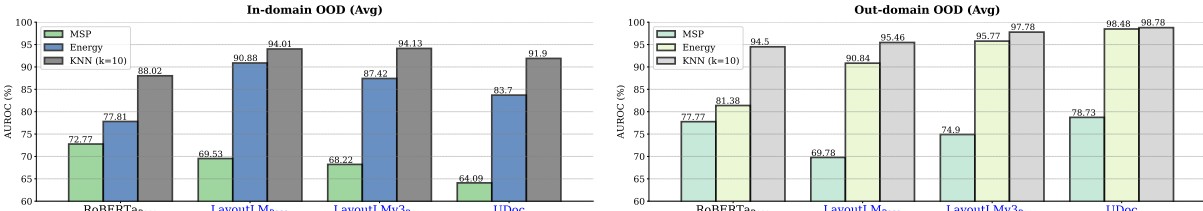

Figure 5: Comparison between representative feature-based scores and logit-based scores for spatial-aware and non-spatial-aware models. Spatial-aware models are colored in blue.

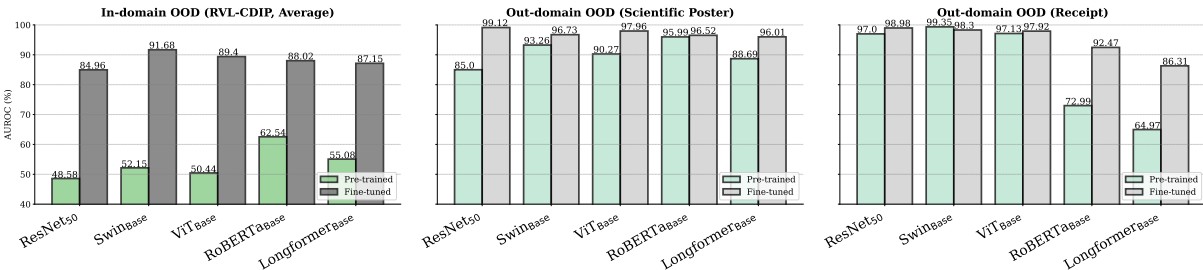

Figure 6: OOD detection performance for pre-trained models w. and w.o. fine-tuning. We use a distance-based method KNN+ as the OOD scoring function. Fine-tuning significantly improves performance for both in and out-domain OOD data.

samples. To further validate our conclusion, we consider two additional in-domain OOD settings for our analysis: (1) selecting the classes the model performs well on, as in-domain OOD categories; (2) randomly selecting classes as OOD categories (Appendix A.2). We find that fine-tuning improves OOD detection for both settings, further verifying our observations.

Next, we take a closer look at the impact of model modality on out-domain OOD detection. As shown in Fig. 6 (mid and right), both vision and text-based models demonstrate strong reliability against scientific posters (OOD). However, vision-based models display stronger performance than text-based models for Receipts (OOD). This can be explained by the fact that ViT was first pre-trained on ImageNet while scientific posters and receipts contain diverse visual information such as colors and edges for vision models to utilize (see Fig. 3). On the other hand, although fine-tuning text-based models largely improves the detection performance compared to pre-trained counterparts, utilizing only textual information can be inherently limited for out-domain OOD detection.

# 5 The Importance of Spatial-Awareness

In previous sections, we mainly focus on mainstream text-based and vision-based models for in- and out-domain OOD detection. Next, we consider

models tailored to document processing, which we refer to as *spatial-aware models*, such as LayoutLMv3 and UDoc. Given fine-tuned models, we compare the performance of logit-based and distance-based OOD scores.

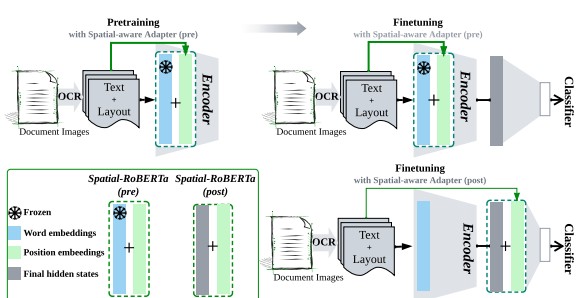

Figure 7: Illustration of our spatial-aware adapter for language models. We present 2 adapter designs (marked in green box): *(1)* insert the adapter into the word embedding layer during pre-training and fine-tuning; *(2)* insert the adapter into the output layer for fine-tuning only. For the first design, we freeze the word embedding layer and learn the adapter and transformer layers.

## 5.1 Analysis of Spatial-Aware Models

We summarize key comparisons in Fig. 5, where we use MSP and Energy as exemplar logit-based scores and KNN+ as the distance-based score. Full results are in Appendix C. We can see that the simple KNN-based score (KNN+) consistently outperforms logit-based scores for both in-domain and

out-domain OOD data across different models with different modalities. This is *in contrast with* recent works that investigate large-scale pre-trained models in the vision-language domain, where logit-based scores demonstrate strong OOD detection performance (Fort et al., 2021). As documents are distinct from natural image-text pairs, observations in the vision-language domain do not seamlessly translate to the document domain. Moreover, spatial-aware models demonstrate stronger OOD detection performance for both in and out-domain OOD. For example, with the best scoring function (KNN+), LayoutLMv3 improves the average AUROC by 7.09% for out-domain OOD and 7.54% for in-domain OOD data compared to RoBERTa. This further highlights the value of spatial information for improving OOD robustness for documents.

Despite the impressive improvements brought by spatial-aware models, acquiring a large-scale pre-training dataset that includes spatial information remains challenging. In contrast, there is a growing abundance of pre-trained language models that are based on textual data. This motivates us to explore the possibility of leveraging these pre-trained language models by training an adapter on a small dataset containing document-specific information. By adopting this approach, we can effectively utilize existing models while minimizing the time and cost required for training.

## 5.2 Towards Effective Spatial-Aware Adapter

During our investigation into the effects of model modality, pre-training, and fine-tuning on various types of OOD inputs, we find that spatial/layout information plays a critical role in the document domain. However, existing pre-training models such as LayoutLM series, SelfDoc, and UDoc do not fully leverage the benefits of well-pre-trained language models. This raises the question of whether a large-scale language model, such as RoBERTa, can be adapted to detect OOD documents effectively. In this section, we demonstrate that incorporating an adapter module that accounts for spatial information with transformer-based pre-trained models can achieve strong performance with minimal changes to the code. To the best of our knowledge, this is the first study to apply the adapter idea to documents.

**Spatial-aware adapter.** Given a pre-trained language model such as RoBERTa, we propose an adapter that utilizes spatial information. We consider two potential designs: 1) the adapter is ap-

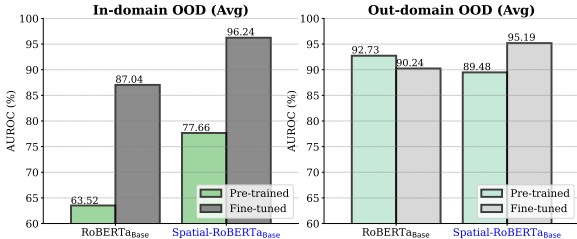

Figure 8: Comparison of OOD detection performance of Spatial-RoBERTa and RoBERTa. All models are initialized with public pre-trained checkpoints trained on purely textual data and further pre-trained on IIT-CDIP. The only difference is that Spatial-RoBERTa has an additional spatial-ware adapter and takes word bounding boxes as additional inputs.

pended to the word embedding layer, denoted as Spatial-RoBERTa (pre), which requires both pre-training and fine-tuning. This architecture is illustrated in the top row of Fig. 7. 2) The adapter is appended to the final layer of the text encoder, denoted as Spatial-BoBERTa (post), which only requires fine-tuning as the model can utilize the pre-trained textual encoder, as shown in the bottom row of Fig. 7.

For Spatial-RoBERTa (pre), we freeze the word embedding layer during pre-training for several considerations: 1) word embeddings learned from large-scale corpus already cover most of those words from documents; 2) pre-training on documents without strong language dependency may not help improve word embeddings. For example, in semi-structured documents (*e.g.*, forms, receipts), language dependencies are not as strong as in text-rich documents (*e.g.*, letters, resumes), which may degenerate the learned word representations. In practice, each word has a normalized bounding box $(x_0, y_0, x_1, y_1)$, where $(x_0, y_0)$ / $(x_1, y_1)$ corresponds to the position of the upper left / lower right in the bounding box. To encode positional information, we employ four position embedding layers, where each layer= encodes one coordinate (*e.g.*, $x_0$) and produces a corresponding position embedding. The special tokens ([CLS], [SEP], and [PAD]) are attached with an empty bounding box $(0, 0, 0, 0)$. As depicted in the top row of Fig. 7, the spatial-aware word embeddings are formed by adding position embeddings to their corresponding word embeddings.

For Spatial-RoBERTa (post), position embeddings are added through late fusion in the final hidden states during fine-tuning without affecting the

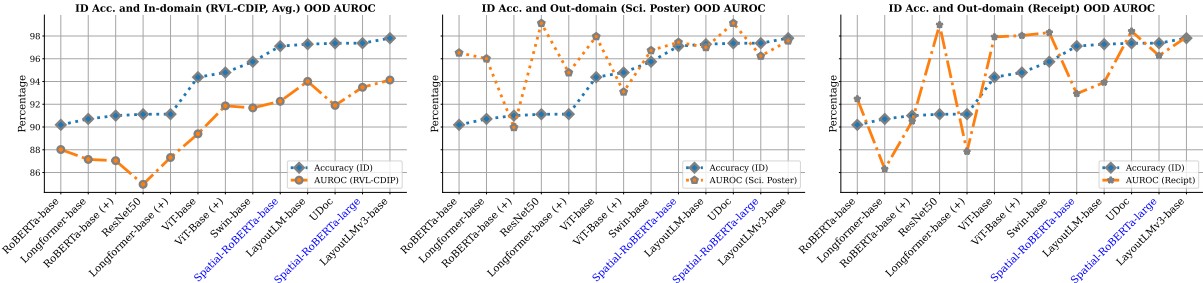

Figure 9: Correlation between ID accuracy and OOD detection performance. For most models, ID accuracy is positively correlated with OOD detection performance. Language models with spatial-aware adapters (highlighted in blue) achieve significantly higher ID accuracy and stronger OOD robustness (in AUROC) compared to language models without adapters. Here, $(+)$ represents further pre-training on the IIT-CDIP dataset.

pre-trained encoder. Our experiments demonstrate that introducing spatial-aware adapters during pre-training yields better results than only adding position embeddings during fine-tuning. For additional details[6], please refer to Appendix C. In the following, we focus on analyzing Spatial-RoBERTa (pre) and comparing both ID and OOD performance with that of the pure-text pre-trained RoBERTa.

**Spatial-RoBERTa significantly outperforms RoBERTa.** To verify the effectiveness of Spatial-RoBERTa, we compare the OOD detection performance of pre-trained and fine-tuned models. The results are shown in Fig. 8, where OOD performance is based on KNN+ (K=10). Full results can be seen in Table 6. Spatial-RoBERTa significantly improves the OOD detection performance, especially after fine-tuning. For example, compared to RoBERTa (base), Spatial-RoBERTa (base) improves AUROC significantly by 4.24% averaged over four in-domain OOD datasets. This further confirms the importance of spatial information for OOD detection in the document domain.

**Spatial-RoBERTa is competitive for both ID classification and OOD detection.** Beyond OOD detection performance, we also examine the multi-class ID classification accuracy and plot the two metrics for all models with different modalities in Fig. 9. We can clearly observe a positive correlation between ID accuracy and OOD detection performance (measured by AUROC) for both in-domain and out-domain OOD data. Moreover, spatial-aware models display superior ID accuracy and OOD robustness compared to text-only and

vision-only models. Overall, Spatial-RoBERTa greatly improves upon RoBERTa and matches the performance of models with more complex and specialized architectures such as LayoutLM. Specifically, Spatial-RoBERTa$_{Large}$ achieves 97.37 ID accuracy, which is even higher than LayoutLM (97.28) and UDoc (97.36).

To summarize, our spatial-aware adapter effectively adapts pre-trained transformer-based text models to the document domain, improving both ID and OOD performance. In addition, by freezing the original word embeddings during pre-training, the models (Spatial-RoBERTa$_{Base}$ and Spatial-RoBERTa$_{Large}$) are parameter-efficient and thus reduce the training cost.

## 6 Conclusions

In this work, we provide a comprehensive and in-depth study on the impacts of pre-training, fine-tuning, model-modality, and OOD scores on a broad variety of document OOD detection tasks. We present novel insights on document OOD detection, which are under-explored or in contrast with OOD detection works based on vision-language models. In particular, we highlight that spatial information is critical for OOD detection in documents. We further propose a spatial-aware adapter as an add-on module to transformer-based models. Our module adapts pre-trained language models to the document domain. Extensive experiments on a broad range of datasets verify the effectiveness of our design. We hope our work will inspire future research toward improving OOD robustness for reliable document understanding.

---

[6]Spatial-RoBERTa$_{Base}$ (pre) incorporates position information during both pre-training and fine-tuning, while Spatial-RoBERTa$_{Base}$ (post) only inserts the adapter into the output layer for fine-tuning.

# 7 Limitations

In this work, our main focus is on OOD detection for document understanding, with a specific emphasis on the context of document classification. As OOD detection based on document pre-trained models remains largely underexplored, we believe establishing an in-depth and extensive study of OOD detection for document classification would be a valuable stepping stone towards more complex tasks. Apart from document classification, in the Appendix B, we also investigate OOD detection for two entity-level tasks: document entity recognition and document object detection. We leave a more comprehensive treatment for future works.

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

## A Dataset and Model Details

### A.1 Datasets

The full RVL-CDIP dataset consists of 320K/40K/40K training/validation/testing images under 16 categories. We select 12 of them as the ID (In-domain) data. We employ the Google OCR engine[7] to extract the text and layout information, which provides tokens, text blocks and the corresponding bounding boxes.

### A.2 Quantifying OOD Dataset Construction

The distance between datasets can be measured via Optimal Transport Dataset Distance (OTDD)[8]. We visualize the OTDD distance between ID and the OOD (both in-domain and out-domain) data in Fig. 10a, where we highlight the in-domain OOD data in blue and the out-domain OOD data in green. Specifically, we randomly sample 1000 images from each dataset and calculate the average distance between pairs of datasets. We can see a significant gap between the OTDD of in-domain OOD data and out-domain OOD data. To make the analysis more thorough, we consider two additional in-domain OOD settings: (1) select the classes the model performs well as OOD data; (2) randomly select classes as OOD data. The results are shown in Fig. 10b and Fig. 10c. We can see that the distance between ID and in-domain OOD is similar to the original scheme (Fig. 10a). This suggests that most in-domain OOD categories are not far from ID data.

While this paper represents an initial endeavor, we hope that our work will serve as a stepping stone towards constructing more comprehensive and diverse OOD benchmarks in the document domain, akin to those available in the NLP and natural image domain.

### A.3 Models and Training Details

All models reported in Fig. 2b, except UDoc, are initialized with pre-trained weights from Huggingface[9] and fine-tuned on the full RVL-CDIP training set. During fine-tuning, we train these models on RVL-CDIP with the cross-entropy loss. The models were optimized with Adam optimizer (Kingma and Ba, 2014) for 30 epochs with a batch size of 50 and a learning rate of $2 \times 10^{-5}$ on 8 A100 GPUs.

---

[7] https://cloud.google.com/vision/docs/ocr
[8] https://github.com/microsoft/otdd
[9] https://huggingface.co/models

The following are the hyperparameters of the models used in our paper:

**Text-only:**

- **BERT** and **RoBERTa**: We adopt RoBERTa$_{Base}$ (12 layers) and BERT$_{Base}$ (12 layers) as backbones and set the maximum sequence length to 512. For RoBERTa, the classifier consists of two linear layers followed by a tanh activation function.

- **Longformer$_{Base}$**: We also employ Longformer$_{Base}$ (12 layers) as the backbone and set the maximum sequence length to 4,096.

**Vision-only:**

- **ResNet50**: We adopt ResNet50 pre-trained on ImageNet-1k as the backbone. We fine-tune the model at a resolution of 224×224.

- **ViT**: We consider ViT$_{Base}$ (vit-base-patch16-224, pre-trained on ImageNet-21k) as the backbon and fine-tune at a resolution of 224×224.

- **SwinB**: We also use the Swin Transformer (swin-base-patch4-window7-224-in22k, pre-trained on ImageNet-21k) as the backbone and fine-tune the model at a resolution of 224×224.

**Text+Layout:**

- **LayoutLMv1**: This model employs the LayoutLM (layoutlm-base-uncased, 12 layers, pre-trained on IIT-CDIP) as the backbone. We set the maximum sequence length to 512.

- **Spatial-RoBERTa$_{Base}$ (Pre)**: This model combines our spatial-aware adapter to the pre-trained RoBERTa$_{Base}$ model. The adapter is applied to the word embedding layer. We freeze the pre-trained word embeddings and optimize the spatial-aware adapter and transformers.

- **Spatial-RoBERTa$_{Base}$ (Post)**: Instead of inserting the spatial-aware adapter in the input layer, this model integrates the spatial-aware adapter at the output layer of the transformer.

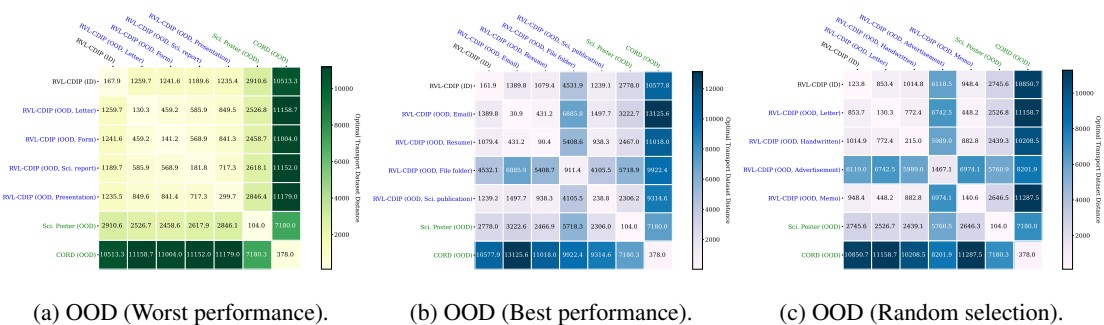

(a) OOD (Worst performance).    (b) OOD (Best performance).    (c) OOD (Random selection).

Figure 10: Visualization of optimal transport dataset distance for ID and OOD (in-domain and out-domain) datasets. We highlight the in-domain OOD data in blue and the out-domain OOD data in green.

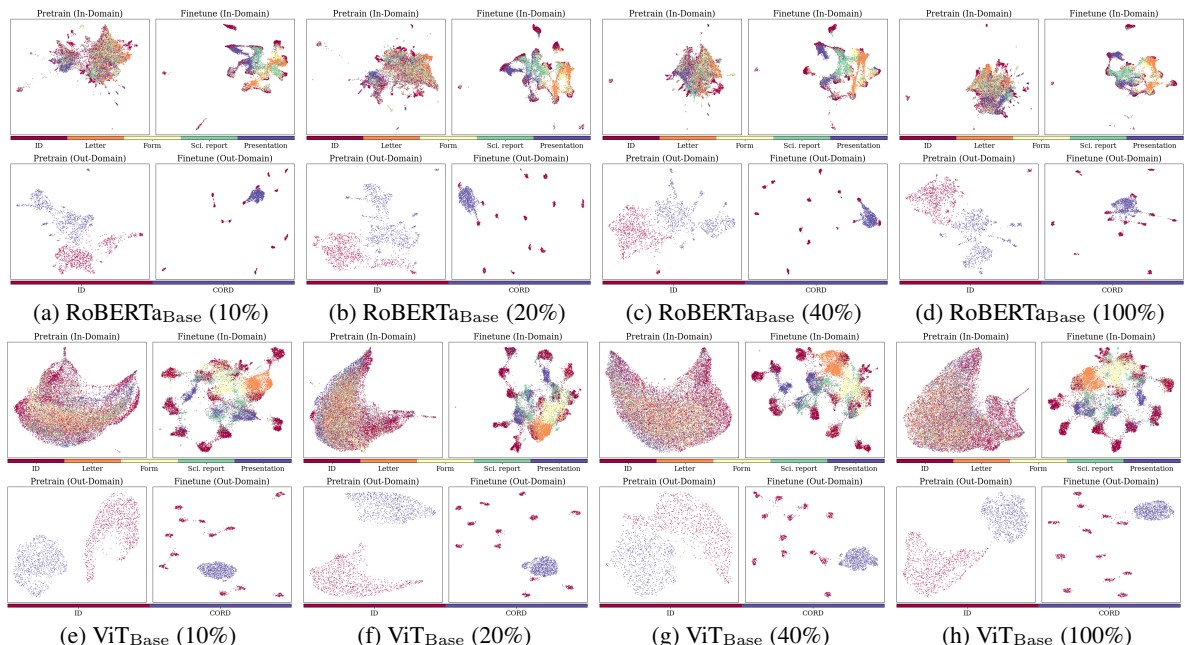

(a) RoBERTa$_{Base}$ (10%)    (b) RoBERTa$_{Base}$ (20%)    (c) RoBERTa$_{Base}$ (40%)    (d) RoBERTa$_{Base}$ (100%)

(e) ViT$_{Base}$ (10%)    (f) ViT$_{Base}$ (20%)    (g) ViT$_{Base}$ (40%)    (h) ViT$_{Base}$ (100%)

Figure 11: Feature visualization for pre-trained (with different numbers of pre-training data) and fine-tuned models. We show both in-domain (RVL-CDIP) and out-domain (CORD) OOD datasets.

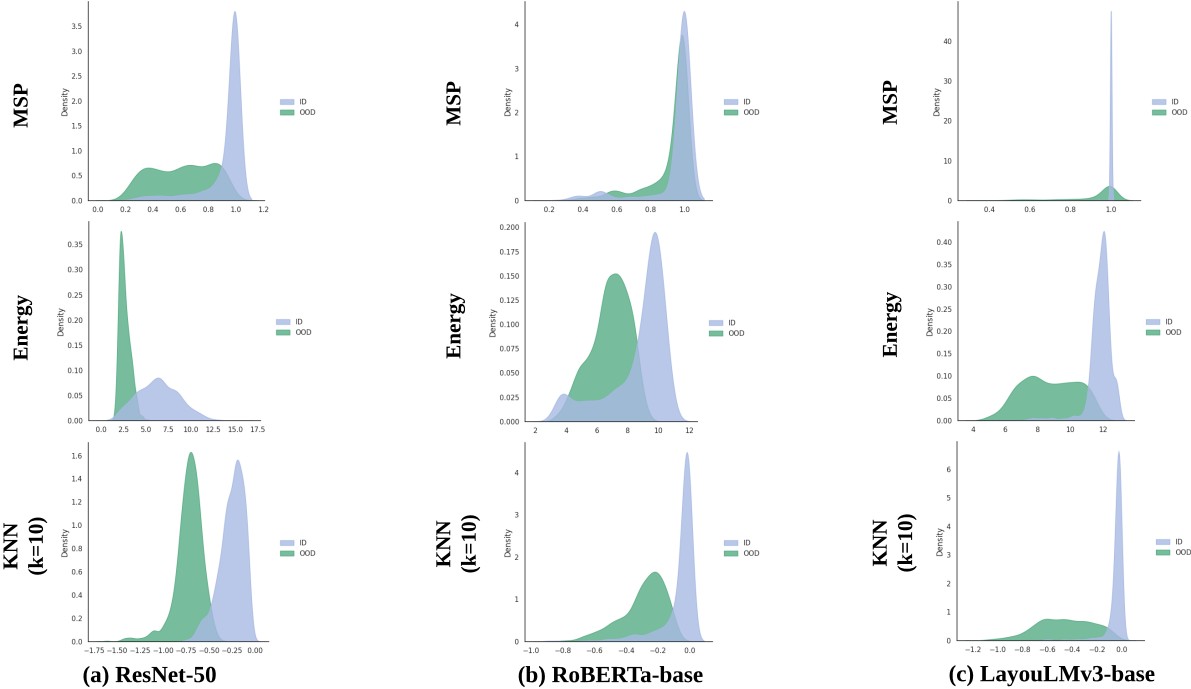

Figure 12: MSP, Energy, KNN, and Maha score histogram distributions of ID (*blue*) and OOD (*green*) inputs derived from fine-tuned ResNet-50, RoBERTa, and LayoutLMv3. The KNN scores calculated from both vision and language models naturally form smooth distributions. In contrast, MSP and Maha scores for both in- and out-of-distribution data concentrate on high values. Overall our experiments show that using feature space makes the scores more distinguishable between and out-of-distributions and, as a result, enables more effective OOD detection.

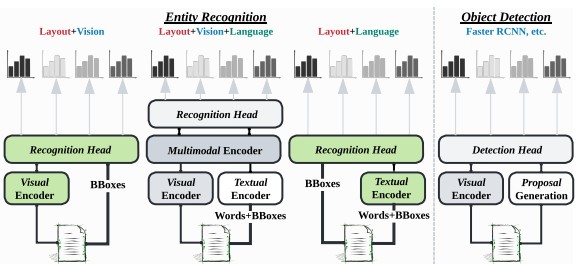

Figure 13: The network architectures in green blocks are our proposed models. We also show the modality information on top of each architecture.

**Vision+Text+Layout:**

- **LaytouLMv3**: We use LayoutLMv3 (layoutlmv3-base, 12 layers, pre-trained on IIT-CDIP) as the backbone.

- **UDoc**: We use a slight variant of UDoc with the only difference in the sentence encoder, where we adopt a smaller version of the pre-trained sentence encoder (all-MiniLM-L6-v2, 6 layers) instead of the larger sentence encoder (bert-base-nli-mean-tokens, 12 layers).

## B  Beyond Document Classification

In the main paper, we mainly focus on document classification to provide a thorough and in-depth analysis. In this section, we go beyond document classification and explore OOD detection for two entity-level tasks in documents: document entity recognition and document object detection. It is natural to detect and recognize basic units in documents such as text, tables, and figures. Document entity recognition aims to predict the label for each semantic entity with given bounding boxes. Document object detection is an object detection task for document images. Specifically, we denote the input as $x$, the bounding box coordinates associated with object instances in the image as $\boldsymbol{b} \in \mathbb{R}^4$, and use the model with parameters $\theta$ to model the bounding box regression $p_\theta(\boldsymbol{b}|x)$ and the label classification $p_\theta(y|x, \boldsymbol{b})$. Given a test input $\hat{x}$, the OOD detection scoring function for entity detection and recognition can be unified as $S(\hat{x}, \hat{\boldsymbol{b}})$, where $\hat{\boldsymbol{b}}$ denotes the object instance predicted by the object detector. In particular, for document entity recognition, since the bounding boxes are provided, the OOD score can be simplified as $S(\hat{x}, \bar{\boldsymbol{b}})$, where $\bar{\boldsymbol{b}}$ is the given object instance.

**Document Object Detection.** For document object detection, we use PubLayNet as the ID dataset and construct the OOD dataset from IIIT-AR-13K. Unlike PubLayNet, where the documents are scientific articles, IIIT-AR-13K is a dataset for graphical object detection in business documents (*e.g.*, annual reports), thus there exists an obvious domain gap. We select *natural images* as the OOD entity and filter images that contain the OOD entity. Two object detection models are considered in this paper: *(1)* Vanilla Faster-RCNN with ResNet-50 visual backbone, and *(2)* Faster-RCNN with VOS (Du et al., 2022), a recent unknown-aware learning framework to improve OOD detection performance for natural images. Following the original paper, we use 1,000 samples for each ID class to estimate the class-conditional Gaussian statistics. The models are trained for 180k iterations with a base learning rate of 0.01 and a batch size of 8 using the Detectron2 framework (Wu et al., 2019). The performance of the models is measured using the mean average precision (MAP) @ intersection over union (IOU) [0.50:0.95] of bounding boxes.

**Document Entity Recognition.** For entity recognition, we construct ID and OOD datasets from FUNSD. Each semantic entity includes a list of words, a label, and a bounding box. The standard label set for this dataset contains four categories: *question*, *answer*, *header*, and *other*. In this paper, we select entities labeled as *other* or *header* as OOD data, and the entities belonging to the other three categories as ID. Instead of treating entity recognition as a named-entity recognition problem, we follow UDoc and solve this problem at the semantic region level. We replace the sentence encoder in UDoc with a smaller sentence encoder (all-MiniLM-L6-v2[10]) from Huggingface (Wolf et al., 2019). We also have the following model variants to verify the effectiveness of the combination of modalities: textual-only, visual-only, textual+spatial, visual+spatial, and visual+textual+spatial.

We provide details on datasets and models as follows.

### B.1 Datasets

The original FUNSD (Jaume et al., 2019) dataset contains 149 training and 50 testing images. For document entity recognition, we treat entities with the category *other*/*header* as OOD entities. After

---

[10] https://huggingface.co/sentence-transformers

the split, if we consider *other* as OOD, we have a total of 8,330 ID and 1,019 OOD entities. Otherwise, if we consider *header* as OOD, we have 8,981 ID and 368 OOD entities in total.

For document object detection, we consider PubLayNet (Zhong et al., 2019), which contains 336K/11K training/validation images with 6 categories (*text*, *title*, *list*, *fig.*, and *table*). The original IIIT-AR-13K (Mondal et al., 2020) contains (*table*, *fig.*, *natural image*, *logo*, and *signature*). In this paper, considering the overlap between IIIT-AR-13K and PubLayNet, we select those images containing *natural images* as the OOD test set. After filtering, we obtain 2,880 OOD entities across 1,837 document images.

We consider three ID datasets in this experiment. *(1)* PubLayNet: This is the original PubLayNet dataset. We treat all the entities in training/validation images as ID entities. *(2)* Considering the domain shift between ID data (PubLayNet) and OOD data (IIIT-AR-13K). We combine the PubLayNet training data with the images from IIIT-AR-13K with overlapping annotations (*table* and *figure*) and train the object detection model.

### B.2 Models

Fig. 13 illustrates the entity recognition models used in this paper. We consider the entities on regions instead of tokens, as regions provide richer semantic information. As for the pre-trained model, we adopt UDoc (trained on IIT-CDIP) since it models inputs at the regional level. Based on the UDoc framework, we develop the following models.

**Vision/Vision+Layout:**

- **ResNet-50:** This model is composed of the ResNet-50 from pre-trained UDoc. It adopts the RoI pooling followed by a classifier to extract the entity features.

- **ResNet-50+Position:** This model also adapts UDoc's pre-trained ResNet-50 for further improvement. It makes the RoI features spatially aware by adding position embeddings, which are mapped from the bounding boxes via a linear mapping layer.

**Text/Text+Layout:**

- **Sentence BERT:** This model adopts the language branch of UDoc and appends the classifier to the output of the sentence encoder.

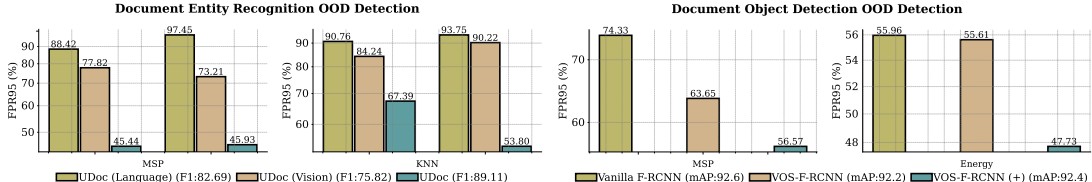

(a) Comparison of OOD detection methods on different models on two OOD classes: *other* and *header*.

(b) OOD detection results from different object detection methods and models.

Figure 14: Ablation on document entity recognition and object detection. Numbers are reported in FPR95.

- **Sentence BERT+Position:** This model is close to the above model but adds position embeddings to the sentence embeddings.

**Vision+Text+Layout:**

- **ResNet-50+sentence BERT:** This model follows the same framework as UDoc, but replaces the sentence encoder in their original design with a more miniature sentence encoder (all-MiniLM-L6-v2).

- **SwinT+Sentence BERT:** This model replaces the ResNet-50 visual backbone with a pre-trained tiny Swin Transformer (swin-tiny-patch4-window7-224) adopted from the Huggingface.

All the models are fine-tuned with the cross-entropy loss for 100 epochs, using a learning rate of $10^{-5}$ and a batch size of 8 on an A100 GPU.

### B.3 Summary of Observations

We provide a summary of observations here and hope to inspire future works on a thorough investigation of OOD detection for entity-level tasks. To identify entity types, models should not only understand the words but also utilize spatial and visual information.

For document entity recognition, the comparison of distance-based and logit-based OOD detection methods with different models are shown in Fig. 14a. More details are shown in Table 2. We see that models can better predict the entity type and also achieve better OOD robustness with the help of spatial information. Considering the weak language dependency between entities, it is not surprising that vision-based models achieve better performance than text-based models. In particular, UDoc with ResNet-50 achieves the best performance on two OOD test sets, illustrating that visual information plays a major role in increasing the discrimination of entities with similar semantics. For document object detection, we summarize our findings in Fig. 14b and describe them in more

detail in Table 1. We can see that the OOD detection performance is further improved by introducing document images from IIIT-AR-13K with the same ID annotations as training data.

To provide more intuitions, in Fig. 15, we visualize the document entity recognition OOD detection results. In Fig. 16, we visualize the prediction on sample OOD images, using object detection models trained without VOS (top) and with VOS (bottom), respectively. We can see that vanilla Faster RCNN trained on PubLayNet produces false positives when applied to the OOD document images from IIIT-AR-13K. Table 1 shows that introducing the unknown-aware learning method optimized for both ID and OOD can reduce the FPR95 while preserving the mAP on the ID data. This experiment indicates that incorporating uncertainty estimation into the entity detection training procedure can improve the reliability of the document object detection system.

## C Detailed Experimental Results

- Table 2 corresponds to the results shown in Fig. 15 and Fig. 14a.

- Table 1 corresponds to the results shown in Fig. 16 and Fig. 14b.

- Table 3 and Table 7 correspond to the results shown in Fig. 4a.

- Table 4 and Table 5 correspond to the results shown in Fig. 4c.

- Table 6 corresponds to the results shown in Fig. 8 and Fig. 9.

- Table 9 and Table 8 correspond to the results shown in Fig. 6 and Fig. 9.

- Table 10 and Table 11 correspond to the analysis for Sec. 4 and Sec. 4.2.

- Table 12 corresponds to the results shown in Fig. 9.

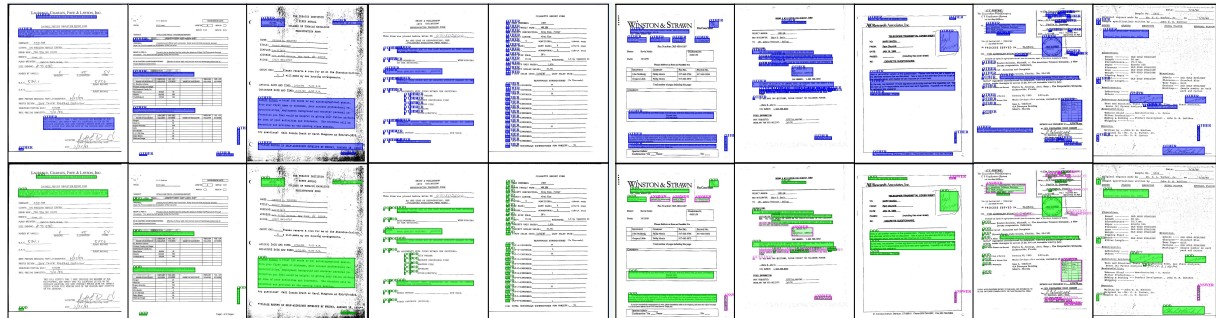

Figure 15: Visualization of detected OOD entities on the form images. The top part shows the entities in blue are entities annotated as *other*. The bottom part shows the detected OOD entities (green). We also show failure cases on the right part.

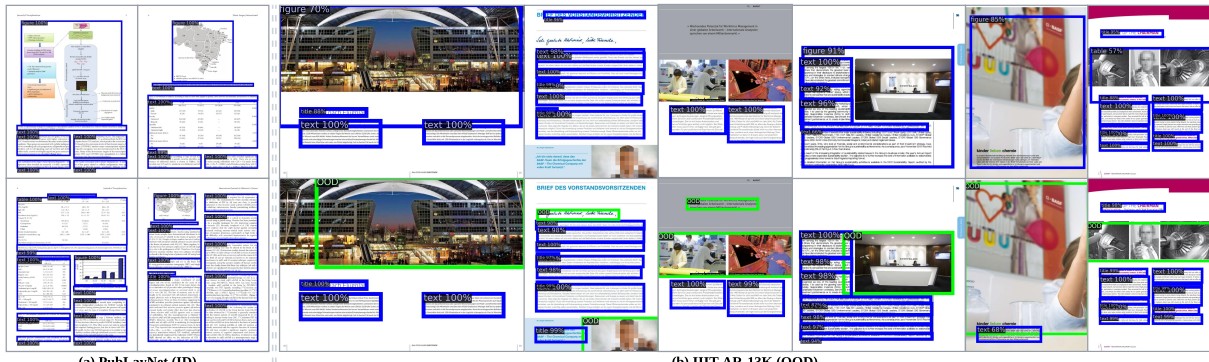

(a) PubLayNet (ID)                       (b) IIIT-AR-13K (OOD)

Figure 16: Visualization of detected objects on the OOD images (from IIIT-AR-13K) by a vanilla Faster-RCNN (top) and Faster-RCNN with VOS (bottom) is shown. Objects in blue boxes are detected and classified as one of the ID classes. The detected OOD objects (green) reduce false positives among detected objects. We also visualize detected objects on the ID images. There is a clear difference between PubLayNet and IIIT-AR-13K – entities and annotations of *natural images* rarely exist in PubLayNet.

Table 1: Comparison with different training and detection methods.

| Models | ID Dataset | OOD Score | IIIT-AR-13K (*Natural Image* as OOD) | | | PubLayNet (ID) |
|---|---|---|---|---|---|---|
| | | | FPR95 | AUROC | AUPR | mAP |
| Vanilla Faster-RCNN | PubLayNet | MSP | 74.33 | 79.12 | 98.41 | 92.6 |
| | | Energy | 55.96 | 83.55 | 98.73 | |
| Faster-RCNN with VOS | PubLayNet | MSP | 63.65 | 79.37 | 98.57 | 92.2 |
| | | Energy | 55.61 | 80.60 | 98.67 | |
| Faster-RCNN with VOS | PubLayNet+IIIT-AR-13K(ID) | MSP | 56.57 | 82.94 | 98.59 | 92.4 |
| | | Energy | 47.73 | 84.04 | 98.67 | |

Table 2: Comparison with different models on FUNSD OOD setting. All models are initialized with UDoc pre-trained on IIT-CDIP and fine-tuned on FUNSD data with ID entities. All values are percentages. S-BERT deontes Sentence BERT. A lower FPR95 or a higher AUROC value indicates better performance.

| | Test F1 | Method | Other (OOD) FPR95 | AUROC | ID F1 | Header (OOD) FPR95 | AUROC | ID F1 | | Test F1 | Method | Other (OOD) FPR95 | AUROC | ID F1 | Header (OOD) FPR95 | AUROC | ID F1 |
|---|---|---|---|---|---|---|---|---|---|---|---|---|---|---|---|---|---|
| ResNet-50 | 75.15 | $KNN_{10}$ | 59.47 | 79.14 | | 81.79 | 63.97 | | ResNet-50+Position | 75.82 | $KNN_{10}$ | 73.21 | 73.19 | | 90.22 | 61.42 | |
| | | $KNN_{20}$ | 69.97 | 78.15 | | 81.25 | 63.66 | | | | $KNN_{20}$ | 72.91 | 73.44 | | 88.04 | 61.54 | |
| | | $KNN_{50}$ | 84.49 | 77.40 | | 82.61 | 62.86 | | | | $KNN_{50}$ | 75.96 | 74.43 | | 82.88 | 60.93 | |
| | | $KNN_{100}$ | 97.94 | 77.08 | 77.65 | 84.24 | 61.62 | 78.04 | | | $KNN_{100}$ | 79.69 | 74.85 | 77.65 | 83.70 | 59.39 | 77.98 |
| | | $KNN_{200}$ | 97.84 | 77.15 | | 94.29 | 59.74 | | | | $KNN_{200}$ | 86.06 | 75.14 | | 91.58 | 57.42 | |
| | | $KNN_{400}$ | 97.15 | 76.09 | | 94.84 | 57.53 | | | | $KNN_{400}$ | 87.93 | 74.92 | | 95.92 | 55.37 | |
| | | MSP | 50.54 | 75.80 | | 75.82 | 76.55 | | | | MSP | 77.82 | 67.60 | | 84.24 | 66.58 | |
| | | MaxLogit | 52.40 | 73.70 | | 73.64 | 76.72 | | | | MaxLogit | 76.94 | 67.05 | | 84.24 | 65.41 | |
| | | Energy | 52.50 | 73.70 | | 75.82 | 76.55 | | | | Energy | 76.64 | 66.93 | | 84.51 | 64.98 | |
| S-BERT | 77.15 | $KNN_{10}$ | 93.72 | 48.44 | | 92.66 | 60.99 | | S-BERT+Position | 82.69 | $KNN_{10}$ | 97.45 | 41.24 | | 93.75 | 62.38 | |
| | | $KNN_{20}$ | 93.92 | 47.65 | | 92.93 | 59.00 | | | | $KNN_{20}$ | 97.55 | 39.91 | | 93.48 | 61.51 | |
| | | $KNN_{50}$ | 93.62 | 48.94 | | 93.21 | 57.90 | | | | $KNN_{50}$ | 97.15 | 39.56 | | 92.39 | 61.76 | |
| | | $KNN_{100}$ | 93.92 | 48.79 | 82.12 | 93.21 | 55.07 | 82.41 | | | $KNN_{100}$ | 97.06 | 41.67 | 87.08 | 91.85 | 60.99 | 87.01 |
| | | $KNN_{200}$ | 93.92 | 47.85 | | 93.48 | 52.86 | | | | $KNN_{200}$ | 96.57 | 41.85 | | 89.67 | 59.08 | |
| | | $KNN_{400}$ | 94.11 | 46.21 | | 95.38 | 49.86 | | | | $KNN_{400}$ | 97.25 | 40.83 | | 90.22 | 54.03 | |
| | | MSP | 93.62 | 54.91 | | 94.29 | 52.14 | | | | MSP | 88.42 | 61.11 | | 90.76 | 59.58 | |
| | | MaxLogit | 93.72 | 54.75 | | 94.57 | 56.51 | | | | MaxLogit | 89.70 | 60.19 | | 88.86 | 60.92 | |
| | | Energy | 93.23 | 54.88 | | 93.21 | 58.22 | | | | Energy | 90.48 | 59.61 | | 89.95 | 61.12 | |
| ResNet-50+S-BERT | 89.11 | $KNN_{10}$ | 45.93 | 87.85 | | 53.80 | 87.97 | | SwinT+S-BERT | 86.00 | $KNN_{10}$ | 63.30 | 83.64 | | 81.52 | 64.08 | |
| | | $KNN_{20}$ | 53.58 | 86.71 | | 55.71 | 87.06 | | | | $KNN_{20}$ | 66.73 | 82.53 | | 81.52 | 61.50 | |
| | | $KNN_{50}$ | 73.21 | 84.36 | | 62.77 | 85.49 | | | | $KNN_{50}$ | 70.17 | 80.21 | | 82.34 | 57.77 | |
| | | $KNN_{100}$ | 89.70 | 83.01 | 93.13 | 69.02 | 83.60 | 93.18 | | | $KNN_{100}$ | 83.91 | 77.71 | 90.82 | 83.15 | 54.97 | 90.40 |
| | | $KNN_{200}$ | 96.66 | 81.90 | | 75.54 | 80.85 | | | | $KNN_{200}$ | 95.39 | 75.79 | | 95.38 | 50.57 | |
| | | $KNN_{400}$ | 98.82 | 81.00 | | 91.58 | 77.42 | | | | $KNN_{400}$ | 96.76 | 75.49 | | 99.73 | 47.45 | |
| | | MSP | 45.44 | 87.82 | | 67.39 | 72.85 | | | | MSP | 69.28 | 70.70 | | 80.71 | 52.02 | |
| | | MaxLogit | 45.53 | 90.58 | | 63.04 | 72.39 | | | | MaxLogit | 67.12 | 74.41 | | 81.79 | 52.77 | |
| | | Energy | 45.53 | 90.57 | | 63.86 | 72.37 | | | | Energy | 67.22 | 74.41 | | 81.79 | 52.77 | |

Table 3: OOD detection performance for document classification with different number of pre-training data from IIT-CDIP. ID (Acc) denotes the ID accuracy obtained by testing on ID test data. We report the KNN-based scores for both pre-trained and fine-tuned models. *Sci. Poster* denotes the document images converted from NJU-Fudan Paper-Poster Dataset. *Receipt* denotes the receipt images collected from the CORD receipt understanding dataset. For in-domain OOD test data, we also report the averaged scores.

| | ID Acc | Method | OOD Dataset (In-domain) | | | | | | | | | | OOD Dataset (Out-domain) | | | |
| | | | Sci. Report | | Presentation | | Form | | Letter | | *Average* | | Sci. Poster | | Receipt | |
| | | | FPR95 | AUROC | FPR95 | AUROC | FPR95 | AUROC | FPR95 | AUROC | FPR95 | AUROC | FPR95 | AUROC | FPR95 | AUROC |
| **RoBERTa$_{Base}$ (10%)** | | | **Pre-train on 10% IIT-CDIP→ fine-tune on RVL-CDIP ID data** | | | | | | | | | | | | | |
| | 90.59 | MSP | 92.75 | 69.24 | 92.21 | 66.93 | 94.65 | 65.40 | 92.00 | 70.09 | 92.90 | 67.92 | 96.51 | 66.93 | 99.10 | 52.90 |
| | | MaxLogit | 98.36 | 77.85 | 97.23 | 78.51 | 98.76 | 72.84 | 98.86 | 78.08 | 98.30 | 76.82 | 100.00 | 78.69 | 100.00 | 63.74 |
| | | Energy | 98.60 | 77.81 | 97.55 | 78.49 | 98.96 | 72.79 | 98.94 | 78.00 | 98.51 | 76.77 | 100.00 | 78.68 | 100.00 | 63.70 |
| | | GradNorm | 98.04 | 79.26 | 97.07 | 76.85 | 98.56 | 72.83 | 98.62 | 80.55 | 98.07 | 77.37 | 100.00 | 85.23 | 100.00 | 64.10 |
| | | KNN$_{10}$ | 63.21 | 88.18 | 65.81 | 88.05 | 73.02 | 84.63 | 67.74 | 88.92 | 67.45 | 87.44 | 69.77 | 88.49 | 90.50 | 84.44 |
| | | KNN$_{20}$ | 63.53 | 88.07 | 65.89 | 87.90 | 72.75 | 84.48 | 67.33 | 88.81 | 67.38 | 87.32 | 68.60 | 88.13 | 91.10 | 84.09 |
| | | KNN$_{50}$ | 64.17 | 87.89 | 66.97 | 87.77 | 73.34 | 84.23 | 67.21 | 88.60 | 67.92 | 87.12 | 72.09 | 87.47 | 91.60 | 83.59 |
| | | KNN$_{100}$ | 64.49 | 87.64 | 67.78 | 87.55 | 73.46 | 83.94 | 67.29 | 88.37 | 68.26 | 86.88 | 72.09 | 86.83 | 91.50 | 83.21 |
| | | **Pre-train on 10% IIT-CDIP (no fine-tune)** | | | | | | | | | | | | | | |
| | — | KNN$_{10}$ | 88.07 | 66.94 | 92.13 | 66.62 | 94.13 | 61.90 | 94.40 | 54.57 | 92.18 | 62.51 | 67.44 | 87.04 | 62.10 | 84.94 |
| | | KNN$_{20}$ | 88.59 | 66.02 | 92.65 | 65.25 | 94.13 | 60.83 | 94.72 | 53.79 | 92.52 | 61.47 | 77.91 | 85.38 | 64.60 | 83.86 |
| | | KNN$_{50}$ | 89.75 | 64.40 | 93.53 | 63.12 | 94.37 | 58.98 | 95.17 | 52.33 | 93.20 | 59.71 | 82.97 | 82.29 | 69.20 | 82.29 |
| | | KNN$_{100}$ | 90.23 | 62.94 | 93.85 | 61.28 | 94.41 | 57.45 | 95.13 | 51.28 | 93.40 | 58.24 | 83.72 | 80.91 | 70.10 | 81.05 |
| **RoBERTa$_{Base}$ (20%)** | | | **Pre-train on 20% IIT-CDIP→ fine-tune on RVL-CDIP ID data** | | | | | | | | | | | | | |
| | 90.71 | MSP | 94.28 | 68.02 | 94.46 | 65.98 | 96.01 | 62.98 | 94.81 | 65.98 | 94.89 | 65.74 | 95.35 | 63.55 | 99.10 | 54.99 |
| | | MaxLogit | 97.36 | 77.82 | 97.19 | 79.16 | 98.40 | 72.64 | 98.34 | 77.68 | 97.82 | 76.82 | 100.00 | 77.36 | 99.60 | 66.63 |
| | | Energy | 98.04 | 77.80 | 97.43 | 79.15 | 98.76 | 72.61 | 98.58 | 77.64 | 98.20 | 76.80 | 100.00 | 77.32 | 99.60 | 66.61 |
| | | GradNorm | 97.36 | 80.68 | 96.83 | 76.04 | 98.44 | 73.29 | 97.89 | 81.37 | 97.63 | 77.85 | 100.00 | 86.18 | 99.50 | 67.49 |
| | | KNN$_{10}$ | 63.57 | 88.30 | 67.06 | 87.06 | 73.66 | 83.92 | 73.09 | 87.80 | 69.34 | 86.77 | 69.77 | 88.01 | 87.60 | 83.81 |
| | | KNN$_{20}$ | 63.85 | 88.20 | 67.46 | 86.90 | 73.94 | 83.78 | 72.93 | 87.70 | 69.54 | 86.64 | 69.77 | 87.63 | 88.30 | 83.53 |
| | | KNN$_{50}$ | 63.89 | 88.02 | 67.54 | 86.71 | 74.38 | 83.55 | 72.24 | 87.46 | 69.51 | 86.43 | 70.93 | 87.09 | 88.20 | 83.12 |
| | | KNN$_{100}$ | 64.85 | 87.81 | 67.62 | 86.45 | 74.90 | 83.25 | 72.65 | 87.24 | 70.00 | 86.19 | 72.09 | 86.65 | 88.30 | 82.89 |
| | | **Pre-train on 20% IIT-CDIP (no fine-tune)** | | | | | | | | | | | | | | |
| | — | KNN$_{10}$ | 87.15 | 68.27 | 90.88 | 66.89 | 92.26 | 62.39 | 95.01 | 53.02 | 91.32 | 62.64 | 43.02 | 92.29 | 57.00 | 87.67 |
| | | KNN$_{20}$ | 87.31 | 67.35 | 92.04 | 65.54 | 91.54 | 61.40 | 94.97 | 52.33 | 91.46 | 61.66 | 47.67 | 91.18 | 62.60 | 86.61 |
| | | KNN$_{50}$ | 88.39 | 65.71 | 92.69 | 63.45 | 92.18 | 59.57 | 95.25 | 50.97 | 92.13 | 59.92 | 56.98 | 89.64 | 65.70 | 85.20 |
| | | KNN$_{100}$ | 88.83 | 64.20 | 93.13 | 61.61 | 92.22 | 57.99 | 95.45 | 49.95 | 92.41 | 58.44 | 58.14 | 88.36 | 66.90 | 84.17 |
| **RoBERTa$_{Base}$ (40%)** | | | **Pre-train on 40% IIT-CDIP→ fine-tune on RVL-CDIP ID data** | | | | | | | | | | | | | |
| | 90.76 | MSP | 92.67 | 70.09 | 93.93 | 65.69 | 95.05 | 63.19 | 95.50 | 65.54 | 94.29 | 66.13 | 95.35 | 63.63 | 95.40 | 64.97 |
| | | MaxLogit | 98.08 | 78.72 | 97.87 | 79.85 | 98.44 | 71.63 | 98.30 | 75.41 | 98.17 | 76.40 | 98.84 | 78.07 | 98.90 | 75.65 |
| | | Energy | 98.48 | 78.69 | 97.91 | 79.83 | 98.50 | 71.61 | 98.50 | 75.40 | 98.39 | 76.38 | 100.00 | 78.04 | 98.50 | 75.60 |
| | | GradNorm | 98.04 | 81.03 | 97.47 | 76.73 | 98.44 | 72.77 | 97.40 | 79.11 | 97.84 | 77.41 | 100.00 | 87.47 | 97.60 | 77.12 |
| | | KNN$_{10}$ | 60.57 | 88.79 | 68.86 | 86.36 | 75.26 | 83.55 | 73.90 | 87.12 | 69.65 | 86.46 | 67.44 | 89.90 | 72.70 | 89.49 |
| | | KNN$_{20}$ | 61.37 | 88.72 | 69.06 | 86.24 | 75.46 | 83.43 | 73.46 | 87.00 | 69.84 | 86.35 | 68.60 | 89.66 | 73.50 | 89.25 |
| | | KNN$_{50}$ | 62.21 | 88.52 | 69.18 | 86.08 | 75.66 | 83.21 | 73.42 | 86.71 | 70.12 | 86.13 | 70.93 | 89.20 | 74.70 | 88.89 |
| | | KNN$_{100}$ | 63.77 | 88.30 | 69.79 | 85.84 | 76.02 | 82.93 | 74.19 | 86.46 | 70.94 | 85.88 | 74.42 | 88.84 | 75.30 | 88.69 |
| | | **Pre-train on 40% IIT-CDIP (no fine-tune)** | | | | | | | | | | | | | | |
| | — | KNN$_{10}$ | 85.71 | 69.08 | 90.84 | 68.68 | 90.46 | 62.52 | 94.76 | 51.76 | 90.44 | 63.01 | 25.58 | 95.83 | 57.30 | 88.60 |
| | | KNN$_{20}$ | 85.27 | 68.21 | 91.64 | 67.48 | 89.74 | 61.32 | 94.81 | 51.01 | 90.36 | 62.00 | 29.07 | 95.22 | 62.30 | 87.61 |
| | | KNN$_{50}$ | 86.19 | 66.60 | 92.21 | 65.54 | 90.30 | 59.35 | 94.93 | 49.60 | 90.91 | 60.27 | 41.86 | 94.32 | 66.80 | 86.25 |
| | | KNN$_{100}$ | 87.19 | 65.04 | 92.57 | 63.83 | 90.50 | 57.74 | 95.09 | 48.44 | 91.34 | 58.76 | 45.35 | 93.66 | 68.30 | 85.14 |
| **RoBERTa$_{Base}$ (100%)** | | | **Pre-train on 100% IIT-CDIP→ fine-tune on RVL-CDIP ID data** | | | | | | | | | | | | | |
| | 91.00 | MSP | 93.23 | 68.88 | 94.54 | 65.83 | 96.65 | 63.11 | 94.12 | 68.28 | 94.64 | 66.53 | 98.84 | 62.52 | 95.10 | 71.25 |
| | | MaxLogit | 97.84 | 78.86 | 97.95 | 80.23 | 98.48 | 74.01 | 98.25 | 77.59 | 98.13 | 77.67 | 100.00 | 78.73 | 98.90 | 79.36 |
| | | Energy | 98.20 | 78.84 | 97.95 | 80.22 | 98.52 | 74.00 | 98.78 | 77.55 | 98.36 | 77.65 | 100.00 | 78.72 | 98.70 | 79.29 |
| | | GradNorm | 97.88 | 80.81 | 97.91 | 76.37 | 98.28 | 75.25 | 98.25 | 80.09 | 98.08 | 78.13 | 100.00 | 86.10 | 98.30 | 77.50 |
| | | KNN$_{10}$ | 62.57 | 88.26 | 68.90 | 86.96 | 72.39 | 84.73 | 70.37 | 88.23 | 68.56 | 87.04 | 72.09 | 89.97 | 65.90 | 90.51 |
| | | KNN$_{20}$ | 63.41 | 88.11 | 69.59 | 86.88 | 73.10 | 84.56 | 70.70 | 88.11 | 69.20 | 86.92 | 74.42 | 89.58 | 67.20 | 90.37 |
| | | KNN$_{50}$ | 63.85 | 87.87 | 69.79 | 86.79 | 73.90 | 84.30 | 71.14 | 87.87 | 69.67 | 86.71 | 76.74 | 88.95 | 67.90 | 90.22 |
| | | KNN$_{100}$ | 65.13 | 87.61 | 70.27 | 86.58 | 74.86 | 84.00 | 71.75 | 87.65 | 70.50 | 86.46 | 79.07 | 88.44 | 68.30 | 90.19 |
| | | **Pre-train on 100% IIT-CDIP (no fine-tune)** | | | | | | | | | | | | | | |
| | — | KNN$_{10}$ | 84.43 | 70.20 | 90.20 | 68.54 | 90.98 | 63.18 | 94.72 | 52.16 | 90.08 | 63.52 | 27.91 | 94.10 | 46.00 | 91.37 |
| | | KNN$_{20}$ | 84.51 | 69.30 | 91.28 | 67.35 | 90.38 | 61.96 | 94.72 | 51.43 | 90.22 | 62.51 | 33.72 | 93.39 | 51.50 | 90.55 |
| | | KNN$_{50}$ | 85.67 | 67.75 | 91.92 | 65.35 | 90.82 | 59.79 | 94.89 | 49.77 | 90.82 | 60.66 | 39.53 | 92.28 | 56.70 | 89.32 |
| | | KNN$_{100}$ | 86.55 | 66.08 | 92.97 | 63.46 | 91.46 | 58.00 | 95.41 | 48.39 | 91.60 | 58.98 | 44.19 | 91.29 | 61.60 | 88.18 |

Table 4: OOD detection performance for document classification with different number of pre-training data from IIT-CDIP⁻ (remove *pseudo* OOD categories).

| | ID Acc | Method | OOD Dataset (In-domain) | | | | | | | | | | OOD Dataset (Out-domain) | | | |
|---|---|---|---|---|---|---|---|---|---|---|---|---|---|---|---|---|
| | | | Sci. Report | | Presentation | | Form | | Letter | | *Average* | | Sci. Poster | | Receipt | |
| | | | FPR95 | AUROC | FPR95 | AUROC | FPR95 | AUROC | FPR95 | AUROC | FPR95 | AUROC | FPR95 | AUROC | FPR95 | AUROC |
| **RoBERTa_Base (10%)** | | | Pre-train on 10% IIT-CDIP⁻ → fine-tune on RVL-CDIP ID data | | | | | | | | | | | | | |
| | 90.62 | MSP | 90.07 | 69.00 | 89.92 | 68.86 | 92.58 | 64.16 | 91.07 | 66.78 | 90.91 | 67.20 | 96.51 | 54.47 | 96.70 | 59.63 |
| | | MaxLogit | 97.76 | 78.40 | 97.71 | 80.58 | 98.64 | 71.26 | 98.70 | 76.38 | 98.20 | 76.66 | 100.00 | 73.51 | 99.80 | 73.32 |
| | | Energy | 98.16 | 78.35 | 97.75 | 80.55 | 98.84 | 71.20 | 98.90 | 76.32 | 98.41 | 76.60 | 100.00 | 73.46 | 99.80 | 73.31 |
| | | GradNorm | 97.68 | 79.92 | 97.27 | 79.42 | 98.56 | 71.31 | 98.50 | 79.44 | 98.00 | 77.52 | 100.00 | 82.62 | 99.60 | 75.85 |
| | | KNN₁₀ | 65.85 | 87.89 | 66.69 | 88.12 | 75.98 | 82.82 | 74.55 | 86.85 | 70.77 | 86.42 | 87.21 | 85.16 | 83.90 | 87.91 |
| | | KNN₂₀ | 66.33 | 87.80 | 66.85 | 88.04 | 75.94 | 82.70 | 73.94 | 86.75 | 70.76 | 86.32 | 87.21 | 84.63 | 83.60 | 87.71 |
| | | KNN₅₀ | 66.77 | 87.66 | 67.30 | 88.00 | 76.02 | 82.49 | 73.66 | 86.52 | 70.94 | 86.17 | 88.37 | 83.73 | 83.90 | 87.34 |
| | | KNN₁₀₀ | 67.25 | 87.42 | 67.74 | 87.84 | 76.18 | 82.18 | 73.99 | 86.26 | 71.29 | 85.92 | 89.53 | 82.85 | 83.90 | 86.98 |
| | — | Pre-train on 10% IIT-CDIP⁻ (no fine-tune) | | | | | | | | | | | | | | |
| | | KNN₁₀ | 86.35 | 65.48 | 85.74 | 70.84 | 92.94 | 59.55 | 93.14 | 56.62 | 89.54 | 63.12 | 29.07 | 95.42 | 87.60 | 83.13 |
| | | KNN₂₀ | 86.87 | 64.48 | 87.14 | 69.68 | 93.30 | 58.41 | 93.30 | 55.91 | 90.15 | 62.12 | 37.21 | 94.75 | 88.00 | 81.44 |
| | | KNN₅₀ | 87.75 | 62.73 | 88.99 | 67.80 | 93.50 | 56.54 | 93.75 | 54.52 | 91.00 | 60.40 | 47.67 | 93.71 | 90.30 | 78.97 |
| | | KNN₁₀₀ | 88.43 | 61.17 | 89.59 | 66.05 | 93.62 | 54.91 | 93.99 | 53.40 | 91.41 | 58.88 | 48.84 | 93.09 | 91.50 | 77.00 |
| **RoBERTa_Base (20%)** | | | Pre-train on 20% IIT-CDIP⁻ → fine-tune on RVL-CDIP ID data | | | | | | | | | | | | | |
| | 90.65 | MSP | 96.04 | 67.58 | 94.90 | 68.32 | 96.05 | 64.92 | 96.23 | 68.62 | 95.80 | 67.36 | 100.00 | 61.49 | 99.50 | 56.38 |
| | | MaxLogit | 97.96 | 76.92 | 97.59 | 80.68 | 98.48 | 72.31 | 98.74 | 77.72 | 98.19 | 76.91 | 100.00 | 75.91 | 99.50 | 69.21 |
| | | Energy | 98.16 | 76.89 | 98.23 | 80.65 | 98.88 | 72.26 | 99.07 | 77.67 | 98.58 | 76.87 | 100.00 | 75.89 | 99.50 | 69.18 |
| | | GradNorm | 97.84 | 78.23 | 97.31 | 78.57 | 98.00 | 71.44 | 98.46 | 80.34 | 97.90 | 77.16 | 100.00 | 85.80 | 99.00 | 69.54 |
| | | KNN₁₀ | 66.05 | 87.60 | 67.70 | 87.94 | 73.42 | 83.10 | 73.50 | 87.96 | 70.17 | 86.65 | 77.91 | 90.19 | 90.10 | 84.32 |
| | | KNN₂₀ | 66.17 | 87.50 | 68.38 | 87.83 | 73.90 | 82.93 | 73.66 | 87.82 | 70.53 | 86.52 | 77.91 | 89.84 | 89.80 | 84.13 |
| | | KNN₅₀ | 67.21 | 87.26 | 68.46 | 87.73 | 74.18 | 82.63 | 73.66 | 87.58 | 70.88 | 86.30 | 79.07 | 89.24 | 89.80 | 83.80 |
| | | KNN₁₀₀ | 68.78 | 86.98 | 69.14 | 87.53 | 75.50 | 82.30 | 74.27 | 87.36 | 71.92 | 86.04 | 82.56 | 88.68 | 89.80 | 83.59 |
| | — | Pre-train on 20% IIT-CDIP⁻ (no fine-tune) | | | | | | | | | | | | | | |
| | | KNN₁₀ | 85.63 | 66.10 | 85.17 | 70.34 | 92.58 | 60.29 | 93.43 | 56.85 | 89.20 | 63.40 | 30.23 | 95.72 | 83.20 | 83.84 |
| | | KNN₂₀ | 86.31 | 65.17 | 85.98 | 69.13 | 93.30 | 59.09 | 93.47 | 56.05 | 89.77 | 62.36 | 34.88 | 95.08 | 84.90 | 82.16 |
| | | KNN₅₀ | 87.31 | 63.50 | 87.63 | 67.11 | 93.38 | 57.17 | 94.16 | 54.60 | 90.62 | 60.60 | 44.19 | 94.07 | 87.50 | 79.74 |
| | | KNN₁₀₀ | 87.83 | 62.06 | 88.27 | 65.31 | 93.62 | 55.65 | 94.32 | 53.56 | 91.01 | 59.14 | 48.84 | 93.48 | 88.80 | 77.77 |
| **RoBERTa_Base (40%)** | | | Pre-train on 40% IIT-CDIP⁻ → fine-tune on RVL-CDIP ID data | | | | | | | | | | | | | |
| | 90.72 | MSP | 93.84 | 68.86 | 93.69 | 67.62 | 95.41 | 63.91 | 94.20 | 65.25 | 94.28 | 66.41 | 96.51 | 63.32 | 98.90 | 54.02 |
| | | MaxLogit | 97.16 | 78.56 | 96.87 | 80.18 | 98.68 | 71.84 | 98.58 | 74.44 | 97.82 | 76.26 | 100.00 | 76.72 | 99.10 | 65.41 |
| | | Energy | 97.40 | 78.53 | 97.15 | 80.17 | 98.68 | 71.79 | 98.78 | 74.39 | 98.00 | 76.22 | 100.00 | 76.67 | 99.50 | 65.39 |
| | | GradNorm | 97.24 | 80.59 | 96.95 | 78.01 | 98.52 | 72.12 | 98.34 | 77.16 | 97.76 | 76.97 | 100.00 | 86.94 | 99.70 | 67.46 |
| | | KNN₁₀ | 66.89 | 87.91 | 68.58 | 86.90 | 77.61 | 82.31 | 76.58 | 85.39 | 72.41 | 85.63 | 75.58 | 89.45 | 86.40 | 84.23 |
| | | KNN₂₀ | 67.57 | 87.80 | 68.90 | 86.79 | 77.77 | 82.19 | 76.30 | 85.22 | 72.64 | 85.50 | 80.23 | 89.17 | 86.80 | 83.85 |
| | | KNN₅₀ | 67.97 | 87.58 | 69.67 | 86.67 | 78.01 | 81.98 | 76.66 | 84.85 | 73.08 | 85.27 | 80.23 | 88.63 | 87.20 | 83.21 |
| | | KNN₁₀₀ | 69.46 | 87.34 | 71.23 | 86.47 | 79.01 | 81.72 | 77.48 | 84.57 | 74.30 | 85.02 | 82.56 | 88.19 | 88.00 | 82.72 |
| | — | Pre-train on 40% IIT-CDIP⁻ (no fine-tune) | | | | | | | | | | | | | | |
| | | KNN₁₀ | 88.79 | 66.14 | 88.35 | 68.92 | 93.50 | 60.30 | 95.54 | 51.09 | 91.54 | 61.61 | 37.21 | 95.37 | 55.90 | 91.90 |
| | | KNN₂₀ | 89.59 | 65.07 | 89.80 | 67.61 | 93.89 | 59.10 | 95.58 | 50.17 | 92.21 | 60.49 | 46.51 | 94.41 | 61.50 | 91.00 |
| | | KNN₅₀ | 90.59 | 63.39 | 91.64 | 65.68 | 93.77 | 57.35 | 95.66 | 48.76 | 92.92 | 58.76 | 53.49 | 93.06 | 64.40 | 89.72 |
| | | KNN₁₀₀ | 91.19 | 61.79 | 92.37 | 63.90 | 93.66 | 55.78 | 95.62 | 47.42 | 93.21 | 57.22 | 65.12 | 91.99 | 68.30 | 88.72 |
| **RoBERTa_Base (100%)** | | | Pre-train on 100% IIT-CDIP⁻ → fine-tune on RVL-CDIP ID data | | | | | | | | | | | | | |
| | 90.74 | MSP | 94.12 | 68.24 | 94.29 | 66.18 | 95.93 | 63.83 | 95.21 | 65.66 | 94.89 | 65.98 | 98.84 | 59.25 | 96.50 | 65.42 |
| | | MaxLogit | 97.24 | 78.15 | 97.19 | 80.27 | 98.36 | 72.16 | 98.38 | 75.82 | 97.79 | 76.60 | 100.00 | 73.28 | 99.30 | 75.58 |
| | | Energy | 97.32 | 78.13 | 97.51 | 80.26 | 98.64 | 72.12 | 98.70 | 75.78 | 98.04 | 76.57 | 100.00 | 73.27 | 99.60 | 75.52 |
| | | GradNorm | 97.16 | 80.07 | 97.39 | 77.86 | 98.40 | 71.83 | 98.05 | 79.08 | 97.75 | 77.21 | 100.00 | 86.32 | 99.40 | 73.52 |
| | | KNN₁₀ | 66.81 | 87.86 | 69.67 | 86.91 | 77.49 | 82.60 | 74.59 | 86.28 | 72.14 | 85.91 | 81.40 | 87.74 | 76.90 | 88.49 |
| | | KNN₂₀ | 66.73 | 87.75 | 70.31 | 86.78 | 77.89 | 82.51 | 75.28 | 86.13 | 72.55 | 85.79 | 81.40 | 87.43 | 77.50 | 88.39 |
| | | KNN₅₀ | 67.25 | 87.54 | 70.59 | 86.62 | 77.85 | 82.32 | 75.41 | 85.84 | 72.78 | 85.58 | 83.72 | 86.85 | 77.80 | 88.23 |
| | | KNN₁₀₀ | 68.13 | 87.34 | 71.47 | 86.39 | 78.05 | 82.08 | 76.14 | 85.60 | 73.45 | 85.35 | 83.72 | 86.39 | 78.50 | 88.21 |
| | — | Pre-train on 100% IIT-CDIP⁻ (no fine-tune) | | | | | | | | | | | | | | |
| | | KNN₁₀ | 87.95 | 66.44 | 84.49 | 72.34 | 95.01 | 58.47 | 96.23 | 49.07 | 90.92 | 61.58 | 31.40 | 96.19 | 41.60 | 94.78 |
| | | KNN₂₀ | 88.91 | 65.39 | 85.70 | 71.25 | 95.33 | 57.19 | 96.59 | 48.06 | 91.63 | 60.47 | 34.88 | 95.50 | 48.40 | 94.12 |
| | | KNN₅₀ | 90.59 | 63.69 | 87.14 | 69.45 | 95.53 | 54.93 | 97.08 | 46.26 | 92.58 | 58.58 | 43.02 | 94.51 | 55.20 | 93.05 |
| | | KNN₁₀₀ | 91.75 | 62.08 | 88.55 | 67.85 | 95.89 | 53.05 | 97.20 | 44.81 | 93.35 | 56.95 | 50.00 | 93.60 | 61.10 | 92.04 |

Table 5: OOD detection performance for document classification with different number of pre-training data from IIT-CDIP⁻ (remove *pseudo* OOD categories).

| | | | OOD Dataset (In-domain) | | | | | | | | | | OOD Dataset (Out-domain) | | | |
|---|---|---|---|---|---|---|---|---|---|---|---|---|---|---|---|---|
| | ID Acc | Method | Sci. Report | | Presentation | | Form | | Letter | | *Average* | | Sci. Poster | | Receipt | |
| | | | FPR95 | AUROC | FPR95 | AUROC | FPR95 | AUROC | FPR95 | AUROC | FPR95 | AUROC | FPR95 | AUROC | FPR95 | AUROC |
| **LayoutLMv1$_{Base}$ (10%)** | | **Pre-train on 10% IIT-CDIP⁻ → fine-tune on RVL-CDIP ID data** | | | | | | | | | | | | | | |
| | 95.89 | MSP | 42.43 | 76.31 | 56.05 | 69.39 | 54.31 | 70.25 | 47.00 | 73.93 | 49.95 | 72.47 | 43.02 | 76.55 | 44.10 | 75.68 |
| | | MaxLogit | 41.91 | 91.27 | 55.04 | 89.33 | 54.19 | 85.20 | 44.97 | 90.93 | 49.03 | 89.18 | 38.37 | 94.27 | 41.30 | 91.38 |
| | | Energy | 41.83 | 91.29 | 54.92 | 89.35 | 54.11 | 85.22 | 45.01 | 90.97 | 48.97 | 89.21 | 38.37 | 94.29 | 41.10 | 91.42 |
| | | GradNorm | 39.15 | 91.80 | 54.04 | 86.93 | 51.88 | 86.05 | 42.49 | 91.65 | 46.89 | 89.11 | 38.37 | 91.79 | 41.40 | 91.82 |
| | | KNN$_{10}$ | 31.63 | 94.25 | 46.52 | 90.98 | 46.77 | 90.49 | 40.83 | 92.79 | 41.44 | 92.13 | 24.42 | 95.95 | 30.30 | 95.66 |
| | | KNN$_{20}$ | 32.03 | 94.11 | 46.65 | 90.89 | 47.01 | 90.32 | 41.60 | 92.63 | 41.82 | 91.99 | 26.74 | 95.76 | 31.80 | 95.44 |
| | | KNN$_{50}$ | 34.39 | 93.75 | 49.34 | 90.46 | 49.36 | 89.94 | 44.52 | 92.23 | 44.40 | 91.60 | 33.72 | 95.33 | 33.20 | 95.38 |
| | | KNN$_{100}$ | 36.15 | 93.47 | 51.27 | 90.19 | 51.36 | 89.65 | 46.63 | 91.99 | 46.35 | 91.32 | 33.72 | 95.10 | 35.10 | 95.16 |
| | | **Pre-train on 10% IIT-CDIP⁻ (no fine-tune)** | | | | | | | | | | | | | | |
| | — | KNN$_{10}$ | 90.95 | 72.30 | 94.66 | 65.49 | 90.94 | 72.38 | 94.40 | 67.32 | 92.74 | 69.37 | 48.84 | 91.56 | 56.00 | 75.08 |
| | | KNN$_{20}$ | 91.59 | 70.54 | 94.98 | 63.91 | 91.66 | 70.74 | 94.81 | 65.95 | 93.26 | 67.78 | 53.49 | 90.41 | 57.60 | 73.51 |
| | | KNN$_{50}$ | 93.07 | 67.76 | 95.54 | 61.24 | 92.78 | 68.27 | 95.25 | 64.01 | 94.16 | 65.32 | 55.81 | 88.37 | 58.50 | 71.06 |
| | | KNN$_{100}$ | 93.55 | 65.41 | 95.90 | 59.13 | 93.10 | 66.19 | 95.54 | 62.41 | 94.52 | 63.28 | 67.44 | 86.44 | 60.20 | 69.09 |
| **LayoutLMv1$_{Base}$ (20%)** | | **Pre-train on 20% IIT-CDIP⁻ → fine-tune on RVL-CDIP ID data** | | | | | | | | | | | | | | |
| | 95.84 | MSP | 49.20 | 76.78 | 61.51 | 70.13 | 62.37 | 69.49 | 55.52 | 73.64 | 57.15 | 72.51 | 50.00 | 77.99 | 50.70 | 75.90 |
| | | MaxLogit | 41.03 | 91.57 | 54.00 | 88.45 | 56.42 | 85.70 | 47.00 | 90.19 | 49.61 | 88.98 | 38.37 | 93.62 | 41.80 | 90.56 |
| | | Energy | 40.95 | 91.60 | 53.76 | 88.47 | 56.19 | 85.72 | 46.79 | 90.22 | 49.42 | 89.00 | 38.37 | 93.65 | 41.70 | 90.59 |
| | | GradNorm | 37.15 | 91.89 | 54.16 | 84.99 | 53.03 | 86.28 | 43.95 | 90.94 | 47.07 | 88.52 | 40.70 | 90.41 | 42.40 | 90.91 |
| | | KNN$_{10}$ | 31.63 | 94.17 | 47.69 | 90.29 | 47.49 | 90.50 | 40.54 | 92.92 | 41.84 | 91.97 | 31.40 | 95.65 | 34.50 | 95.15 |
| | | KNN$_{20}$ | 32.55 | 94.03 | 47.89 | 90.22 | 48.32 | 90.34 | 40.91 | 92.76 | 42.42 | 91.84 | 33.72 | 95.45 | 35.40 | 94.97 |
| | | KNN$_{50}$ | 35.71 | 93.67 | 49.74 | 89.82 | 51.04 | 89.99 | 44.12 | 92.39 | 45.15 | 91.47 | 36.05 | 95.01 | 36.20 | 94.92 |
| | | KNN$_{100}$ | 36.75 | 93.38 | 50.30 | 89.60 | 51.68 | 89.71 | 44.97 | 92.17 | 45.92 | 91.22 | 36.05 | 94.73 | 36.50 | 94.71 |
| | | **Pre-train on 20% IIT-CDIP⁻ (no fine-tune)** | | | | | | | | | | | | | | |
| | — | KNN$_{10}$ | 90.39 | 75.25 | 79.59 | 79.43 | 93.14 | 72.41 | 97.12 | 66.99 | 90.06 | 73.52 | 50.00 | 91.36 | 24.70 | 96.34 |
| | | KNN$_{20}$ | 90.63 | 73.75 | 80.47 | 78.51 | 93.81 | 70.58 | 97.16 | 65.54 | 90.52 | 72.10 | 55.81 | 89.91 | 26.90 | 95.94 |
| | | KNN$_{50}$ | 91.67 | 71.19 | 82.56 | 76.90 | 94.45 | 67.82 | 97.36 | 62.98 | 91.51 | 69.72 | 67.44 | 87.29 | 29.10 | 95.31 |
| | | KNN$_{100}$ | 91.95 | 69.19 | 83.73 | 75.55 | 95.33 | 65.37 | 97.36 | 60.84 | 92.09 | 67.74 | 74.42 | 84.78 | 30.30 | 94.75 |
| **LayoutLMv1$_{Base}$ (40%)** | | **Pre-train on 40% IIT-CDIP⁻ → fine-tune on RVL-CDIP ID data** | | | | | | | | | | | | | | |
| | 96.01 | MSP | 51.76 | 75.76 | 62.39 | 69.63 | 63.37 | 68.75 | 54.22 | 74.03 | 57.94 | 72.04 | 55.81 | 71.69 | 42.50 | 80.56 |
| | | MaxLogit | 42.03 | 91.29 | 54.24 | 89.47 | 57.30 | 84.44 | 45.66 | 90.02 | 49.81 | 88.80 | 52.33 | 93.08 | 33.00 | 92.89 |
| | | Energy | 41.87 | 91.31 | 54.20 | 89.49 | 57.26 | 84.47 | 45.50 | 90.05 | 49.71 | 88.83 | 52.33 | 93.13 | 32.50 | 92.92 |
| | | GradNorm | 38.19 | 91.66 | 53.64 | 86.85 | 55.03 | 86.58 | 43.18 | 91.45 | 47.51 | 88.90 | 52.33 | 92.39 | 34.60 | 92.95 |
| | | KNN$_{10}$ | 31.47 | 94.43 | 47.13 | 90.63 | 48.20 | 90.45 | 38.11 | 93.30 | 41.23 | 92.20 | 27.91 | 95.78 | 24.70 | 96.09 |
| | | KNN$_{20}$ | 32.59 | 94.29 | 47.61 | 90.55 | 49.60 | 90.27 | 39.25 | 93.14 | 42.26 | 92.06 | 32.56 | 95.60 | 25.50 | 95.95 |
| | | KNN$_{50}$ | 34.87 | 93.93 | 49.50 | 90.10 | 52.11 | 89.87 | 42.29 | 92.75 | 44.69 | 91.66 | 38.37 | 95.16 | 26.40 | 95.95 |
| | | KNN$_{100}$ | 36.55 | 93.65 | 50.38 | 89.82 | 53.55 | 89.57 | 43.71 | 92.51 | 46.05 | 91.39 | 43.02 | 94.89 | 27.70 | 95.77 |
| | | **Pre-train on 40% IIT-CDIP⁻ (no fine-tune)** | | | | | | | | | | | | | | |
| | — | KNN$_{10}$ | 87.07 | 80.44 | 71.76 | 83.72 | 86.75 | 82.31 | 96.10 | 76.36 | 85.42 | 80.71 | 75.58 | 84.96 | 5.90 | 98.24 |
| | | KNN$_{20}$ | 88.95 | 79.03 | 74.93 | 82.31 | 88.99 | 81.11 | 96.71 | 75.01 | 87.40 | 79.36 | 80.23 | 82.56 | 7.20 | 97.93 |
| | | KNN$_{50}$ | 91.47 | 77.23 | 80.39 | 79.90 | 91.78 | 79.75 | 97.40 | 72.60 | 90.26 | 77.37 | 87.21 | 78.19 | 9.00 | 97.92 |
| | | KNN$_{100}$ | 90.75 | 75.27 | 84.77 | 77.48 | 91.74 | 78.31 | 97.16 | 70.26 | 91.10 | 75.33 | 89.53 | 74.11 | 14.20 | 97.49 |
| **LayoutLMv1$_{Base}$ (100%)** | | **Pre-train on 100% IIT-CDIP⁻ → fine-tune on RVL-CDIP ID data** | | | | | | | | | | | | | | |
| | 96.38 | MSP | 43.43 | 76.12 | 57.21 | 69.16 | 58.38 | 68.56 | 46.14 | 74.76 | 51.29 | 72.15 | 38.37 | 78.67 | 28.30 | 83.78 |
| | | MaxLogit | 35.19 | 91.29 | 50.22 | 88.98 | 53.19 | 84.54 | 39.98 | 90.71 | 44.64 | 88.88 | 24.42 | 96.39 | 21.40 | 95.57 |
| | | Energy | 35.23 | 91.32 | 50.22 | 89.00 | 53.19 | 84.55 | 39.98 | 90.73 | 44.65 | 88.90 | 24.42 | 96.44 | 21.40 | 95.58 |
| | | GradNorm | 30.30 | 92.54 | 48.61 | 88.18 | 48.96 | 86.58 | 36.16 | 92.63 | 41.01 | 89.98 | 19.77 | 96.71 | 19.20 | 96.35 |
| | | KNN$_{10}$ | 26.50 | 94.95 | 43.47 | 91.69 | 45.09 | 90.95 | 34.09 | 93.86 | 37.29 | 92.86 | 19.77 | 97.39 | 17.80 | 96.37 |
| | | KNN$_{20}$ | 27.22 | 94.83 | 44.07 | 91.58 | 45.41 | 90.79 | 34.62 | 93.71 | 37.83 | 92.73 | 19.77 | 97.22 | 18.40 | 96.26 |
| | | KNN$_{50}$ | 29.46 | 94.49 | 46.28 | 91.12 | 47.69 | 90.45 | 37.50 | 93.33 | 40.23 | 92.35 | 17.44 | 97.04 | 18.70 | 96.80 |
| | | KNN$_{100}$ | 32.15 | 94.26 | 48.17 | 90.85 | 50.64 | 90.21 | 40.38 | 93.12 | 42.83 | 92.11 | 19.77 | 96.88 | 20.70 | 96.74 |
| | | **Pre-train on 100% IIT-CDIP⁻ (no fine-tune)** | | | | | | | | | | | | | | |
| | — | KNN$_{10}$ | 78.74 | 81.67 | 74.45 | 80.86 | 80.53 | 83.71 | 95.01 | 77.33 | 82.18 | 80.89 | 38.37 | 94.62 | 17.70 | 96.12 |
| | | KNN$_{20}$ | 82.39 | 80.13 | 77.86 | 79.31 | 83.48 | 82.75 | 95.45 | 75.93 | 84.80 | 79.53 | 44.19 | 93.42 | 14.60 | 96.13 |
| | | KNN$_{50}$ | 86.03 | 77.65 | 82.80 | 76.60 | 86.91 | 81.30 | 96.10 | 73.07 | 87.96 | 77.16 | 54.65 | 91.09 | 9.60 | 97.21 |
| | | KNN$_{100}$ | 89.11 | 75.51 | 88.03 | 74.08 | 90.62 | 79.78 | 96.71 | 70.43 | 91.12 | 74.95 | 66.28 | 88.50 | 18.00 | 96.82 |

Table 6: OOD detection performance for document classification. Spatial-RoBERTa$_{Base}$ (Pre) or SR$_{Base}$ (Pre) denotes applying the spatial-aware adapter in the word embedding layer. Spatial-RoBERTa$_{Base}$ (Post) or SR$_{Base}$ (Post) denotes applying the spatial-aware adaptor at the output layer.

| Model | ID Acc | Method | OOD Dataset (In-domain) | | | | | | | | | | OOD Dataset (Out-domain) | | | |
| | | | Sci. Report | | Presentation | | Form | | Letter | | Average | | Sci. Poster | | Receipt | |
| | | | FPR95 | AUROC | FPR95 | AUROC | FPR95 | AUROC | FPR95 | AUROC | FPR95 | AUROC | FPR95 | AUROC | FPR95 | AUROC |
|---|---|---|---|---|---|---|---|---|---|---|---|---|---|---|---|---|
| RoBERTa$_{Base}$ | | **Fine-tune on RVL-CDIP (ID)** | | | | | | | | | | | | | | |
| | 90.19 | MSP | 91.19 | 73.70 | 90.84 | 73.49 | 91.82 | 71.53 | 91.03 | 72.35 | 91.22 | 72.77 | 93.02 | 80.94 | 97.60 | 74.59 |
| | | MaxLogit | 96.88 | 79.04 | 96.87 | 79.38 | 98.04 | 75.85 | 98.54 | 77.45 | 97.58 | 77.93 | 100.00 | 82.76 | 99.40 | 79.99 |
| | | Energy | 97.48 | 78.96 | 97.23 | 79.31 | 98.40 | 75.71 | 99.07 | 77.25 | 98.04 | 77.81 | 100.00 | 82.71 | 99.20 | 80.06 |
| | | KNN$_{10}$ | 53.20 | 88.94 | 58.50 | 88.62 | 61.37 | 86.25 | 63.72 | 88.29 | 59.20 | 88.02 | 22.09 | 96.52 | 68.60 | 92.47 |
| | | KNN$_{20}$ | 53.44 | 88.81 | 58.90 | 88.50 | 61.65 | 86.07 | 63.60 | 88.15 | 59.40 | 87.88 | 27.91 | 96.38 | 71.70 | 92.02 |
| | | KNN$_{50}$ | 53.84 | 88.52 | 59.42 | 88.42 | 62.01 | 85.81 | 64.16 | 87.80 | 59.86 | 87.64 | 32.56 | 96.07 | 74.30 | 91.37 |
| | | KNN$_{100}$ | 55.56 | 88.10 | 60.67 | 88.20 | 63.69 | 85.41 | 64.77 | 87.42 | 61.17 | 87.28 | 34.88 | 95.67 | 76.50 | 90.81 |
| | | **No fine-tune** | | | | | | | | | | | | | | |
| | — | KNN$_{10}$ | 93.11 | 63.52 | 88.15 | 66.34 | 94.57 | 66.92 | 98.42 | 53.37 | 93.56 | 62.54 | 25.58 | 95.99 | 86.00 | 72.99 |
| | | KNN$_{20}$ | 92.99 | 63.18 | 88.39 | 65.78 | 94.57 | 66.08 | 98.42 | 52.10 | 93.59 | 61.78 | 26.74 | 95.71 | 87.30 | 70.44 |
| | | KNN$_{50}$ | 92.67 | 62.41 | 89.31 | 64.72 | 94.17 | 64.74 | 98.34 | 50.07 | 93.62 | 60.48 | 26.74 | 95.02 | 90.80 | 66.04 |
| | | KNN$_{100}$ | 92.67 | 61.57 | 89.59 | 63.57 | 94.01 | 63.45 | 98.17 | 48.33 | 93.61 | 59.23 | 29.07 | 94.34 | 92.80 | 61.62 |
| SR$_{Base}$ (Pre) | | **Pre-train on IIT-CDIP → fine-tune on RVL-CDIP (ID)** | | | | | | | | | | | | | | |
| | 97.11 | MSP | 46.80 | 74.52 | 54.64 | 70.58 | 56.26 | 69.72 | 54.30 | 70.74 | 53.00 | 71.39 | 44.19 | 75.79 | 57.20 | 69.23 |
| | | MaxLogit | 39.43 | 88.64 | 46.48 | 89.92 | 49.96 | 85.75 | 48.30 | 87.66 | 46.04 | 87.99 | 33.72 | 93.42 | 50.60 | 88.70 |
| | | Energy | 39.43 | 88.66 | 46.48 | 89.94 | 50.00 | 85.76 | 48.30 | 87.67 | 46.05 | 88.01 | 33.72 | 93.45 | 50.60 | 88.71 |
| | | KNN$_{10}$ | 31.91 | 94.41 | 42.19 | 92.65 | 46.65 | 89.31 | 42.09 | 92.65 | 40.71 | 92.26 | 10.47 | 97.45 | 52.10 | 92.93 |
| | | KNN$_{20}$ | 32.31 | 94.28 | 42.59 | 92.64 | 47.01 | 89.21 | 43.43 | 92.53 | 41.34 | 92.16 | 11.63 | 97.31 | 53.30 | 92.80 |
| | | KNN$_{50}$ | 34.39 | 93.99 | 43.83 | 92.36 | 49.04 | 88.93 | 45.41 | 92.19 | 43.17 | 91.87 | 12.79 | 97.01 | 53.10 | 92.51 |
| | | KNN$_{100}$ | 35.15 | 93.76 | 44.27 | 92.15 | 49.48 | 88.65 | 46.14 | 91.97 | 43.76 | 91.63 | 15.12 | 96.81 | 49.70 | 92.44 |
| | | **Pre-train on IIT-CDIP (no fine-tune)** | | | | | | | | | | | | | | |
| | — | KNN$_{10}$ | 78.82 | 78.92 | 79.99 | 73.89 | 77.69 | 81.32 | 91.48 | 76.52 | 82.00 | 77.66 | 10.47 | 98.08 | 87.30 | 80.89 |
| | | KNN$_{20}$ | 79.74 | 77.95 | 82.64 | 72.17 | 79.81 | 80.40 | 92.13 | 75.11 | 83.58 | 76.41 | 16.28 | 97.60 | 92.10 | 76.94 |
| | | KNN$_{50}$ | 80.42 | 76.87 | 85.13 | 69.62 | 82.12 | 78.93 | 92.98 | 73.01 | 85.16 | 74.61 | 22.09 | 96.66 | 95.20 | 70.53 |
| | | KNN$_{100}$ | 81.43 | 75.70 | 86.90 | 67.19 | 83.40 | 77.12 | 93.38 | 71.07 | 86.28 | 72.77 | 27.91 | 95.86 | 96.60 | 64.56 |
| SR$_{Base}$ (Post) | | **Fine-tune on RVL-CDIP (ID)** | | | | | | | | | | | | | | |
| | 97.10 | MSP | 58.05 | 78.37 | 76.46 | 65.44 | 65.80 | 75.00 | 61.81 | 77.59 | 65.53 | 74.10 | 54.65 | 81.65 | 93.50 | 52.85 |
| | | MaxLogit | 49.20 | 89.82 | 72.36 | 80.28 | 57.82 | 87.28 | 52.52 | 90.04 | 57.98 | 86.86 | 34.88 | 94.88 | 91.60 | 73.37 |
| | | Energy | 47.56 | 89.87 | 71.96 | 80.30 | 56.58 | 87.32 | 51.18 | 90.10 | 56.82 | 86.90 | 34.88 | 95.04 | 91.30 | 73.39 |
| | | KNN$_{10}$ | 37.43 | 93.37 | 64.08 | 86.83 | 49.44 | 89.82 | 46.92 | 92.17 | 49.47 | 90.55 | 26.74 | 96.38 | 90.10 | 80.21 |
| | | KNN$_{20}$ | 38.27 | 93.25 | 65.33 | 86.52 | 50.80 | 89.66 | 48.09 | 91.99 | 50.62 | 90.35 | 26.74 | 96.23 | 91.20 | 79.57 |
| | | KNN$_{50}$ | 40.43 | 92.98 | 67.38 | 86.02 | 52.83 | 89.38 | 50.65 | 91.58 | 52.82 | 89.99 | 26.74 | 95.89 | 92.10 | 78.48 |
| | | KNN$_{100}$ | 41.99 | 92.77 | 67.94 | 85.62 | 53.87 | 89.17 | 51.22 | 91.33 | 53.76 | 89.72 | 29.07 | 95.67 | 92.60 | 77.68 |
| SR$_{Large}$ (Pre) | | **Pre-train on IIT-CDIP → fine-tune on RVL-CDIP (ID)** | | | | | | | | | | | | | | |
| | 97.37 | MSP | 62.37 | 67.82 | 71.27 | 63.36 | 72.87 | 62.54 | 70.25 | 63.84 | 69.19 | 64.39 | 76.74 | 60.61 | 67.00 | 65.48 |
| | | MaxLogit | 33.39 | 90.15 | 39.25 | 89.87 | 42.30 | 88.12 | 37.05 | 91.66 | 38.00 | 89.95 | 31.40 | 92.41 | 27.70 | 94.23 |
| | | Energy | 33.39 | 90.16 | 39.25 | 89.88 | 42.30 | 88.13 | 37.05 | 91.66 | 38.00 | 89.96 | 31.40 | 92.42 | 27.70 | 94.22 |
| | | KNN$_{10}$ | 28.18 | 94.47 | 42.43 | 93.01 | 37.43 | 91.74 | 31.13 | 94.72 | 34.79 | 93.49 | 25.58 | 96.24 | 18.60 | 96.28 |
| | | KNN$_{20}$ | 28.78 | 94.32 | 42.43 | 92.90 | 38.07 | 91.58 | 32.02 | 94.55 | 35.33 | 93.34 | 25.58 | 96.02 | 18.60 | 96.33 |
| | | KNN$_{50}$ | 30.22 | 93.95 | 43.71 | 92.69 | 40.06 | 91.26 | 34.54 | 94.10 | 37.13 | 93.00 | 26.74 | 95.52 | 21.40 | 96.14 |
| | | KNN$_{100}$ | 30.86 | 93.71 | 44.11 | 92.56 | 40.66 | 91.05 | 35.47 | 93.88 | 37.78 | 92.80 | 26.74 | 95.22 | 21.70 | 96.11 |
| | | **Pre-train on IIT-CDIP (no fine-tune)** | | | | | | | | | | | | | | |
| | — | KNN$_{10}$ | 68.49 | 80.43 | 88.23 | 69.83 | 71.75 | 83.11 | 88.11 | 73.32 | 79.14 | 76.67 | 75.58 | 84.36 | 49.80 | 92.02 |
| | | KNN$_{20}$ | 71.74 | 78.77 | 90.24 | 67.41 | 75.66 | 81.38 | 89.04 | 71.14 | 81.67 | 74.68 | 81.40 | 81.55 | 62.20 | 90.29 |
| | | KNN$_{50}$ | 75.46 | 76.49 | 92.81 | 63.82 | 80.17 | 78.72 | 90.42 | 67.84 | 84.72 | 71.72 | 82.56 | 77.15 | 78.20 | 87.49 |
| | | KNN$_{100}$ | 77.62 | 74.59 | 94.42 | 60.94 | 83.16 | 76.25 | 91.80 | 65.30 | 86.75 | 69.27 | 84.88 | 73.34 | 88.20 | 84.96 |

Table 7: OOD detection performance for document classification with the different number of pre-training data from IIT-CDIP.

| ID Acc | Method | Sci. Report FPR95 | AUROC | Presentation FPR95 | AUROC | Form FPR95 | AUROC | Letter FPR95 | AUROC | Average FPR95 | AUROC | Sci. Poster FPR95 | AUROC | Receipt FPR95 | AUROC |
|---|---|---|---|---|---|---|---|---|---|---|---|---|---|---|---|
| | | | | | | OOD Dataset (In-domain) | | | | | | OOD Dataset (Out-domain) | | | |
| **Pre-train on 10% IIT-CDIP→ fine-tune on RVL-CDIP (ID)** | | | | | | | | | | | | | | | |
| 94.89 (ViT_Base 10%) | MSP | 55.80 | 88.37 | 48.61 | 91.38 | 63.93 | 83.83 | 55.52 | 88.55 | 55.96 | 88.03 | 52.05 | 89.60 | 34.10 | 95.04 |
| | MaxLogit | 50.36 | 91.51 | 37.77 | 94.30 | 62.37 | 87.97 | 53.69 | 92.11 | 51.05 | 91.47 | 38.36 | 94.24 | 28.60 | 96.06 |
| | Energy | 50.56 | 91.48 | 37.08 | 94.33 | 63.49 | 87.89 | 55.19 | 92.00 | 51.58 | 91.42 | 38.36 | 94.29 | 29.40 | 95.96 |
| | GradNorm | 55.56 | 79.75 | 45.96 | 84.79 | 66.92 | 74.07 | 58.44 | 81.07 | 56.72 | 79.92 | 47.95 | 82.04 | 34.90 | 91.68 |
| | $KNN_{10}$ | 50.40 | 92.60 | 43.51 | 93.92 | 51.60 | 90.54 | 74.47 | 88.87 | 55.00 | 91.48 | 20.55 | 97.19 | 9.20 | 98.21 |
| | $KNN_{20}$ | 49.80 | 92.70 | 40.38 | 94.43 | 53.39 | 90.26 | 74.72 | 88.77 | 54.57 | 91.54 | 23.29 | 96.98 | 10.40 | 98.05 |
| | $KNN_{50}$ | 46.72 | 92.89 | 34.27 | 95.24 | 56.07 | 89.92 | 74.55 | 88.45 | 52.90 | 91.62 | 27.40 | 96.56 | 12.80 | 97.80 |
| | $KNN_{100}$ | 45.48 | 92.89 | 29.33 | 95.67 | 57.62 | 89.56 | 75.04 | 88.25 | 51.87 | 91.59 | 30.14 | 96.21 | 15.00 | 97.57 |
| **Pre-train on IIT-CDIP (no fine-tune)** | | | | | | | | | | | | | | | |
| – | $KNN_{10}$ | 98.92 | 43.08 | 97.67 | 49.00 | 99.52 | 54.41 | 99.35 | 40.26 | 98.86 | 46.69 | 93.15 | 92.51 | 6.90 | 98.06 |
| | $KNN_{20}$ | 98.88 | 42.47 | 97.75 | 48.57 | 99.52 | 53.75 | 99.35 | 39.56 | 98.88 | 46.09 | 94.52 | 92.24 | 8.60 | 97.91 |
| | $KNN_{50}$ | 98.80 | 41.70 | 97.83 | 48.04 | 99.52 | 52.91 | 99.35 | 38.62 | 98.88 | 45.32 | 95.89 | 91.80 | 10.60 | 97.66 |
| | $KNN_{100}$ | 98.76 | 41.20 | 97.79 | 47.70 | 99.48 | 52.32 | 99.35 | 38.01 | 98.84 | 44.81 | 98.63 | 91.31 | 14.50 | 97.41 |
| **Pre-train on 20% IIT-CDIP→ fine-tune on RVL-CDIP (ID)** | | | | | | | | | | | | | | | |
| 94.62 (ViT_Base 20%) | MSP | 54.36 | 89.01 | 51.63 | 91.31 | 64.57 | 85.23 | 60.51 | 88.67 | 57.77 | 88.56 | 60.27 | 89.34 | 44.20 | 93.73 |
| | MaxLogit | 44.32 | 92.16 | 38.21 | 94.18 | 64.92 | 87.63 | 58.56 | 91.33 | 51.50 | 91.32 | 45.21 | 92.63 | 39.70 | 94.36 |
| | Energy | 44.36 | 92.17 | 37.89 | 94.24 | 66.56 | 87.51 | 60.39 | 91.22 | 52.30 | 91.28 | 46.58 | 92.62 | 41.50 | 94.18 |
| | GradNorm | 90.51 | 54.92 | 92.04 | 51.67 | 94.29 | 45.41 | 98.13 | 32.36 | 93.74 | 46.09 | 95.89 | 40.44 | 89.70 | 59.01 |
| | $KNN_{10}$ | 52.20 | 92.58 | 45.84 | 93.73 | 53.79 | 90.75 | 77.84 | 87.02 | 57.42 | 91.02 | 17.81 | 97.33 | 16.90 | 97.40 |
| | $KNN_{20}$ | 51.60 | 92.66 | 43.55 | 94.15 | 55.63 | 90.46 | 78.04 | 86.79 | 57.20 | 91.02 | 19.18 | 97.06 | 19.40 | 97.11 |
| | $KNN_{50}$ | 50.12 | 92.86 | 39.98 | 94.82 | 58.02 | 90.18 | 78.77 | 86.54 | 56.72 | 91.10 | 19.18 | 96.63 | 23.10 | 96.68 |
| | $KNN_{100}$ | 48.04 | 92.91 | 34.75 | 95.28 | 60.38 | 89.88 | 78.98 | 86.42 | 55.54 | 91.12 | 20.55 | 96.27 | 26.20 | 96.35 |
| **Pre-train on IIT-CDIP (no fine-tune)** | | | | | | | | | | | | | | | |
| – | $KNN_{10}$ | 98.16 | 41.13 | 97.51 | 47.12 | 99.48 | 53.05 | 99.31 | 38.79 | 98.62 | 45.02 | 94.52 | 91.80 | 8.00 | 97.41 |
| | $KNN_{20}$ | 98.12 | 40.71 | 97.51 | 46.79 | 99.48 | 52.52 | 99.31 | 38.31 | 98.60 | 44.58 | 94.52 | 91.48 | 8.70 | 97.25 |
| | $KNN_{50}$ | 98.04 | 40.10 | 97.55 | 46.31 | 99.48 | 51.84 | 99.39 | 37.63 | 98.62 | 43.97 | 95.89 | 91.01 | 11.50 | 96.99 |
| | $KNN_{100}$ | 98.00 | 39.74 | 97.55 | 45.98 | 99.48 | 51.34 | 99.39 | 37.26 | 98.60 | 43.58 | 97.26 | 90.55 | 14.60 | 96.70 |
| **Pre-train on 40% IIT-CDIP→ fine-tune on RVL-CDIP (ID)** | | | | | | | | | | | | | | | |
| 94.63 (ViT_Base 40%) | MSP | 55.48 | 88.65 | 52.27 | 91.54 | 64.49 | 85.52 | 58.08 | 89.20 | 57.58 | 88.73 | 67.12 | 84.62 | 45.80 | 93.82 |
| | MaxLogit | 47.12 | 91.74 | 40.06 | 94.09 | 61.05 | 88.68 | 56.57 | 92.07 | 51.20 | 91.63 | 69.86 | 89.81 | 32.90 | 95.46 |
| | Energy | 47.12 | 91.73 | 39.94 | 94.10 | 62.33 | 88.62 | 58.60 | 91.88 | 52.00 | 91.58 | 69.86 | 89.65 | 32.70 | 95.44 |
| | GradNorm | 47.00 | 85.76 | 41.90 | 89.64 | 60.69 | 81.37 | 53.73 | 87.06 | 50.83 | 85.96 | 64.38 | 81.12 | 34.00 | 92.93 |
| | $KNN_{10}$ | 53.28 | 92.13 | 48.33 | 92.99 | 46.45 | 92.20 | 75.61 | 88.87 | 55.92 | 91.55 | 34.25 | 95.53 | 6.80 | 98.56 |
| | $KNN_{20}$ | 52.76 | 92.24 | 45.88 | 93.57 | 48.12 | 91.95 | 74.84 | 88.75 | 55.40 | 91.63 | 32.88 | 95.21 | 7.80 | 98.36 |
| | $KNN_{50}$ | 51.28 | 92.52 | 40.94 | 94.51 | 50.52 | 91.70 | 75.08 | 88.46 | 54.46 | 91.80 | 35.62 | 94.67 | 10.90 | 98.04 |
| | $KNN_{100}$ | 50.32 | 92.62 | 36.16 | 95.12 | 53.35 | 91.36 | 75.93 | 88.24 | 53.94 | 91.84 | 39.73 | 94.25 | 13.60 | 97.76 |
| **Pre-train on IIT-CDIP (no fine-tune)** | | | | | | | | | | | | | | | |
| – | $KNN_{10}$ | 97.56 | 40.60 | 97.03 | 46.28 | 99.24 | 53.76 | 99.15 | 39.62 | 98.24 | 45.06 | 82.19 | 92.02 | 1.00 | 99.59 |
| | $KNN_{20}$ | 97.56 | 40.00 | 96.95 | 45.86 | 99.24 | 53.18 | 99.15 | 39.12 | 98.22 | 44.54 | 82.19 | 91.63 | 1.00 | 99.55 |
| | $KNN_{50}$ | 97.56 | 39.24 | 96.99 | 45.20 | 99.24 | 52.39 | 99.15 | 38.49 | 98.24 | 43.83 | 86.30 | 91.07 | 1.00 | 99.50 |
| | $KNN_{100}$ | 97.60 | 38.78 | 97.03 | 44.79 | 99.24 | 51.76 | 99.15 | 38.15 | 98.26 | 43.37 | 90.41 | 90.67 | 1.20 | 99.45 |
| **Pre-train on 100% IIT-CDIP→ fine-tune on RVL-CDIP (ID)** | | | | | | | | | | | | | | | |
| 94.79 (ViT_Base 100%) | MSP | 54.28 | 88.80 | 49.14 | 91.80 | 64.60 | 84.45 | 58.85 | 88.78 | 56.72 | 88.46 | 61.64 | 89.44 | 41.00 | 94.27 |
| | MaxLogit | 44.96 | 92.13 | 38.01 | 94.52 | 63.97 | 87.97 | 56.49 | 91.81 | 50.86 | 91.61 | 68.49 | 90.65 | 34.60 | 95.26 |
| | Energy | 45.72 | 92.11 | 38.01 | 94.55 | 65.84 | 87.86 | 57.91 | 91.70 | 51.87 | 91.56 | 72.60 | 90.41 | 34.80 | 95.14 |
| | GradNorm | 48.72 | 84.21 | 44.36 | 87.50 | 63.49 | 78.07 | 56.25 | 84.79 | 53.20 | 83.64 | 60.27 | 82.96 | 35.60 | 91.24 |
| | $KNN_{10}$ | 45.16 | 93.14 | 39.13 | 94.62 | 51.68 | 90.85 | 73.58 | 88.81 | 52.39 | 91.86 | 50.68 | 93.09 | 10.40 | 98.04 |
| | $KNN_{20}$ | 44.88 | 93.14 | 36.64 | 95.04 | 53.35 | 90.59 | 74.27 | 88.67 | 52.28 | 91.86 | 50.68 | 92.67 | 12.00 | 97.81 |
| | $KNN_{50}$ | 43.67 | 93.19 | 31.18 | 95.60 | 56.74 | 90.29 | 75.28 | 88.49 | 51.72 | 91.89 | 57.53 | 92.23 | 15.60 | 97.45 |
| | $KNN_{100}$ | 43.63 | 93.15 | 27.52 | 95.94 | 58.74 | 90.02 | 76.18 | 88.38 | 51.52 | 91.87 | 61.64 | 92.01 | 18.90 | 97.18 |
| **Pre-train on IIT-CDIP (no fine-tune)** | | | | | | | | | | | | | | | |
| – | $KNN_{10}$ | 97.04 | 42.35 | 93.97 | 50.17 | 97.41 | 52.68 | 98.01 | 43.19 | 96.61 | 47.10 | 12.33 | 97.47 | 3.10 | 98.38 |
| | $KNN_{20}$ | 97.16 | 41.99 | 94.01 | 49.96 | 97.81 | 52.01 | 98.09 | 42.73 | 96.77 | 46.67 | 15.07 | 96.95 | 3.00 | 98.31 |
| | $KNN_{50}$ | 96.96 | 41.62 | 94.34 | 49.56 | 98.00 | 51.20 | 98.05 | 42.24 | 96.84 | 46.16 | 21.92 | 96.08 | 2.70 | 98.18 |
| | $KNN_{100}$ | 97.00 | 41.48 | 94.90 | 49.31 | 98.12 | 50.65 | 98.13 | 42.03 | 97.04 | 45.87 | 36.99 | 95.29 | 2.30 | 98.27 |

Table 8: OOD detection performance for document classification. $Longformer_{4096}$ denotes the original model adopted from the Huggingface model hub. $Longformer_{4096}$ (+) denotes the additional pre-training on IIT-CDIP.

| ID Acc | Method | Sci. Report FPR95 | AUROC | Presentation FPR95 | AUROC | Form FPR95 | AUROC | Letter FPR95 | AUROC | Average FPR95 | AUROC | Sci. Poster FPR95 | AUROC | Receipt FPR95 | AUROC |
|---|---|---|---|---|---|---|---|---|---|---|---|---|---|---|---|
| | | | | | | OOD Dataset (In-domain) | | | | | | OOD Dataset (Out-domain) | | | |
| **Fine-tune on RVL-CDIP (ID)** ($Longformer_{4096}$) | | | | | | | | | | | | | | | |
| 90.71 | MSP | 95.00 | 64.32 | 95.62 | 62.17 | 95.89 | 60.53 | 93.95 | 66.89 | 95.12 | 63.48 | 88.37 | 77.50 | 98.60 | 54.72 |
| | MaxLogit | 97.12 | 72.84 | 97.07 | 75.22 | 98.24 | 70.39 | 95.82 | 77.57 | 97.06 | 74.00 | 90.70 | 86.62 | 99.60 | 68.10 |
| | Energy | 97.48 | 72.82 | 97.35 | 75.21 | 98.36 | 70.37 | 96.59 | 77.56 | 97.44 | 73.99 | 91.86 | 86.63 | 99.80 | 68.08 |
| | $KNN_{10}$ | 58.45 | 88.21 | 65.65 | 86.88 | 67.80 | 83.99 | 56.78 | 89.53 | 62.17 | 87.15 | 27.91 | 96.01 | 82.10 | 86.31 |
| | $KNN_{20}$ | 58.97 | 88.04 | 65.57 | 86.60 | 68.12 | 83.80 | 57.35 | 89.34 | 62.50 | 86.94 | 29.07 | 95.82 | 82.60 | 85.93 |
| | $KNN_{50}$ | 60.25 | 87.64 | 66.57 | 86.25 | 68.91 | 83.41 | 58.81 | 88.96 | 63.64 | 86.56 | 30.23 | 95.46 | 82.70 | 85.27 |
| | $KNN_{100}$ | 61.97 | 87.19 | 68.14 | 85.81 | 70.15 | 82.95 | 60.47 | 88.60 | 65.18 | 86.14 | 34.88 | 95.04 | 82.80 | 84.75 |
| **No fine-tune** | | | | | | | | | | | | | | | |
| – | $KNN_{10}$ | 98.04 | 55.45 | 97.63 | 59.97 | 98.76 | 51.75 | 98.13 | 53.16 | 98.14 | 55.08 | 70.93 | 88.69 | 100.00 | 64.97 |
| | $KNN_{20}$ | 98.12 | 55.19 | 97.67 | 59.64 | 98.80 | 51.27 | 98.17 | 52.71 | 98.19 | 54.70 | 70.93 | 88.51 | 100.00 | 64.08 |
| | $KNN_{50}$ | 98.00 | 54.82 | 97.63 | 59.13 | 98.80 | 50.57 | 98.30 | 52.07 | 98.18 | 54.15 | 73.26 | 88.29 | 100.00 | 62.82 |
| | $KNN_{100}$ | 97.92 | 54.48 | 97.67 | 58.62 | 98.84 | 50.00 | 98.34 | 51.62 | 98.19 | 53.68 | 74.42 | 88.14 | 100.00 | 61.70 |
| **Pre-train on IIT-CDIP→ fine-tune on RVL-CDIP (ID)** ($Longformer_{4096}$ (+)) | | | | | | | | | | | | | | | |
| 91.13 | MSP | 95.20 | 64.08 | 95.62 | 61.38 | 96.05 | 59.47 | 94.48 | 63.13 | 95.34 | 62.02 | 90.70 | 67.26 | 98.00 | 55.52 |
| | MaxLogit | 96.96 | 75.41 | 96.54 | 76.03 | 97.89 | 70.15 | 96.71 | 74.56 | 97.02 | 74.04 | 100.00 | 78.65 | 99.70 | 72.88 |
| | Energy | 97.28 | 75.40 | 96.54 | 76.03 | 98.28 | 70.14 | 97.16 | 74.55 | 97.32 | 74.03 | 100.00 | 78.59 | 99.70 | 72.86 |
| | $KNN_{10}$ | 58.73 | 89.25 | 66.21 | 87.57 | 72.03 | 83.76 | 63.68 | 88.72 | 65.16 | 87.32 | 48.84 | 94.78 | 86.40 | 87.84 |
| | $KNN_{20}$ | 58.61 | 89.18 | 65.97 | 87.45 | 71.67 | 83.69 | 63.39 | 88.61 | 64.91 | 87.23 | 48.84 | 94.62 | 85.30 | 87.70 |
| | $KNN_{50}$ | 61.17 | 88.96 | 66.97 | 87.29 | 72.83 | 83.47 | 65.83 | 88.33 | 66.70 | 87.01 | 55.81 | 94.25 | 85.70 | 87.39 |
| | $KNN_{100}$ | 61.73 | 88.79 | 66.93 | 87.11 | 73.30 | 83.24 | 66.15 | 88.15 | 67.03 | 86.82 | 55.81 | 94.00 | 84.70 | 87.21 |
| **Pre-train on IIT-CDIP (no fine-tune)** | | | | | | | | | | | | | | | |
| – | $KNN_{10}$ | 95.48 | 61.40 | 98.07 | 53.66 | 97.73 | 55.55 | 98.66 | 48.70 | 97.49 | 54.83 | 81.40 | 91.12 | 97.40 | 46.27 |
| | $KNN_{20}$ | 95.56 | 60.92 | 97.95 | 52.95 | 97.49 | 54.97 | 98.50 | 48.21 | 97.38 | 54.26 | 84.88 | 90.62 | 97.50 | 45.55 |
| | $KNN_{50}$ | 95.60 | 59.94 | 97.95 | 51.77 | 97.41 | 53.97 | 98.62 | 47.29 | 97.40 | 53.24 | 87.21 | 89.95 | 98.20 | 44.18 |
| | $KNN_{100}$ | 95.60 | 59.04 | 97.99 | 50.74 | 97.21 | 52.99 | 98.58 | 46.51 | 97.34 | 52.32 | 88.37 | 89.52 | 98.50 | 43.09 |

Table 9: OOD detection performance for document classification. All models are pre-trained on ImageNet.

| | ID Acc | Method | OOD Dataset (In-domain) | | | | | | | | | | OOD Dataset (Out-domain) | | | |
| --- | --- | --- | --- | --- | --- | --- | --- | --- | --- | --- | --- | --- | --- | --- | --- | --- |
| | | | Sci. Report | | Presentation | | Form | | Letter | | *Average* | | Sci. Poster | | Receipt | |
| | | | FPR95 | AUROC | FPR95 | AUROC | FPR95 | AUROC | FPR95 | AUROC | FPR95 | AUROC | FPR95 | AUROC | FPR95 | AUROC |
| **ResNet-50** | | Pre-train on ImageNet→ fine-tune on RVL-CDIP (ID) | | | | | | | | | | | | | | |
| | 91.12 | MSP | 64.49 | 87.87 | 55.89 | 90.94 | 66.60 | 87.31 | 77.88 | 80.87 | 66.22 | 86.75 | 51.16 | 92.76 | 63.10 | 90.36 |
| | | MaxLogit | 64.89 | 88.59 | 47.97 | 92.81 | 65.40 | 87.52 | 77.56 | 81.87 | 63.96 | 87.70 | 41.86 | 94.62 | 54.00 | 93.29 |
| | | Energy | 67.09 | 88.30 | 47.81 | 92.86 | 66.68 | 87.24 | 78.53 | 81.75 | 65.03 | 87.54 | 39.53 | 94.73 | 48.50 | 93.68 |
| | | $KNN_{10}$ | 73.38 | 86.82 | 67.98 | 87.46 | 71.31 | 87.84 | 92.90 | 77.74 | 76.39 | 84.96 | 6.98 | 99.12 | 5.20 | 98.98 |
| | | $KNN_{20}$ | 74.90 | 86.41 | 66.29 | 87.79 | 73.82 | 87.21 | 93.95 | 76.51 | 77.24 | 84.48 | 6.98 | 98.96 | 5.50 | 98.85 |
| | | $KNN_{50}$ | 76.66 | 86.04 | 66.41 | 88.48 | 78.29 | 86.39 | 95.50 | 74.76 | 79.22 | 83.92 | 5.81 | 98.68 | 5.90 | 98.70 |
| | | $KNN_{100}$ | 77.54 | 85.61 | 65.41 | 88.99 | 82.16 | 85.43 | 96.23 | 73.37 | 80.33 | 83.35 | 6.98 | 98.34 | 6.30 | 98.51 |
| | | Pre-train on ImageNet | | | | | | | | | | | | | | |
| | – | $KNN_{10}$ | 96.96 | 51.14 | 94.62 | 51.75 | 98.76 | 53.84 | 99.59 | 37.60 | 97.48 | 48.58 | 83.56 | 85.00 | 20.80 | 97.00 |
| | | $KNN_{20}$ | 96.96 | 50.37 | 94.34 | 51.54 | 98.92 | 52.98 | 99.59 | 36.60 | 97.45 | 47.87 | 83.56 | 84.49 | 22.70 | 96.71 |
| | | $KNN_{50}$ | 96.92 | 49.29 | 94.29 | 51.30 | 99.00 | 51.84 | 99.59 | 35.15 | 97.45 | 46.90 | 83.56 | 84.03 | 26.70 | 96.21 |
| | | $KNN_{100}$ | 97.12 | 48.60 | 94.54 | 51.25 | 99.16 | 51.11 | 99.55 | 34.36 | 97.59 | 46.33 | 82.19 | 83.31 | 29.40 | 95.67 |
| **Swin$_{Base}$** | | Pre-train on ImageNet→ fine-tune on RVL-CDIP (ID) | | | | | | | | | | | | | | |
| | 95.74 | MSP | 47.64 | 88.09 | 49.90 | 88.11 | 58.22 | 83.14 | 50.28 | 88.90 | 51.51 | 87.06 | 49.32 | 91.31 | 36.50 | 93.63 |
| | | MaxLogit | 42.39 | 93.11 | 42.47 | 93.45 | 58.62 | 88.79 | 45.90 | 93.18 | 47.34 | 92.13 | 50.68 | 92.50 | 32.20 | 95.65 |
| | | Energy | 43.15 | 93.05 | 42.95 | 93.40 | 59.02 | 88.70 | 46.71 | 93.07 | 47.96 | 92.06 | 52.05 | 92.38 | 33.60 | 95.49 |
| | | $KNN_{10}$ | 49.44 | 92.82 | 46.73 | 92.87 | 42.90 | 92.57 | 72.69 | 88.45 | 52.94 | 91.68 | 16.44 | 96.73 | 6.10 | 98.30 |
| | | $KNN_{20}$ | 48.84 | 92.95 | 43.27 | 93.51 | 44.53 | 92.32 | 72.28 | 88.35 | 52.23 | 91.78 | 17.81 | 96.52 | 7.40 | 98.10 |
| | | $KNN_{50}$ | 46.44 | 93.26 | 39.25 | 94.57 | 47.41 | 92.09 | 73.34 | 87.87 | 51.61 | 91.95 | 26.03 | 96.15 | 8.60 | 97.80 |
| | | $KNN_{100}$ | 43.76 | 93.42 | 35.03 | 95.29 | 50.08 | 91.72 | 75.77 | 87.42 | 51.16 | 91.96 | 28.77 | 95.94 | 11.30 | 97.55 |
| | | Pre-train on ImageNet | | | | | | | | | | | | | | |
| | – | $KNN_{10}$ | 98.56 | 52.75 | 95.06 | 55.14 | 99.36 | 58.85 | 99.80 | 41.86 | 98.20 | 52.15 | 65.75 | 93.26 | 2.10 | 99.35 |
| | | $KNN_{20}$ | 98.44 | 51.86 | 95.18 | 54.72 | 99.32 | 57.88 | 99.80 | 40.66 | 98.18 | 51.28 | 68.49 | 92.52 | 2.60 | 99.22 |
| | | $KNN_{50}$ | 98.52 | 50.69 | 95.38 | 54.13 | 99.16 | 56.61 | 99.76 | 39.01 | 98.20 | 50.11 | 78.08 | 91.14 | 3.40 | 98.99 |
| | | $KNN_{100}$ | 98.72 | 49.96 | 95.66 | 53.80 | 99.16 | 55.84 | 99.76 | 38.16 | 98.32 | 49.44 | 79.45 | 89.89 | 4.30 | 98.77 |
| **ViT$_{Base}$** | | Pre-train on ImageNet→ fine-tune on RVL-CDIP (ID) | | | | | | | | | | | | | | |
| | 94.38 | MSP | 56.81 | 89.14 | 52.19 | 91.80 | 67.48 | 84.26 | 59.90 | 88.77 | 59.10 | 88.49 | 47.67 | 92.98 | 59.50 | 91.99 |
| | | MaxLogit | 50.76 | 91.37 | 44.60 | 93.75 | 68.04 | 86.94 | 55.15 | 91.81 | 54.64 | 90.97 | 40.70 | 94.20 | 52.40 | 93.16 |
| | | Energy | 51.16 | 91.31 | 44.52 | 93.75 | 69.43 | 86.81 | 56.09 | 91.77 | 55.30 | 90.91 | 38.37 | 94.11 | 53.20 | 93.11 |
| | | $KNN_{10}$ | 62.57 | 90.12 | 57.73 | 90.91 | 53.67 | 90.36 | 84.50 | 86.19 | 64.62 | 89.40 | 12.79 | 97.96 | 13.00 | 97.92 |
| | | $KNN_{20}$ | 63.01 | 90.24 | 56.01 | 91.51 | 55.03 | 90.02 | 84.38 | 86.01 | 64.61 | 89.44 | 15.12 | 97.76 | 14.90 | 97.67 |
| | | $KNN_{50}$ | 61.97 | 90.62 | 53.23 | 92.62 | 58.26 | 89.57 | 84.25 | 85.64 | 64.43 | 89.61 | 16.28 | 97.38 | 19.80 | 97.24 |
| | | $KNN_{100}$ | 60.29 | 90.85 | 49.70 | 93.53 | 60.38 | 89.07 | 84.01 | 85.43 | 63.60 | 89.72 | 16.28 | 97.05 | 23.60 | 96.82 |
| | | Pre-train on ImageNet | | | | | | | | | | | | | | |
| | – | $KNN_{10}$ | 98.48 | 52.15 | 95.02 | 56.94 | 99.48 | 53.77 | 99.47 | 38.90 | 98.11 | 50.44 | 93.15 | 90.27 | 20.40 | 97.13 |
| | | $KNN_{20}$ | 98.48 | 51.41 | 95.06 | 56.61 | 99.44 | 52.92 | 99.55 | 37.61 | 98.13 | 49.64 | 94.52 | 89.44 | 22.60 | 96.80 |
| | | $KNN_{50}$ | 98.32 | 50.43 | 94.86 | 56.21 | 99.40 | 51.86 | 99.59 | 35.82 | 98.04 | 48.58 | 97.26 | 88.23 | 26.60 | 96.25 |
| | | $KNN_{100}$ | 98.40 | 49.76 | 95.06 | 55.90 | 99.44 | 51.15 | 99.59 | 34.59 | 98.12 | 47.85 | 98.63 | 87.24 | 31.20 | 95.76 |

Table 10: OOD detection performance for document classification (select OOD categories achieve the best performance across most of the models with different modalities).

| | ID Acc | Method | OOD Dataset (In-domain) | | | | | | | | | | OOD Dataset (Out-domain) | | | |
| | | | Email | | Resume | | File folder | | Sci. publication | | Average | | Sci. Poster | | Receipt | |
| | | | FPR95 | AUROC | FPR95 | AUROC | FPR95 | AUROC | FPR95 | AUROC | FPR95 | AUROC | FPR95 | AUROC | FPR95 | AUROC |
|---|---|---|---|---|---|---|---|---|---|---|---|---|---|---|---|---|
| RoBERTa_Base | | **Pre-train on pure-text data→ fine-tune on RVL-CDIP (ID)** | | | | | | | | | | | | | | |
| | 86.13 | MSP | 96.22 | 60.38 | 90.67 | 71.72 | 93.82 | 59.47 | 93.86 | 65.51 | 93.64 | 64.27 | 91.86 | 70.57 | 93.00 | 69.99 |
| | | MaxLogit | 99.21 | 66.57 | 95.80 | 73.66 | 95.47 | 66.81 | 97.09 | 65.63 | 96.89 | 68.17 | 94.19 | 77.17 | 94.60 | 74.69 |
| | | Energy | 99.60 | 66.53 | 96.64 | 73.57 | 95.14 | 66.82 | 97.21 | 65.35 | 97.15 | 68.07 | 94.19 | 77.44 | 95.60 | 74.90 |
| | | KNN$_{10}$ | 83.70 | 82.77 | 69.02 | 84.28 | 88.32 | 74.06 | 86.11 | 74.02 | 81.79 | 78.78 | 43.02 | 92.74 | 72.00 | 88.87 |
| | | KNN$_{20}$ | 84.50 | 82.35 | 69.06 | 84.21 | 88.20 | 73.71 | 86.72 | 74.02 | 82.12 | 78.57 | 48.84 | 92.38 | 73.80 | 88.31 |
| | | KNN$_{50}$ | 84.98 | 81.57 | 68.86 | 84.06 | 88.08 | 73.01 | 87.08 | 73.94 | 82.25 | 78.14 | 54.65 | 91.92 | 75.40 | 87.44 |
| | | KNN$_{100}$ | 86.25 | 80.88 | 70.26 | 83.80 | 88.28 | 72.40 | 87.44 | 73.89 | 83.06 | 77.74 | 58.14 | 91.50 | 78.20 | 86.68 |
| | | **Pre-train on pure-text data** | | | | | | | | | | | | | | |
| | — | KNN$_{10}$ | 86.09 | 75.63 | 95.12 | 58.62 | 97.71 | 59.75 | 98.95 | 50.54 | 94.47 | 61.14 | 10.47 | 98.46 | 89.80 | 63.01 |
| | | KNN$_{20}$ | 86.29 | 74.92 | 95.00 | 58.14 | 97.71 | 58.88 | 99.03 | 49.49 | 94.51 | 60.36 | 12.79 | 98.35 | 90.80 | 60.59 |
| | | KNN$_{50}$ | 87.32 | 73.55 | 94.64 | 57.53 | 97.83 | 57.56 | 99.15 | 48.11 | 94.73 | 59.19 | 12.79 | 98.11 | 93.30 | 56.61 |
| | | KNN$_{100}$ | 89.27 | 72.48 | 94.28 | 57.12 | 97.99 | 56.52 | 99.11 | 47.37 | 95.16 | 58.37 | 11.63 | 97.89 | 94.30 | 52.98 |
| Longformer$_{4096}$ | | **Pre-train on pure-text data→ fine-tune on RVL-CDIP (ID)** | | | | | | | | | | | | | | |
| | 88.34 | MSP | 96.90 | 60.55 | 96.20 | 59.14 | 96.31 | 55.72 | 97.82 | 55.12 | 96.81 | 57.63 | 95.35 | 80.44 | 99.60 | 52.82 |
| | | MaxLogit | 98.97 | 68.97 | 97.60 | 65.64 | 95.67 | 63.42 | 98.63 | 62.87 | 97.72 | 65.23 | 97.67 | 88.42 | 99.70 | 71.54 |
| | | Energy | 99.44 | 68.96 | 97.92 | 65.63 | 95.83 | 63.42 | 98.71 | 62.83 | 97.98 | 65.21 | 97.67 | 88.46 | 99.90 | 71.55 |
| | | KNN$_{10}$ | 68.28 | 88.72 | 69.62 | 83.36 | 78.17 | 85.08 | 90.88 | 74.98 | 76.74 | 83.04 | 16.28 | 96.90 | 81.60 | 86.94 |
| | | KNN$_{20}$ | 68.04 | 88.61 | 70.10 | 83.22 | 77.53 | 84.92 | 90.75 | 74.95 | 76.60 | 82.92 | 16.28 | 96.84 | 81.80 | 86.49 |
| | | KNN$_{50}$ | 69.28 | 88.29 | 70.98 | 82.92 | 78.29 | 84.46 | 90.96 | 74.82 | 77.38 | 82.62 | 19.77 | 96.59 | 83.40 | 85.71 |
| | | KNN$_{100}$ | 69.28 | 88.15 | 71.34 | 82.69 | 78.49 | 84.21 | 90.43 | 74.86 | 77.39 | 82.48 | 22.09 | 96.38 | 83.90 | 85.17 |
| | | **Pre-train on pure-text data** | | | | | | | | | | | | | | |
| | — | KNN$_{10}$ | 97.42 | 47.77 | 95.72 | 50.09 | 97.67 | 46.58 | 99.52 | 38.61 | 97.58 | 45.76 | 45.35 | 93.92 | 100.00 | 63.03 |
| | | KNN$_{20}$ | 97.46 | 46.91 | 95.60 | 49.80 | 97.71 | 46.02 | 99.52 | 38.21 | 97.57 | 45.24 | 46.51 | 93.77 | 100.00 | 61.92 |
| | | KNN$_{50}$ | 97.58 | 45.68 | 95.56 | 49.45 | 97.75 | 45.19 | 99.52 | 37.72 | 97.60 | 44.51 | 50.00 | 93.60 | 100.00 | 60.35 |
| | | KNN$_{100}$ | 97.66 | 44.78 | 95.60 | 49.17 | 97.87 | 44.63 | 99.56 | 37.57 | 97.67 | 44.04 | 51.16 | 93.48 | 100.00 | 58.89 |
| ResNet-50 | | **Pre-train on ImageNet→ fine-tune on RVL-CDIP (ID)** | | | | | | | | | | | | | | |
| | 85.25 | MSP | 60.53 | 87.26 | 69.53 | 87.00 | 27.86 | 95.13 | 94.05 | 75.79 | 62.99 | 86.30 | 91.78 | 74.40 | 27.80 | 95.47 |
| | | MaxLogit | 59.98 | 89.27 | 72.61 | 88.02 | 30.04 | 95.41 | 93.39 | 75.38 | 64.00 | 87.02 | 80.82 | 79.89 | 30.00 | 95.29 |
| | | Energy | 63.71 | 89.14 | 75.64 | 87.55 | 45.71 | 94.15 | 92.77 | 75.02 | 69.46 | 86.46 | 78.08 | 81.07 | 62.20 | 93.44 |
| | | KNN$_{10}$ | 72.46 | 85.68 | 85.69 | 85.30 | 68.62 | 76.01 | 96.15 | 55.35 | 80.73 | 75.59 | 36.99 | 94.56 | 2.20 | 99.37 |
| | | KNN$_{20}$ | 76.15 | 84.55 | 88.65 | 84.22 | 66.13 | 80.67 | 96.54 | 56.31 | 81.87 | 76.44 | 38.36 | 93.81 | 2.70 | 99.28 |
| | | KNN$_{50}$ | 80.37 | 82.61 | 92.00 | 82.49 | 60.98 | 86.77 | 96.93 | 59.06 | 82.57 | 77.73 | 47.95 | 92.42 | 3.80 | 99.11 |
| | | KNN$_{100}$ | 84.70 | 80.54 | 95.15 | 80.64 | 51.29 | 91.78 | 97.16 | 61.19 | 82.08 | 78.54 | 50.68 | 91.01 | 4.70 | 98.91 |
| | | **Pre-train on ImageNet** | | | | | | | | | | | | | | |
| | — | KNN$_{10}$ | 99.72 | 40.94 | 99.65 | 21.52 | 52.47 | 91.03 | 98.33 | 45.40 | 87.54 | 49.72 | 84.93 | 84.38 | 20.40 | 97.12 |
| | | KNN$_{20}$ | 99.68 | 41.18 | 99.65 | 20.68 | 50.61 | 91.63 | 98.41 | 44.65 | 87.09 | 49.54 | 86.30 | 83.94 | 23.40 | 96.87 |
| | | KNN$_{50}$ | 99.64 | 41.58 | 99.65 | 19.48 | 46.97 | 92.36 | 98.37 | 43.49 | 86.16 | 49.23 | 84.93 | 83.70 | 26.90 | 96.43 |
| | | KNN$_{100}$ | 99.64 | 42.19 | 99.65 | 18.98 | 44.91 | 92.84 | 98.33 | 42.86 | 85.63 | 49.22 | 84.12 | 83.12 | 29.20 | 95.98 |
| Swin$_{Base}$ | | **Pre-train on ImageNet→ fine-tune on RVL-CDIP (ID)** | | | | | | | | | | | | | | |
| | 91.25 | MSP | 70.23 | 81.87 | 67.68 | 85.31 | 43.97 | 92.68 | 83.78 | 79.40 | 66.42 | 84.82 | 86.30 | 78.23 | 54.10 | 91.62 |
| | | MaxLogit | 54.73 | 87.04 | 46.51 | 92.30 | 17.25 | 96.51 | 90.86 | 74.11 | 52.34 | 87.49 | 82.19 | 83.20 | 34.40 | 94.82 |
| | | Energy | 54.05 | 87.11 | 44.38 | 92.49 | 16.38 | 96.63 | 91.29 | 73.59 | 51.53 | 87.46 | 84.93 | 83.07 | 33.80 | 94.82 |
| | | KNN$_{10}$ | 56.08 | 90.66 | 48.80 | 92.84 | 38.31 | 93.31 | 91.02 | 62.91 | 58.55 | 85.93 | 27.40 | 96.03 | 3.30 | 98.84 |
| | | KNN$_{20}$ | 54.61 | 90.95 | 49.98 | 92.68 | 27.58 | 95.24 | 91.44 | 68.54 | 55.90 | 86.85 | 26.03 | 96.35 | 4.00 | 98.76 |
| | | KNN$_{50}$ | 55.25 | 90.68 | 52.15 | 92.37 | 15.75 | 97.28 | 91.25 | 71.62 | 53.60 | 87.99 | 28.77 | 96.10 | 4.90 | 98.59 |
| | | KNN$_{100}$ | 56.20 | 90.31 | 54.75 | 92.17 | 9.14 | 98.00 | 91.13 | 75.11 | 52.80 | 88.90 | 30.14 | 95.77 | 6.50 | 98.35 |
| | | **Pre-train on ImageNet** | | | | | | | | | | | | | | |
| | — | KNN$_{10}$ | 99.84 | 43.55 | 99.76 | 20.64 | 47.92 | 93.20 | 98.91 | 37.55 | 86.61 | 48.74 | 58.90 | 93.88 | 1.60 | 99.32 |
| | | KNN$_{20}$ | 99.84 | 43.78 | 99.76 | 19.61 | 44.76 | 93.61 | 98.91 | 37.01 | 85.82 | 48.50 | 65.75 | 93.42 | 2.10 | 99.20 |
| | | KNN$_{50}$ | 99.84 | 44.47 | 99.80 | 18.36 | 41.31 | 94.14 | 99.03 | 36.45 | 85.00 | 48.36 | 72.60 | 92.69 | 2.60 | 99.00 |
| | | KNN$_{100}$ | 99.88 | 45.26 | 99.80 | 17.92 | 39.97 | 94.39 | 99.03 | 36.71 | 84.67 | 48.57 | 79.45 | 91.97 | 3.70 | 98.81 |
| ViT$_{Base}$ | | **Pre-train on ImageNet→ fine-tune on RVL-CDIP (ID)** | | | | | | | | | | | | | | |
| | 89.97 | MSP | 61.25 | 85.84 | 66.57 | 85.04 | 40.44 | 93.10 | 85.84 | 81.83 | 63.52 | 86.45 | 73.97 | 80.66 | 60.30 | 90.41 |
| | | MaxLogit | 53.02 | 90.37 | 55.07 | 88.86 | 19.91 | 96.25 | 92.38 | 79.69 | 55.27 | 88.79 | 76.71 | 85.16 | 50.60 | 93.12 |
| | | Energy | 51.79 | 90.49 | 55.07 | 89.03 | 17.53 | 96.53 | 92.69 | 79.20 | 54.27 | 88.81 | 79.45 | 85.01 | 50.10 | 93.20 |
| | | KNN$_{10}$ | 54.13 | 91.18 | 52.86 | 91.18 | 58.49 | 87.46 | 92.88 | 65.98 | 64.59 | 83.95 | 42.47 | 95.07 | 11.00 | 97.94 |
| | | KNN$_{20}$ | 54.21 | 91.18 | 53.17 | 90.99 | 50.61 | 89.35 | 93.04 | 67.52 | 62.76 | 84.76 | 43.84 | 94.98 | 13.10 | 97.62 |
| | | KNN$_{50}$ | 54.53 | 91.05 | 53.33 | 90.79 | 41.95 | 92.82 | 93.00 | 72.06 | 60.70 | 86.68 | 42.47 | 94.74 | 17.30 | 97.12 |
| | | KNN$_{100}$ | 54.65 | 90.81 | 54.12 | 90.56 | 30.79 | 95.78 | 93.04 | 75.39 | 58.15 | 88.14 | 45.21 | 94.24 | 22.00 | 96.58 |
| | | **Pre-train on ImageNet** | | | | | | | | | | | | | | |
| | — | KNN$_{10}$ | 99.80 | 46.46 | 99.68 | 26.50 | 58.65 | 90.61 | 98.72 | 46.40 | 89.21 | 52.49 | 87.67 | 91.39 | 19.90 | 97.25 |
| | | KNN$_{20}$ | 99.80 | 46.02 | 99.65 | 25.69 | 57.30 | 91.01 | 98.72 | 46.46 | 88.87 | 52.30 | 90.41 | 90.87 | 21.70 | 97.01 |
| | | KNN$_{50}$ | 99.80 | 45.48 | 99.61 | 24.76 | 55.16 | 91.52 | 98.76 | 46.69 | 88.33 | 52.11 | 94.52 | 89.99 | 24.30 | 96.62 |
| | | KNN$_{100}$ | 99.80 | 45.33 | 99.65 | 24.43 | 54.81 | 91.90 | 98.72 | 47.10 | 88.24 | 52.19 | 95.89 | 89.31 | 28.80 | 96.27 |

Table 11: OOD detection performance for document classification (randomly select four categories as OOD).

| | ID Acc | Method | Letter FPR95 | Letter AUROC | Handwritten FPR95 | Handwritten AUROC | Advertisement FPR95 | Advertisement AUROC | Memo FPR95 | Memo AUROC | Average FPR95 | Average AUROC | Sci. Poster FPR95 | Sci. Poster AUROC | Receipt FPR95 | Receipt AUROC |
|---|---|---|---|---|---|---|---|---|---|---|---|---|---|---|---|---|
| | | | | | | | | OOD Dataset (In-domain) | | | | | | OOD Dataset (Out-domain) | | |
| **RoBERTa**Base | 88.86 | *Pre-train on pure-text data→ fine-tune on RVL-CDIP (ID)* | | | | | | | | | | | | | | |
| | | MSP | 70.22 | 79.21 | 50.14 | 87.24 | 84.64 | 67.80 | 91.42 | 57.99 | 74.10 | 73.06 | 95.35 | 59.75 | 94.30 | 55.12 |
| | | MaxLogit | 66.04 | 87.51 | 39.65 | 92.53 | 86.47 | 77.03 | 91.67 | 71.84 | 70.96 | 82.23 | 100.00 | 77.89 | 96.80 | 71.96 |
| | | Energy | 66.20 | 87.57 | 38.19 | 92.59 | 87.35 | 77.03 | 91.67 | 71.89 | 70.85 | 82.27 | 100.00 | 77.92 | 96.80 | 71.96 |
| | | KNN$_{10}$ | 62.62 | 80.19 | 60.98 | 70.90 | 75.62 | 80.24 | 85.84 | 69.20 | 71.26 | 75.13 | 94.19 | 81.99 | 90.40 | 82.48 |
| | | KNN$_{20}$ | 63.18 | 80.10 | 60.07 | 71.17 | 75.90 | 80.03 | 85.72 | 68.88 | 71.22 | 75.04 | 94.19 | 81.75 | 91.20 | 81.89 |
| | | KNN$_{50}$ | 63.78 | 80.00 | 57.30 | 71.70 | 76.34 | 79.67 | 85.88 | 68.38 | 70.82 | 74.94 | 94.19 | 81.45 | 91.80 | 81.09 |
| | | KNN$_{100}$ | 64.77 | 79.98 | 54.33 | 71.94 | 77.37 | 79.32 | 86.08 | 67.80 | 70.64 | 74.76 | 94.19 | 81.20 | 91.90 | 80.47 |
| | | *Pre-train on pure-text data* | | | | | | | | | | | | | | |
| | – | KNN$_{10}$ | 85.53 | 59.90 | 98.61 | 21.79 | 96.21 | 56.72 | 97.69 | 58.39 | 94.51 | 49.20 | 12.79 | 98.01 | 84.50 | 65.73 |
| | | KNN$_{20}$ | 85.45 | 59.27 | 98.73 | 21.19 | 96.21 | 55.63 | 97.90 | 57.05 | 94.57 | 48.28 | 12.79 | 97.91 | 86.10 | 63.57 |
| | | KNN$_{50}$ | 86.80 | 57.94 | 98.77 | 20.45 | 96.89 | 54.12 | 98.30 | 55.35 | 95.19 | 46.96 | 13.95 | 97.60 | 89.30 | 59.64 |
| | | KNN$_{100}$ | 88.47 | 56.71 | 98.81 | 19.97 | 96.81 | 52.89 | 98.18 | 53.93 | 95.57 | 45.88 | 13.95 | 97.38 | 91.10 | 55.17 |
| **Longformer**$_{4096}$ | 92.08 | *Pre-train on pure-text data→ fine-tune on RVL-CDIP (ID)* | | | | | | | | | | | | | | |
| | | MSP | 65.96 | 69.58 | 50.38 | 77.93 | 81.52 | 60.89 | 90.21 | 54.23 | 72.02 | 65.66 | 82.56 | 60.14 | 95.00 | 50.90 |
| | | MaxLogit | 62.19 | 87.35 | 44.64 | 89.79 | 79.97 | 78.84 | 88.39 | 68.08 | 68.80 | 81.02 | 80.23 | 84.19 | 94.30 | 77.36 |
| | | Energy | 61.27 | 87.35 | 43.61 | 89.81 | 79.13 | 78.85 | 88.15 | 68.08 | 68.04 | 81.02 | 80.23 | 84.19 | 94.30 | 77.37 |
| | | KNN$_{10}$ | 58.65 | 79.54 | 50.77 | 71.81 | 66.56 | 83.48 | 80.87 | 75.19 | 64.21 | 77.51 | 58.14 | 92.78 | 90.00 | 77.76 |
| | | KNN$_{20}$ | 57.81 | 79.43 | 51.40 | 71.72 | 67.00 | 83.35 | 81.15 | 74.86 | 64.34 | 77.34 | 58.14 | 92.57 | 89.70 | 77.12 |
| | | KNN$_{50}$ | 58.77 | 79.30 | 51.60 | 71.67 | 66.72 | 83.15 | 81.31 | 74.36 | 64.60 | 77.12 | 61.63 | 92.24 | 89.80 | 76.17 |
| | | KNN$_{100}$ | 61.39 | 79.16 | 52.75 | 71.61 | 67.84 | 82.93 | 81.76 | 73.91 | 65.94 | 76.90 | 62.79 | 91.99 | 89.80 | 75.29 |
| | | *Pre-train on pure-text data* | | | | | | | | | | | | | | |
| | – | KNN$_{10}$ | 99.40 | 47.83 | 100.00 | 27.75 | 98.28 | 47.03 | 93.20 | 60.40 | 97.72 | 45.75 | 46.51 | 93.85 | 100.00 | 63.64 |
| | | KNN$_{20}$ | 99.44 | 47.33 | 100.00 | 27.48 | 98.32 | 46.49 | 93.24 | 60.02 | 97.75 | 45.38 | 48.84 | 93.70 | 100.00 | 62.79 |
| | | KNN$_{50}$ | 99.44 | 46.33 | 100.00 | 27.23 | 98.40 | 45.85 | 93.41 | 60.05 | 97.81 | 44.86 | 51.16 | 93.51 | 100.00 | 61.55 |
| | | KNN$_{100}$ | 99.44 | 45.67 | 100.00 | 27.31 | 98.44 | 45.23 | 93.53 | 59.90 | 97.85 | 44.53 | 52.33 | 93.40 | 100.00 | 60.31 |
| **ResNet-50** | 87.80 | *Pre-train on ImageNet→ fine-tune on RVL-CDIP (ID)* | | | | | | | | | | | | | | |
| | | MSP | 70.58 | 85.35 | 55.29 | 89.88 | 64.29 | 86.54 | 71.15 | 85.58 | 65.33 | 86.84 | 54.79 | 91.70 | 77.20 | 84.67 |
| | | MaxLogit | 64.25 | 87.46 | 53.59 | 90.72 | 49.70 | 90.60 | 64.45 | 88.71 | 58.00 | 89.37 | 36.99 | 95.13 | 78.90 | 86.86 |
| | | Energy | 62.66 | 87.65 | 58.33 | 90.33 | 46.00 | 91.26 | 63.56 | 89.05 | 57.64 | 89.57 | 32.88 | 95.69 | 83.00 | 87.05 |
| | | KNN$_{10}$ | 90.99 | 79.37 | 56.36 | 90.64 | 72.41 | 86.20 | 89.17 | 81.74 | 77.23 | 84.49 | 2.74 | 99.32 | 39.70 | 93.70 |
| | | KNN$_{20}$ | 92.17 | 78.00 | 47.47 | 92.61 | 68.27 | 88.42 | 90.85 | 80.23 | 74.69 | 84.82 | 2.74 | 99.25 | 43.80 | 93.08 |
| | | KNN$_{50}$ | 94.32 | 75.96 | 28.44 | 94.49 | 65.65 | 89.27 | 92.78 | 77.91 | 70.30 | 84.41 | 1.37 | 98.97 | 49.70 | 92.09 |
| | | KNN$_{100}$ | 95.58 | 74.02 | 27.21 | 95.07 | 60.44 | 89.78 | 94.22 | 75.63 | 69.36 | 83.62 | 2.74 | 98.67 | 53.80 | 91.10 |
| | | *Pre-train on ImageNet* | | | | | | | | | | | | | | |
| | – | KNN$_{10}$ | 98.46 | 42.21 | 77.29 | 81.41 | 27.87 | 91.16 | 99.08 | 43.47 | 75.68 | 64.56 | 80.82 | 89.98 | 12.30 | 98.17 |
| | | KNN$_{20}$ | 98.66 | 41.00 | 76.78 | 81.70 | 29.22 | 92.27 | 99.08 | 42.29 | 75.94 | 64.32 | 83.56 | 89.30 | 14.10 | 97.97 |
| | | KNN$_{50}$ | 98.58 | 39.53 | 76.58 | 81.81 | 31.01 | 92.05 | 99.12 | 40.80 | 76.32 | 63.55 | 83.56 | 88.51 | 16.30 | 97.61 |
| | | KNN$_{100}$ | 98.62 | 38.62 | 77.13 | 81.49 | 32.64 | 91.84 | 99.12 | 39.86 | 76.88 | 62.95 | 83.56 | 87.80 | 19.50 | 97.23 |
| **Swin**Base | 92.42 | *Pre-train on ImageNet→ fine-tune on RVL-CDIP (ID)* | | | | | | | | | | | | | | |
| | | MSP | 63.96 | 87.03 | 65.21 | 88.15 | 73.56 | 79.72 | 61.40 | 88.46 | 66.03 | 85.84 | 84.93 | 74.34 | 49.60 | 92.49 |
| | | MaxLogit | 56.49 | 90.22 | 75.36 | 87.00 | 72.64 | 84.26 | 44.22 | 93.01 | 62.18 | 88.62 | 72.60 | 84.16 | 29.10 | 95.70 |
| | | Energy | 57.43 | 90.11 | 77.01 | 86.60 | 73.44 | 84.17 | 43.78 | 93.06 | 62.92 | 88.48 | 73.97 | 84.25 | 28.00 | 95.69 |
| | | KNN$_{10}$ | 60.27 | 90.12 | 66.90 | 90.76 | 49.66 | 89.15 | 47.67 | 92.67 | 56.12 | 90.68 | 42.47 | 94.28 | 7.20 | 98.56 |
| | | KNN$_{20}$ | 61.32 | 90.01 | 61.37 | 91.31 | 48.83 | 90.33 | 49.00 | 92.52 | 55.13 | 91.04 | 30.14 | 95.56 | 8.80 | 98.33 |
| | | KNN$_{50}$ | 62.22 | 89.78 | 56.44 | 91.56 | 50.34 | 89.55 | 48.52 | 92.30 | 54.38 | 90.80 | 26.03 | 95.72 | 11.80 | 97.97 |
| | | KNN$_{100}$ | 62.62 | 89.60 | 54.98 | 91.85 | 50.70 | 88.93 | 47.63 | 92.18 | 53.98 | 90.64 | 30.14 | 95.54 | 13.90 | 97.66 |
| | | *Pre-train on ImageNet* | | | | | | | | | | | | | | |
| | – | KNN$_{10}$ | 99.15 | 45.57 | 86.02 | 79.44 | 32.45 | 90.98 | 99.52 | 46.20 | 79.28 | 65.55 | 24.66 | 96.24 | 0.40 | 99.78 |
| | | KNN$_{20}$ | 99.19 | 44.11 | 86.89 | 80.35 | 33.48 | 92.19 | 99.60 | 44.79 | 79.79 | 65.36 | 27.40 | 95.62 | 0.50 | 99.73 |
| | | KNN$_{50}$ | 99.23 | 42.39 | 87.99 | 81.66 | 36.78 | 91.59 | 99.60 | 43.07 | 80.90 | 64.68 | 43.84 | 94.57 | 0.80 | 99.63 |
| | | KNN$_{100}$ | 99.19 | 41.46 | 89.02 | 82.63 | 40.60 | 91.05 | 99.60 | 42.14 | 82.10 | 64.32 | 52.05 | 93.49 | 1.20 | 99.53 |
| **ViT**Base | 91.03 | *Pre-train on ImageNet→ fine-tune on RVL-CDIP (ID)* | | | | | | | | | | | | | | |
| | | MSP | 69.68 | 86.81 | 69.67 | 87.88 | 72.25 | 80.78 | 69.38 | 86.61 | 70.24 | 85.52 | 67.12 | 85.97 | 58.50 | 91.47 |
| | | MaxLogit | 63.35 | 89.20 | 68.40 | 88.58 | 69.58 | 84.38 | 61.08 | 89.94 | 65.60 | 88.02 | 57.53 | 89.41 | 48.40 | 93.04 |
| | | Energy | 62.22 | 89.21 | 70.34 | 88.43 | 70.26 | 84.37 | 60.75 | 90.03 | 65.89 | 88.01 | 58.90 | 89.47 | 49.70 | 93.03 |
| | | KNN$_{10}$ | 68.10 | 88.99 | 54.90 | 92.30 | 53.44 | 88.05 | 58.19 | 90.17 | 58.66 | 90.17 | 38.36 | 95.02 | 22.90 | 96.71 |
| | | KNN$_{20}$ | 67.61 | 88.95 | 49.01 | 92.85 | 51.53 | 89.25 | 58.59 | 91.16 | 56.68 | 90.55 | 41.10 | 94.47 | 25.40 | 96.35 |
| | | KNN$_{50}$ | 67.29 | 88.91 | 42.54 | 93.15 | 53.96 | 88.43 | 58.75 | 90.88 | 55.64 | 90.34 | 42.47 | 93.60 | 29.90 | 95.78 |
| | | KNN$_{100}$ | 66.19 | 88.90 | 43.80 | 93.19 | 55.71 | 87.73 | 59.11 | 90.64 | 56.20 | 90.12 | 45.21 | 92.86 | 34.90 | 95.27 |
| | | *Pre-train on ImageNet* | | | | | | | | | | | | | | |
| | – | KNN$_{10}$ | 98.90 | 41.98 | 90.96 | 77.15 | 34.87 | 90.69 | 99.40 | 41.01 | 81.03 | 62.76 | 54.79 | 94.27 | 10.80 | 98.47 |
| | | KNN$_{20}$ | 98.94 | 40.54 | 91.67 | 77.20 | 36.82 | 91.71 | 99.44 | 39.85 | 81.72 | 62.32 | 64.38 | 93.57 | 12.70 | 98.25 |
| | | KNN$_{50}$ | 99.07 | 38.75 | 92.61 | 76.99 | 40.00 | 91.17 | 99.52 | 38.14 | 82.80 | 61.26 | 75.34 | 92.47 | 15.90 | 97.87 |
| | | KNN$_{100}$ | 99.11 | 37.43 | 93.25 | 76.56 | 43.38 | 90.68 | 99.56 | 36.93 | 83.82 | 60.40 | 82.19 | 91.52 | 18.90 | 97.49 |

Table 12: OOD detection performance for document classification. All models are pre-trained on IIT-CDIP. For LayoutLM models, we adopt the checkpoints from the Huggingface model hub. For UDoc, we pre-train the model on our side. All models are fine-tuned on RVL-CDIP ID data.

| | ID Acc | Method | Sci. Report FPR95 | Sci. Report AUROC | Presentation FPR95 | Presentation AUROC | Form FPR95 | Form AUROC | Letter FPR95 | Letter AUROC | Average FPR95 | Average AUROC | Sci. Poster FPR95 | Sci. Poster AUROC | Receipt FPR95 | Receipt AUROC |
|---|---|---|---|---|---|---|---|---|---|---|---|---|---|---|---|---|
| | | | | | | | | OOD Dataset (In-domain) | | | | | | OOD Dataset (Out-domain) | | |
| **LayoutLMv1**Base | 97.28 | MSP | 47.48 | 74.91 | 59.74 | 68.72 | 66.40 | 65.36 | 58.89 | 69.12 | 58.13 | 69.53 | 43.02 | 77.15 | 72.40 | 62.40 |
| | | MaxLogit | 27.06 | 92.38 | 37.97 | 91.52 | 45.65 | 88.36 | 35.92 | 91.22 | 36.65 | 90.87 | 24.42 | 94.96 | 57.30 | 86.70 |
| | | Energy | 27.06 | 92.40 | 37.97 | 91.54 | 45.65 | 88.36 | 35.92 | 91.23 | 36.65 | 90.88 | 24.42 | 94.97 | 57.30 | 86.70 |
| | | KNN$_{10}$ | 20.82 | 96.09 | 35.32 | 93.82 | 40.06 | 91.34 | 28.65 | 94.80 | 31.21 | 94.01 | 17.44 | 97.00 | 49.80 | 93.92 |
| | | KNN$_{20}$ | 21.74 | 95.93 | 36.20 | 93.77 | 41.42 | 91.12 | 30.44 | 94.61 | 32.45 | 93.86 | 17.44 | 96.82 | 51.70 | 93.73 |
| | | KNN$_{50}$ | 24.34 | 95.56 | 38.25 | 93.41 | 43.93 | 90.69 | 33.64 | 94.19 | 35.04 | 93.46 | 23.26 | 96.44 | 53.80 | 93.70 |
| | | KNN$_{100}$ | 25.54 | 95.30 | 39.13 | 93.20 | 45.17 | 90.35 | 34.78 | 93.99 | 36.16 | 93.21 | 25.58 | 96.24 | 54.70 | 93.45 |
| **LayoutLMv3** | 97.81 | MSP | 56.16 | 70.81 | 64.49 | 67.17 | 67.16 | 65.30 | 58.60 | 69.58 | 61.34 | 68.22 | 52.33 | 72.70 | 43.60 | 77.10 |
| | | MaxLogit | 30.70 | 89.17 | 40.42 | 88.18 | 42.98 | 84.09 | 33.12 | 88.22 | 36.80 | 87.42 | 19.77 | 94.50 | 11.70 | 97.02 |
| | | Energy | 30.70 | 89.18 | 40.42 | 88.18 | 42.98 | 84.10 | 33.12 | 88.23 | 36.80 | 87.42 | 19.77 | 94.51 | 11.70 | 97.03 |
| | | KNN$_{10}$ | 21.74 | 95.03 | 35.68 | 93.38 | 32.88 | 91.86 | 18.51 | 96.26 | 27.20 | 94.13 | 11.63 | 97.58 | 8.90 | 97.97 |
| | | KNN$_{20}$ | 22.74 | 94.90 | 36.56 | 93.20 | 33.96 | 91.66 | 19.64 | 96.15 | 28.22 | 93.98 | 12.79 | 97.44 | 10.00 | 97.89 |
| | | KNN$_{50}$ | 24.62 | 94.62 | 38.37 | 92.71 | 35.83 | 91.38 | 21.63 | 95.93 | 30.11 | 93.66 | 13.95 | 97.20 | 10.70 | 97.72 |
| | | KNN$_{100}$ | 25.22 | 94.38 | 39.29 | 92.32 | 36.55 | 91.09 | 22.48 | 95.79 | 30.88 | 93.40 | 16.28 | 97.04 | 11.80 | 97.59 |
| **UDoc**ResNet50 | 97.36 | MSP | 66.13 | 65.73 | 69.43 | 64.09 | 71.03 | 63.28 | 71.06 | 63.25 | 69.41 | 64.09 | 40.70 | 78.47 | 39.80 | 78.99 |
| | | MaxLogit | 45.96 | 82.12 | 47.21 | 86.39 | 49.64 | 83.16 | 49.59 | 83.13 | 48.10 | 83.70 | 2.33 | 98.57 | 4.00 | 98.34 |
| | | Energy | 45.96 | 82.12 | 47.21 | 86.40 | 49.64 | 83.16 | 49.59 | 83.13 | 48.10 | 83.70 | 2.33 | 98.60 | 4.00 | 98.36 |
| | | KNN$_{10}$ | 30.02 | 94.47 | 41.22 | 88.66 | 41.90 | 90.99 | 36.65 | 93.48 | 37.45 | 91.90 | 1.16 | 99.13 | 5.50 | 98.42 |
| | | KNN$_{20}$ | 31.10 | 94.36 | 41.98 | 88.44 | 42.10 | 90.90 | 38.03 | 93.35 | 38.30 | 91.76 | 1.16 | 99.04 | 6.90 | 98.32 |
| | | KNN$_{50}$ | 33.95 | 94.07 | 43.35 | 87.89 | 44.01 | 90.72 | 40.71 | 93.06 | 40.51 | 91.43 | 1.16 | 98.84 | 7.40 | 98.26 |
| | | KNN$_{100}$ | 34.83 | 93.84 | 43.75 | 87.51 | 45.01 | 90.61 | 41.96 | 92.90 | 41.39 | 91.22 | 1.16 | 98.72 | 8.30 | 98.16 |

