# OpenReview forum: "A Critical Analysis of Document Out-of-Distribution Detection"
_EMNLP/2023/Conference — EMNLP 2023 Findings_

### Official Review · Reviewer_os2G · 2023-07-27

**Soundness:** 4

**Excitement:**

4: Strong: This paper deepens the understanding of some phenomenon or lowers the barriers to an existing research direction.

**Missing References:**

[1] Hao Lang, Yinhe Zheng, Yixuan Li, Jian Sun, Fei Huang, and Yongbin Li. 2023. A survey on out-of-distribution detection in nlp.

**Paper Topic And Main Contributions:**

This paper presents a comprehensive analysis of out-of-distribution (OOD) detection for document understanding models. The authors investigate the effects of model modality, pre-training, and fine-tuning on OOD detection performance. It finds that spatial information is critical for document OOD detection and proposes an add-on spatial-aware adapter to improve OOD detection performance.

**Questions For The Authors:**

A. Given the wealth of world knowledge encoded in large language models (LLMs), is it possible to integrate LLM knowledge and spatial-awareness for better document understanding?

**Reasons To Accept:**

1. This paper provides comprehensive and in-depth analysis for document OOD detection, which is supported by extensive experimental results.
2. An add-on module is designed for transformer-based models, which has practical implications.

**Reasons To Reject:**

1. It would be better to elaborate on the difference between In-domain OOD and Out-domain OOD, and the motivation of ID/OOD dataset selection.
2. The authors mainly focus on the analysis of different aspects in document OOD detection, some are intuitive. Introducing more ID tasks (e.g., document-based question answering) other than classification could further enhance the understanding.

**Reproducibility:**

4: Could mostly reproduce the results, but there may be some variation because of sample variance or minor variations in their interpretation of the protocol or method.

**Reviewer Confidence:**

4: Quite sure. I tried to check the important points carefully. It's unlikely, though conceivable, that I missed something that should affect my ratings.

---

> ### Author Rebuttal · Authors · 2023-08-28
>
> We sincerely appreciate your insightful comments and support of our work! We address concerns, questions, and comments below in detail.
>
>
> ### Q: The difference between In-domain and Out-domain OOD, and the motivation of ID/OOD dataset selection
>
>
> We highlight the key differences between in-domain and out-domain OOD as follows:
>
> **(1)** Visually, out-domain OOD data are semantically distinct from ID classes, the in-domain OOD datasets resemble ID data in both layout and contents. As shown in **Fig 3**, out-domain OOD data such as scientific posters and receipts are colorful, while in-domain OOD data (letter, form, presentation, and scientific report) are black and white, more resembling ID data.
>
> **(2)** In terms of feature representations, we visualize the pre-trained and fine-tuned features for ID, in-domain OOD, and out-domain OOD data (w. ViT and RoBERTa) in **Fig 10**. We can also see that while out-domain OOD are far from ID data, the four in-domain OOD categories are close to ID clusters in the feature space.
>
> **(3)** Quantitatively, we measure the distance between each OOD dataset and ID data via Optimal Transport (https://github.com/microsoft/otdd) and visualize the results in **Fig 9a** (Page 13). We can see that the distance between out-domain OOD and ID data is significantly (around 10x) larger than in-domain OOD vs. ID data.
>
>
> **Motivation of ID/OOD dataset selection:** We curate both the ID dataset and in-domain OOD datasets from the same RVL-CDIP dataset. This construction follows practical considerations where OOD categories are the ones multiple models (ViT, RoBERTa, UDoc, LayoutLMv1, and LayoutLMv3, etc.) perform poorly on, as shown in **Fig 2 (b)**.  Therefore, we may expect the models to detect such inputs as OOD instead of assigning a specific ID class with low confidence. Additionally, we also incorporated two other OOD category selection criteria in the Appendix (Appendix A.1, Table 10, and Table 11).
>
> On the other hand, we construct out-domain OOD datasets to mimic scenarios in the open world where the test inputs can have significantly different color schemes and layouts compared to ID samples. This results in a substantial distinction between ID and out-domain OOD datasets in the feature space (see **Fig. 9** in Appendix).
>
> ### Q: Is it possible to integrate LLM knowledge and spatial awareness for better document understanding?
>
> Thanks for the insightful comments. LLMs excel in comprehending textual content owing to their extensive exposure to diverse text data. However, they often lack spatial awareness, a critical element in understanding structured documents. As demonstrated in this paper, the incorporation of a spatial-aware adapter significantly enhances both in-domain and out-of-domain OOD performance in comparison to text-only models. Exploring effective algorithms and modules that can extract structured knowledge from LLMs to augment document understanding is a promising avenue for future research.
>
> However, concurrently, integrating spatial-aware information into generative models (e.g., LLama2/GPT/...) would likely require a different approach than what is presented in our work, such as the Spatial-aware RoBERTa with Masked Language Modeling. Existing LLMs are predominantly built upon Causal Language Modeling, which implies a sequence-based order among words. Yet, transitioning from a 1D sequence to a 2D spatial layout introduces a more complex order that extends beyond a simple left-to-right structure. Therefore, it's essential to consider not only the forward prediction but also incorporate spatial-aware Causal Language Modeling, particularly for documents with a 2D layout.

---

### Official Review · Reviewer_Rxki · 2023-07-30

**Typos Grammar Style And Presentation Improvements:** No
**Soundness:** 3

**Excitement:**

3: Ambivalent: It has merits (e.g., it reports state-of-the-art results, the idea is nice), but there are key weaknesses (e.g., it describes incremental work), and it can significantly benefit from another round of revision. However, I won't object to accepting it if my co-reviewers champion it.

**Missing References:**

No

**Paper Topic And Main Contributions:**

Paper summary
Document OOD detection is a practical direction. However, previous works on document OOD detection either only using single-modality rather than multi-modality information, which lost some of the information from the document, or using too large models, which leads to high computational costs.  This paper first provides an extensive study on the impacts of pre-training, fine-tuning, scoring methods and model-modality on the performance of document OOD detection. They highlight several findings, such as pre-training helps, fine-tuning helps, distance-based OOD scoring method outperforms logit-based scoring method and spacial information is important. Based on these observations, they propose a spatial-aware adapter on Roberta model, which can outperform the vanilla Roberta model and is on par with multi-modal document OOD detection methods.

Presentation: Good
This paper is well-written and easy to follow.

Novelty: Fair
For the method, the authors proposed an adapter to encode the spatial information, which is new but intuitive. The paper is more like an engineering paper.
For the findings, the authors provide several findings, but most of the findings are intuitive.

Soundness: Fair
The technical part looks sound to me, but I have a concern about the evaluation part. The authors construct the ID and OOD data on their own to evaluate their method. I understand it may be the common practice in the OOD detection field but is it possible that the ID data and OOD data they used are too far so that using spatial information + Roberta can be on pair with complicated multi-modal method?

Significance: Fair
The proposed method can outperform Roberta, a pure-text model, and be on par with the multi-modal methods.


**Questions For The Authors:**

1. Is it possible that the ID data and OOD data you used are too far so that using spatial information + Roberta can be on pair with the complicated multi-modal method?
2. If the vision model is good enough, I believe there is no need to do the OCR and then extract the textual information. For example, in GPT4 technical report, GPT4 can easily understand the image input with both picture and text. Why using the image feature alone can not do well in document OOD detection? Your Resnet and VIT baseline seems weak to me. Have you tried other advanced vision pre-training models?
3. In the era of the large model, the training set is massive online data and it is hard to tell which is ID and which is OOD, what is the future direction of OOD detection?

**Reasons To Accept:**

1. This paper is well-written and easy to follow.
2. The authors provide a comprehensive empirical study on document OOD detection, with extensive analysis and experimental results., which could provide useful information to people in this field.

**Reasons To Reject:**

1. This paper is too empirical and some of the findings are intuitive, although it is acceptable for empirical NLP conferences.
2. The evaluation of their proposed adapter is not convincing. The baseline method (such as Roberta) is not strong, since it is intuitive that adding multi-modal information can help. And as I mentioned before, adding spatial information can be on par with the multimodal method could be due to the ID and OOD data being too far away from each other.

**Reproducibility:**

4: Could mostly reproduce the results, but there may be some variation because of sample variance or minor variations in their interpretation of the protocol or method.

**Reviewer Confidence:**

4: Quite sure. I tried to check the important points carefully. It's unlikely, though conceivable, that I missed something that should affect my ratings.

---

> ### Author Rebuttal · Authors · 2023-08-28
>
> Thanks for your insightful comments and valuable suggestions! We address concerns, questions, and comments below in detail.
>
>
> ### Q: Are OOD data far from ID data?
>
> We acknowledge that currently there exists no standard benchmark for document OOD detection, given the scarcity of document data, especially for cross-domain document classification datasets. RVL-CDIP stands out as a prominent dataset, and notably, its image data consists of black-and-white or scanned documents.
>
> In this work, we consider both in-domain OOD and out-domain OOD data for RVL-CDIP. While the out-domain OOD datasets are semantically distinct from ID categories (e.g., colorful vs. black-and-white), the in-domain OOD data resembles ID data in both layout and contents (see Figure 3, top row vs. bottom row). We provide further qualitative and quantitative evidence below.
>
> **(1)** Qualitatively, we visualize the pre-trained and fine-tuned features for ID, in-domain OOD, and out-domain OOD data (w. ViT and RoBERTa) in **Fig 10**. We can clearly see that while out-domain OOD are far from ID data, the four in-domain OOD categories are close to ID clusters in the feature space.
>
> **(2)** Quantitatively, we measure the distance between each OOD dataset and ID data via Optimal Transport (https://github.com/microsoft/otdd) and visualize the results in **Fig 9a** (page 13). We can see that the distance between out-domain OOD and ID data is significantly (around 10x) larger than in-domain OOD vs. ID data.
>
> The construction of in-domain OOD categories follows practical considerations where OOD categories are the ones multiple models (ViT, RoBERTa, UDoc, LayoutLMv1, and LayoutLMv3, etc.) perform poorly on. Therefore, we may expect the models to detect such inputs as OOD instead of assigning a specific ID class with low confidence. (L332-L335 in page 5)
>
> Moreover, we consider two additional in-domain OOD dataset selection schemes: (1) select 4 classes out of 16 from RVL-CDIP as OOD where the model performs well; (2) randomly select 4 classes as OOD. The OT distances are visualized in Fig 9b and Fig 9c. We can see that the distance between ID and in-domain OOD is similar to the original scheme (Fig 9a). This suggests that most in-domain OOD categories are not far from ID data.
>
> While this paper represents an initial endeavor, we hope that our work will serve as a stepping stone towards constructing more comprehensive and diverse OOD benchmarks in the document domain, akin to those available in the NLP and natural image domain.
>
> ### Q: Experiment with advanced vision pre-training models
>
> Great question! In fact, we have explored recent well-established image-based pretraining methods such as MIM and MAE to enhance our image encoder. However, our exploration revealed that the impact of natural image pretraining methods on document image data is comparatively modest. We outline several reasons for this outcome:
>
> **(1)** Document images possess sparsity, which implies that not every pixel holds meaningful information. For instance, in scientific papers or resumes, the pixels containing texts constitute a small fraction, while most pixels represent the white background. Consequently, traditional methods like MIM or MAE with random masking struggle to adapt effectively to document images.
>
> **(2)** Document images are inherently multimodal, differing from the conventional image-caption pairs. The multimodal nature of document data signifies that text and image modalities coexist within the same entity. For instance, both textual contents and figures may exist in a single paragraph. This introduces challenges in designing pretraining tasks.
>
> Notably, we have included a recent work DiT [1], an image-based pretraining method that draws inspiration from image tokenization and MIM techniques. We acquired the DiT base model from HuggingFace and performed fine-tuning on RVL-CDIP (ID), achieving an accuracy of 96.52%. (In comparison, the Swin-based model achieved an accuracy of 95.74%, as shown in **Table 9** in the Appendix.) The OOD detection results are also provided below. Comparing the table below with Appendix **Table 12**, we can see that DiT achieves better results (vs. ViT/ResNet) with the KNN-based OOD score. This also highlights the significance of document-specific pre-training in OOD detection.
>
> | OOD Score      | Letter   |   --  |  --    | H.W. |  -- | --  | Adv. |  -- | --  | Memo | --  | --  | Average | -- | --  | Sci. Poster   |   --  |   --   | Receipt | --  | --  |
> |-------------|---------|---------|---------|---------|---------|--------|---------|---------|--------|---------|---------|--------|--------|----------|--------|---------|---------|---------|---------|---------|---------|
> | | FPR95   | AUROC   | AUPR    | FPR95 | AUROC | AUPR | FPR95 | AUROC | AUPR | FPR95 | AUROC | AUPR | FPR95 | AUROC | AUPR | FPR95   | AUROC   | AUPR    | FPR95 | AUROC | AUPR |
> | Energy      | 43.71   | 90.92   | 99.06   | 38.69   | 93.28   | 99.38  | 67.92   | 85.85   | 98.38  | 29.46   | 94.83   | 99.53  | 44.94  | 91.22    | 99.09  | 47.95   | 92.91   | 99.98   | 7.70    | 97.68   | 99.92   |
> | Maha   | 30.06   | 95.40   | 99.59   | 35.15   | 95.15   | 99.59  | 53.67   | 90.97   | 99.13  | 74.88   | 90.33   | 99.13  | 48.44  | 92.96    | 99.36  | 50.68   | 92.89   | 99.98   | 6.80    | 97.74   | 99.92   |
> | KNN10      | 27.22   | 95.65   | 99.55   | 30.86   | 95.36   | 99.54  | 41.06   | 92.84   | 99.25  | 47.24   | 93.58   | 99.40  | 36.60  | 94.36    | 99.44  | 63.01   | 70.49   | 99.78   | 8.50    | 97.72   | 99.90   |
> | KNN20      | 25.66   | 95.65   | 99.56   | 31.30   | 95.43   | 99.58  | 41.34   | 92.68   | 99.24  | 41.15   | 94.08   | 99.45  | 34.86  | 94.46    | 99.46  | 16.44   | 97.79   | 99.99   | 4.00    | 98.95   | 99.96   |
> | KNN50      | 25.46   | 95.71   | 99.61   | 30.37   | 95.57   | 99.63  | 42.94   | 92.49   | 99.29  | 35.23   | 94.45   | 99.48  | 33.50  | 94.56    | 99.50  | 16.44   | 97.72   | 99.99   | 5.60    | 98.83   | 99.96   |
> | KNN100     | 25.90   | 95.62   | 99.60   | 29.33   | 95.66   | 99.63  | 44.73   | 92.21   | 99.26  | 35.55   | 94.49   | 99.48  | 33.88  | 94.50    | 99.49  | 19.18   | 97.51   | 99.99   | 7.80    | 98.59   | 99.95   |
>
> Remarks: *Average* denotes the average for in-domain OOD datasets; KNN10 (20, 50, 100) denotes using the KNN+ score with K = 10 (K = 20, 50, 100); Adv. stands for Advertisement and H.W. stands for Handwritten.
>
> [1] Li et al., DiT: Self-supervised Pre-training for Document Image Transformer, ACM Multimedia 2022
>
> ### Q: The future direction of OOD detection in the era of large models
>
> Indeed, in the era of large-scale pre-training, it is challenging to identify OOD categories not semantically related to the pre-training data. Therefore, in this work and previous works on OOD detection with pre-trained models, OOD is defined with respect to the downstream dataset (ID), instead of the pre-training data which is often hard to characterize (L197 - L200, page 3).  While our work aims to establish an in-depth and extensive study of document OOD detection, we believe large models open up new opportunities for OOD detection and highlight several promising avenues for future research:
>
> **(1)** Develop computationally efficient adaptation (fine-tuning) algorithms to safeguard pre-trained large models against OOD data for a wider range of downstream tasks. In this paper, we notice that retraining large models is both time-consuming and costly. Additionally, OOD scenarios are often task-specific. Therefore, similar to our spatial-aware adapter, exploring how to introduce additional modality information, such as spatial data, into well-pretrained models such as Llama2 [1], while minimizing resource consumption, poses an intriguing question.
>
>
> **(2)** Provide theoretical understanding on how pre-training impacts downstream OOD detection. Despite extensive empirical studies presented in our work, it remains underexplored how various aspects of pre-training (e.g., pre-training objectives, the diversity of pre-training data) impact downstream OOD detection theoretically. A novel theoretical framework would be helpful to further guide the design of practical algorithms.
>
> **(3)** Developing new OOD benchmarks become increasingly vital for pre-trained large models. Merely having large models doesn't ensure they've encountered the diverse range of data users might interact with, especially in specialized areas such as documents. Notably, datasets like Common Crawl may have inherent biases from the crawling process, limiting their coverage of diverse documents (PDFs/images/...). Additionally, documents are created by humans, resulting in long-tailed distributions in the pre-training dataset. Even advanced models like GPT-4 can't be presumed to be adept at handling every document type. Therefore, addressing OOD reliability for large models necessitates revisiting data collection, aiming to construct a more comprehensive and effective OOD benchmark for evaluating the OOD capabilities of pre-trained large models.
>
>
> [1] Touvron et al., Llama 2: Open Foundation and Fine-Tuned Chat Models, Arxiv 2023

---

### Official Review · Reviewer_M9aT · 2023-08-05

**Typos Grammar Style And Presentation Improvements:** The Figure 2b is very difficult to fo…
**Soundness:** 3

**Excitement:**

4: Strong: This paper deepens the understanding of some phenomenon or lowers the barriers to an existing research direction.

**Missing References:**

Kimin Lee et. al. A simple unified framework for detecting out-of-distribution samples and adversarial attacks. In NEURIPS 2018
Podolskiy, A.V., et al. Revisiting Mahalanobis Distance for Transformer-Based Out-of-Domain Detection. in AAAI 2021
Yiyou Sun et al. Out-of-distribution detection with deep nearest neighbors. In ICML 2022

**Paper Topic And Main Contributions:**

The paper investigates how spatial information influence on for document out-of-distribution(OOD) detection. To support the claim the authors suggest using the spatial-aware adapter. The extensive experiments on RVL-CDIP (with OOD samples added from NJU-Fudan and CORD datasets) datasets are provided in different setups. More precisely the paper compares vision-only, text-only, text+layout, and vision+text+layout models. The suggested adapter approach with the RoBERTa model overperforms the baseline RoBERTa model and matches the performance of LayoutLM.

This work is well-written and effectively conveys the main idea. It gives an overview and comparison of existing approaches for document OOD detection. The suggested adapter approach looks promising for incorporating spatial information inside the RoBERTa model. This paper could be beneficial for the researchers in narrow field of OOD for documents.

However, there are several concerns I would like to discuss.
1) First, it is not clear how the selection of baselines is done and why the SOTA approach DocFormer [1] for RVL-CDIP is missed for comparison. To prove that the suggested approach is comparable with LayoutLM v3 it is better to provide a side-by-side comparison. Also, I want to see more discussions on which setup of the proposed adapter approach is better than existing multi-modal approaches.
2) It is not quite obvious that the suggested zero-shot setup pre-training on the IIT-CDIP for vision models is really zero-shot. How can we guarantee that the four OOD classes (letter, form, scientific report, and presentation) are not presented on the documents in implicit form? Again it is not clear why multimodal models are omitted from the comparison
3) In the field of OOD it is common to use AUPR_OOD in case of imbalance classes and FPR with the decision threshold to prove the method's ability to keep good in-domain quality with a given threshold. It will better reflect the OOD ability of the methods.
4) In the recent OOD works the distance-based methods with Mahalanobis distance show great performance both for text and vision tasks, but this comparison is not provided. See [2],[3],[4]

Strengths:

1) The suggested adapter approach looks promising and seems beneficial in some scenarios
2) The extensive analysis of the OOD method for the documents scenario is provided.

Weaknesses:

1) Baseline selection (missing DocFormer from comparison) and non-obvious comparison of suggested approach with others.
2) The zero-shot setup is not clear for vision models and should be clarified.
3) From the OOD point-of-view there is no evidence that the suggested approach is better, better to provide additional metrics to prove it.
4) In the recent OOD works the distance-based methods with Mahalanobis distance show great performance both for text and vision tasks, but this comparison is not provided. See [2],[3],[4]

 [1] Appalaraju, Srikar, et al. "Docformer: End-to-end transformer for document understanding." Proceedings of the IEEE/CVF international conference on computer vision. 2021.
[2] Kimin Lee et. al. A simple unified framework for detecting out-of-distribution samples and adversarial attacks. In NEURIPS 2018
[3] Podolskiy, A.V., et al. Revisiting Mahalanobis Distance for Transformer-Based Out-of-Domain Detection. in AAAI 2021
[4]  Yiyou Sun et al. Out-of-distribution detection with deep nearest neighbors. In ICML 2022

**Questions For The Authors:**

A. Why the SOTA approach DocFormer [1] for RVL-CDIP is missed for comparison?
B. What is the benefits of suggested approach with adapter compared to other multimodal approaches, i.e  LayoutLM v3?
C. It is not quite obvious that the suggested zero-shot setup pre-training on the IIT-CDIP for vision models is really zero-shot. How can we guarantee that the four OOD classes (letter, form, scientific report, and presentation) are not presented on the documents in implicit form? D. D. It is not clear why multimodal models are omitted from the comparison in zero-shot setup?

**Reasons To Accept:**

1) The suggested adapter approach looks promising and seems beneficial in some scenarios
2) The extensive analysis of the OOD method for the documents scenario is provided.

**Reasons To Reject:**

1) Baseline selection (missing DocFormer from comparison) and non-obvious comparison of suggested approach with others.
2) The zero-shot setup is not clear for vision models and should be clarified.
3) From the OOD point-of-view there is no evidence that the suggested approach is better, better to provide additional metrics to prove it.
4) In the recent OOD works the distance-based methods with Mahalanobis distance show great performance both for text and vision tasks, but this comparison is not provided. See [2],[3],[4]

**Reproducibility:**

4: Could mostly reproduce the results, but there may be some variation because of sample variance or minor variations in their interpretation of the protocol or method.

**Reviewer Confidence:**

4: Quite sure. I tried to check the important points carefully. It's unlikely, though conceivable, that I missed something that should affect my ratings.

---

> ### Author Rebuttal · Authors · 2023-08-28
>
> Thanks for your insightful comments and valuable suggestions! We address concerns, questions, and comments below in detail.
>
>
> ### Q: Baseline model selection
> In this work, we categorize recent document pre-training models into four categories (see **L271-L277** in Page 4 and Appendix **A.2**): **(1)** Vision-only models, which exclusively leverage visual features; **(2)** Text-only models, relying solely on textual features; **(3)** Text+Layout models, which combine both visual and textual features; **(4)** Vision+Text+Layout models, integrating supplementary spatial information. For each category, we have selected several representative models to examine the influence of input modality on various OOD detection tasks. Specifically, we have considered ResNet-50, Swin Transformer, and ViT for category **(1)**; RoBERTa and LongFormer for category **(2)**; LayoutLMv1, Spatial-RoBERTa (developed by us) for category **(3)**; and LayoutLMv3 and UDoc for category **(4)**.
>
> We acknowledge that, due to the extensive range of experiments, not every model in each category has been included. We also understand the importance of impactful models like DocFormer. Notably, its model structure shares similarities with LayoutLMv1/v2, both employing single-stream models, in contrast to the two-stream models (e.g., SelfDoc/UDoc). Relevant discussions on DocFormer can be found in **Section 2.1 (L145-L148)** within the related works.
>
> Nevertheless, we have tried our best to reproduce DocFormer's results. However, it came to our attention that *the official implementation by the authors was not made available*. The closest available codebase we found was a third-party implementation [A]. Despite our endeavors using their codebase and checkpoints, regrettably, we were unable to replicate the results reported by DocFormer's authors in their paper within the limited timeframe for this response. After a thorough examination of the codebase and the original paper's descriptions, we identified discrepancies in training configurations. Specifically, the codebase is based on a subset from IDL (https://github.com/furkanbiten/idl_data), while the original DocFormer used a 5M subset from IIT-CDIP. Most notably, there exist differences in the pre-training tasks. The repository only contains Masked Language Modeling (MLM), while the original paper considers three pre-training tasks: Multi-Modal MLM, Learn To Reconstruct, and Text Describes Image [B]. Despite our earnest attempts to reproduce the original implementation within the rebuttal phase, considering the time constraints of extensive pre-training on IIT-CDIP, we plan to incorporate DocFormer in our subsequent manuscript revision.
>
> Moreover, following your advice to incorporate more baselines, we have added **DiT** [C] (w. pre-trained checkpoint from HuggingFace: https://huggingface.co/microsoft/dit-base), a self-supervised pre-trained document image transformer model that employs a large-scale dataset of unlabeled document images. We report the OOD detection performance below, after finetuning the model on RVL-CDIP (ID):
>
> | OOD Score      | Letter   |   --  |  --    | H.W. |  -- | --  | Adv. |  -- | --  | Memo | --  | --  | Average | -- | --  | Sci. Poster   |   --  |   --   | Receipt | --  | --  |
> |-------------|---------|---------|---------|---------|---------|--------|---------|---------|--------|---------|---------|--------|--------|----------|--------|---------|---------|---------|---------|---------|---------|
> | | FPR95   | AUROC   | AUPR    | FPR95 | AUROC | AUPR | FPR95 | AUROC | AUPR | FPR95 | AUROC | AUPR | FPR95 | AUROC | AUPR | FPR95   | AUROC   | AUPR    | FPR95 | AUROC | AUPR |
> | Energy      | 43.71   | 90.92   | 99.06   | 38.69   | 93.28   | 99.38  | 67.92   | 85.85   | 98.38  | 29.46   | 94.83   | 99.53  | 44.94  | 91.22    | 99.09  | 47.95   | 92.91   | 99.98   | 7.70    | 97.68   | 99.92   |
> | Maha   | 30.06   | 95.40   | 99.59   | 35.15   | 95.15   | 99.59  | 53.67   | 90.97   | 99.13  | 74.88   | 90.33   | 99.13  | 48.44  | 92.96    | 99.36  | 50.68   | 92.89   | 99.98   | 6.80    | 97.74   | 99.92   |
> | KNN10     | 27.22   | 95.65   | 99.55   | 30.86   | 95.36   | 99.54  | 41.06   | 92.84   | 99.25  | 47.24   | 93.58   | 99.40  | 36.60  | 94.36    | 99.44  | 63.01   | 70.49   | 99.78   | 8.50    | 97.72   | 99.90   |
> | KNN20      | 25.66   | 95.65   | 99.56   | 31.30   | 95.43   | 99.58  | 41.34   | 92.68   | 99.24  | 41.15   | 94.08   | 99.45  | 34.86  | 94.46    | 99.46  | 16.44   | 97.79   | 99.99   | 4.00    | 98.95   | 99.96   |
> | KNN50      | 25.46   | 95.71   | 99.61   | 30.37   | 95.57   | 99.63  | 42.94   | 92.49   | 99.29  | 35.23   | 94.45   | 99.48  | 33.50  | 94.56    | 99.50  | 16.44   | 97.72   | 99.99   | 5.60    | 98.83   | 99.96   |
> | KNN100     | 25.90   | 95.62   | 99.60   | 29.33   | 95.66   | 99.63  | 44.73   | 92.21   | 99.26  | 35.55   | 94.49   | 99.48  | 33.88  | 94.50    | 99.49  | 19.18   | 97.51   | 99.99   | 7.80    | 98.59   | 99.95   |
>
> Remarks: *Average* denotes the average for in-domain OOD datasets; KNN10 (20, 50, 100) denotes using the KNN+ score with K = 10 (K = 20, 50, 100); Adv. stands for Advertisement and H.W. stands for Handwritten.
>
> [A] Non-official implementation of DocFormer: https://github.com/shabie/docformer
>
> [B] Appalaraju et al., DocFormer: End-to-End Transformer for Document Understanding, ICCV 2021
>
> [C] Li et al., DiT: Self-supervised Pre-training for Document Image Transformer, ACM Multimedia 2022
>
> ### Q: Clarification on zero-shot OOD detection
>
> In this paper, our experimental approach aligns with prior research on OOD detection using large-scale pre-trained models [A,B]: the task of OOD detection is defined with respect to *the downstream dataset (ID)*, instead of the pre-training data which is often hard to characterize (**L197 - L200, Page 3**). Furthermore, in the era of large-scale pre-training, identifying realistic OOD categories that aren't subtly connected to the pre-training data can be subjective and challenging.
>
> To isolate the impact of OOD data from pre-training, we also curated a dataset (IIT-CDIP−) from IIT-CDIP by excluding potential OOD categories (detailed results can be found in **Table 4, 5, 10, 11** in the Appendix). We agree that due to the unlabeled and large-scale nature of the pre-training data, achieving a strict zero-shot setting is nearly impossible. To assess the influence of the pre-training dataset, we pre-trained models using 10, 20, 40, and 100% of randomly sampled data from IIT-CDIP−. Our analysis is provided in **Section 4 (Lines 405 - 416)**. We observed that the removal of pseudo OOD categories has a subtle impact on zero-shot OOD detection.
>
>
> [A] Ming et al., Delving into Out-of-Distribution Detection with Vision-Language Representations, NeurIPS 2022
>
> [B] Fort et al., Exploring the Limits of Out-of-Distribution Detection, NeurIPS 2021
>
>
> ### Q: Spatial-aware adapter vs. other multimodal models
>
> Good question! The advantage of spatial-aware adapters lies in their ability to rapidly adapt existing well pre-trained language models for improved OOD detection. Concurrently, the spatial-aware adapter also leads to a noticeable increase in ID accuracy. As shown in Appendix Table 6, we can clearly observe this advantage (ID Acc): RoBERTa-Base (90.19), Spatial-RoBERT-Base (**97.11**), compared to LayoutLMv1 (97.28), UDoc (97.36), and LayoutLMv3 (97.81). Despite the small performance gap compared to document-oriented models such as LayoutLMv3, it's important to note that models with document-tailored pre-training tasks such as LayoutLMv3 pose inherent challenges when trying to leverage the benefits from other well-pretrained language models.
>
>
> Despite the rise of pre-trained LLMs, document-specific models lag notably behind the general-purpose counterparts. This disparity inspired our design of spatial-aware adapters for LMs. The lightweight design of such adaptors may further unleash the potential of future LLMs pretrained on diverse and abundant data.
> Additionally, in contrast to multimodal models such as LayoutLMv3, the adapter acts as a streamlined add-on module, boosting computational efficiency while maintaining strong performance in both ID classification and OOD detection.
>
> Below, we present a comparison between the parameters of our spatial-aware adapter and other models. It's evident that our approach allows for optimizing fewer parameters based on pre-trained LMs, thereby achieving a balance between the training cost and effectiveness.
>
> | Model Name           | Total Params (M)    | Word Embed. Layer Params (M) | Spatial-Aware Adapter Params (M) | Adapter/Total Params (%) |
> | :---                 | :---                 | :---                 | :---                 | :---                 |
> | RoBERTa$_\text{Base}$| 124.65               | 38.60                | -                 | -               |
> | Longformer$_\text{Base}$| 148.66               | 38.60                | -                  | -                  |
> | LayoutLMv1$_\text{Base}$| 112.63               | 23.44                | -                | -                  |
> | LayoutLMv3$_\text{Base}$| 125.33               | 23.44                | -                | -                  |
> | Spatial-RoBERTa$_\text{Base}$ | 125.17               | 38.60                | 0.52                 | 0.42                 |
> | Spatial-Longformer$_\text{Base}$ | 149.19               | 38.60                | 0.52                 | 0.35                 |
>
> ### Q: Include AUPR and Mahalanobis score
>
> Thanks for the suggestion! We have revised the manuscript (Appendix D) and included the AUPR metric for completeness. Some key results are attached below as well.
>
> We acknowledge that, as highlighted by the reviewer, the Mahalanobis score has demonstrated promising OOD detection performance in the NLP domain. However, the arrangement of text blocks in documents is not solely sequential but also spatial. Moreover, documents frequently incorporate a wealth of visual elements. It remains underexplored how logit-based and feature-based OOD detection methods address the unique blend of visual, textual, and layout information in documents. Our paper's primary goal is to systematically explore these complexities and provide deeper insights.
>
> In this work, we mainly use the KNN+ score as a representative feature-based OOD detection score. Compared to the Mahalanobis score, KNN+ is non-parametric (without assuming features follow class-conditional Gaussian), computationally efficient with the FAISS library (https://github.com/facebookresearch/faiss),  and has shown competitive performance compared to the Mahalanobis score with pre-trained models [A].
>
>
> In the initial version of the manuscript, we have included the results with Mahalanobis score in Appendix due to numerically unstable results when calculating the inverse of the covariance matrix for high-dimensional features. We genuinely appreciate your feedback and have meticulously reviewed our implementation. Our updated manuscript now employs a pseudo-inverse (using torch.pinv) for calculating the Mahalanobis score, resulting in numerically stable outcomes. Some key results are also attached below:
>
> **[1] LayoutLMv1**
>
> | OOD Score      | Letter   |   --  |  --    | H.W. |  -- | --  | Adv. |  -- | --  | Memo | --  | --  | Average | -- | --  | Sci. Poster   |   --  |   --   | Receipt | --  | --  |
> |-------------|---------|---------|---------|---------|---------|--------|---------|---------|--------|---------|---------|--------|--------|----------|--------|---------|---------|---------|---------|---------|---------|
> | | FPR95   | AUROC   | AUPR    | FPR95 | AUROC | AUPR | FPR95 | AUROC | AUPR | FPR95 | AUROC | AUPR | FPR95 | AUROC | AUPR | FPR95   | AUROC   | AUPR    | FPR95 | AUROC | AUPR |
> | Energy|27.06 | 92.40 | 99.21 | 37.97 | 91.54 | 99.15 | 45.65 | 88.36 | 98.79 | 35.92 | 91.23 | 99.09 | 36.65 | 90.88 | 99.06 | 24.42 | 94.96 | 99.98 | 57.30 | 86.70 | 99.47 |
> | Maha|17.85 | 96.66 | 99.70 | 33.91 | 94.24 | 99.48 | 36.99 | 92.46 | 99.23 | 24.88 | 95.31 | 99.57 | 28.41 | 94.67 | 99.50 | 24.42 | 94.97 | 99.98 | 57.30 | 86.70 | 99.47 |
> | KNN10 | 20.82 | 96.09 | 99.61 | 35.32 | 93.82 | 99.39 | 40.06 | 91.34 | 99.03 | 28.65 | 94.80 | 99.52 | 31.21 | 94.01 | 99.39 | 20.93 | 90.94 | 99.94 | 25.80 | 93.06 | 99.64 |
> | KNN20 | 21.74 | 95.93 | 99.61 | 36.20 | 93.77 | 99.43 | 41.42 | 91.12 | 99.03 | 30.44 | 94.61 | 99.50 | 32.45 | 93.86 | 99.39 | 17.44 | 97.00 | 99.99 | 49.80 | 93.92 | 99.78 |
> | KNN50 | 24.34 | 95.56 | 99.60 | 38.25 | 93.41 | 99.40 | 43.93 | 90.69 | 99.02 | 33.64 | 94.19 | 99.46 | 35.04 | 93.46 | 99.37 | 17.44 | 96.82 | 99.99 | 51.70 | 93.73 | 99.77 |
> | KNN100 | 25.58 | 95.30 | 99.57 | 39.13 | 93.20 | 99.38 | 45.17 | 90.35 | 98.98 | 34.78 | 93.99 | 99.45 | 36.17 | 93.21 | 99.34 | 23.26 | 96.44 | 99.99 | 53.80 | 93.70 | 99.78 |
>
> **[2] LayoutLMv3**
> | OOD Score      | Letter   |   --  |  --    | H.W. |  -- | --  | Adv. |  -- | --  | Memo | --  | --  | Average | -- | --  | Sci. Poster   |   --  |   --   | Receipt | --  | --  |
> |-------------|---------|---------|---------|---------|---------|--------|---------|---------|--------|---------|---------|--------|--------|----------|--------|---------|---------|---------|---------|---------|---------|
> | | FPR95   | AUROC   | AUPR    | FPR95 | AUROC | AUPR | FPR95 | AUROC | AUPR | FPR95 | AUROC | AUPR | FPR95 | AUROC | AUPR | FPR95   | AUROC   | AUPR    | FPR95 | AUROC | AUPR |
> | Energy|30.70 | 89.18 | 98.79 | 40.42 | 88.18 | 98.76 | 42.98 | 84.10 | 98.14 | 33.12 | 88.23 | 98.57 | 36.80 | 87.42 | 98.56 | 19.77 | 94.50 | 99.98 | 11.70 | 97.02 | 99.90 |
> | Maha| 20.98 | 95.28 | 99.55 | 35.72 | 93.41 | 99.39 | 30.57 | 92.70 | 99.23 | 17.33 | 96.31 | 99.67 | 26.15 | 94.42 | 99.46 | 19.77 | 94.51 | 99.98 | 11.70 | 97.03 | 99.90 |
> | KNN10| 21.74 | 95.03 | 99.48 | 35.68 | 93.38 | 99.33 | 32.88 | 91.86 | 99.05 | 18.51 | 96.26 | 99.67 | 27.20 | 94.13 | 99.38 | 51.16 | 66.13 | 99.68 | 34.40 | 86.03 | 98.94 |
> | KNN20    | 22.74 | 94.90 | 99.49 | 36.56 | 93.20 | 99.37 | 33.96 | 91.66 | 99.04 | 19.64 | 96.15 | 99.66 | 28.22 | 93.98 | 99.39 | 11.63 | 97.58 | 99.99 | 8.90  | 97.97 | 99.93 |
> | KNN50      | 24.62 | 94.62 | 99.48 | 38.37 | 92.71 | 99.32 | 35.83 | 91.38 | 99.06 | 21.63 | 95.93 | 99.64 | 30.11 | 93.66 | 99.38 | 12.79 | 97.44 | 99.99 | 10.00 | 97.89 | 99.93 |
> | KNN100      | 25.22 | 94.38 | 99.45 | 39.29 | 92.32 | 99.28 | 36.55 | 91.09 | 99.03 | 22.48 | 95.79 | 99.63 | 30.88 | 93.40 | 99.35 | 13.95 | 97.20 | 99.99 | 10.70 | 97.72 | 99.92 |
>
> **[3] UDoc**
>
> |  OOD Score     | Letter   |   --  |  --    | H.W. |  -- | --  | Adv. |  -- | --  | Memo | --  | --  | Average | -- | --  | Sci. Poster   |   --  |   --   | Receipt | --  | --  |
> |-------------|---------|---------|---------|---------|---------|--------|---------|---------|--------|---------|---------|--------|--------|----------|--------|---------|---------|---------|---------|---------|---------|
> | | FPR95   | AUROC   | AUPR    | FPR95 | AUROC | AUPR | FPR95 | AUROC | AUPR | FPR95 | AUROC | AUPR | FPR95 | AUROC | AUPR | FPR95   | AUROC   | AUPR    | FPR95 | AUROC | AUPR |
> | Energy| 45.96 | 82.12 | 97.82 | 47.21 | 86.40 | 98.58 | 49.64 | 83.16 | 98.14 | 49.59 | 83.13 | 98.08 | 48.10 | 83.70 | 98.15 | 2.33  | 98.57 | 100.00| 4.00  | 98.34 | 99.94  |
> | Maha | 25.38 | 94.71 | 99.49 | 39.61 | 88.87 | 98.84 | 42.18 | 90.45 | 99.01 | 34.74 | 93.53 | 99.40 | 35.48 | 91.89 | 99.18 | 2.33  | 98.60 | 100.00| 4.00  | 98.36 | 99.94  |
> | KNN10| 30.02 | 94.47 | 99.45 | 41.22 | 88.66 | 98.77 | 41.90 | 90.99 | 99.02 | 36.65 | 93.48 | 99.41 | 37.45 | 91.90 | 99.16 | 41.86 | 82.13 | 99.84 | 48.60 | 76.17 | 97.34  |
> | KNN20| 31.10 | 94.36 | 99.46 | 41.98 | 88.44 | 98.78 | 42.10 | 90.90 | 99.03 | 38.03 | 93.35 | 99.40 | 38.30 | 91.76 | 99.17 | 1.16  | 99.13 | 100.00| 5.50  | 98.42 | 99.95  |
> | KNN50| 33.95 | 94.07 | 99.45 | 43.35 | 87.89 | 98.70 | 44.01 | 90.72 | 99.05 | 40.71 | 93.06 | 99.37 | 40.51 | 91.43 | 99.14 | 1.16  | 99.04 | 100.00| 6.90  | 98.32 | 99.94  |
> | KNN100| 34.83 | 93.84 | 99.43 | 43.75 | 87.51 | 98.65 | 45.01 | 90.61 | 99.04 | 41.96 | 92.90 | 99.36 | 41.39 | 91.22 | 99.12 | 1.16  | 98.84 | 100.00| 7.40  | 98.26 | 99.94  |
>
> **[5]  Spatial-RoBERTa$_\text{Large}$**
> | OOD Score      | Letter   |   --  |  --    | H.W. |  -- | --  | Adv. |  -- | --  | Memo | --  | --  | Average | -- | --  | Sci. Poster   |   --  |   --   | Receipt | --  | --  |
> |-------------|---------|---------|---------|---------|---------|--------|---------|---------|--------|---------|---------|--------|--------|----------|--------|---------|---------|---------|---------|---------|---------|
> | | FPR95   | AUROC   | AUPR    | FPR95 | AUROC | AUPR | FPR95 | AUROC | AUPR | FPR95 | AUROC | AUPR | FPR95 | AUROC | AUPR | FPR95   | AUROC   | AUPR    | FPR95 | AUROC | AUPR |
> | Energy| 33.39 | 90.16 | 98.89 | 39.25 | 89.88 | 98.84 | 42.30 | 88.13 | 98.65 | 37.05 | 91.66 | 99.10 | 38.00 | 89.96 | 98.87 | 31.40 | 92.42 | 99.97 | 27.70 | 94.22 | 99.78 |
> | Maha | 26.74 | 94.45 | 99.46 | 42.19 | 92.21 | 99.24 | 36.51 | 92.21 | 99.17 | 27.44 | 95.14 | 99.55 | 33.22 | 93.50 | 99.36 | 24.42 | 96.47 | 99.99 | 18.30 | 96.34 | 99.88 |
> | KNN10| 28.18 | 94.47 | 99.43 | 42.43 | 93.01 | 99.30 | 37.43 | 91.74 | 99.06 | 31.13 | 94.72 | 99.53 | 34.79 | 93.49 | 99.33 | 25.58 | 96.24 | 99.99 | 18.60 | 96.28 | 99.87 |
> | KNN20| 28.78 | 94.32 | 99.42 | 42.43 | 92.90 | 99.29 | 38.07 | 91.58 | 99.05 | 32.02 | 94.55 | 99.51 | 35.33 | 93.34 | 99.32 | 25.58 | 96.02 | 99.99 | 18.60 | 96.33 | 99.88 |
> | KNN50| 30.22 | 93.95 | 99.42 | 43.71 | 92.69 | 99.32 | 40.06 | 91.26 | 99.07 | 34.54 | 94.10 | 99.47 | 37.13 | 93.00 | 99.32 | 26.74 | 95.52 | 99.99 | 21.40 | 96.14 | 99.87 |
> | KNN100| 30.86 | 93.71 | 99.39 | 44.11 | 92.56 | 99.30 | 40.66 | 91.05 | 99.05 | 35.47 | 93.88 | 99.45 | 37.78 | 92.80 | 99.30 | 26.74 | 95.22 | 99.99 | 21.70 | 96.11 | 99.87 |
>
> Remarks: *Average* denotes the average for in-domain OOD datasets; KNN10 (20, 50, 100) denotes using the KNN+ score with K = 10 (K = 20, 50, 100); Adv. stands for Advertisement and H.W. stands for Handwritten.
>
> [A] Sun et al., Out-of-Distribution Detection with Deep Nearest Neighbors, ICML 2022
>
> ### Q: Clarification on multimodal models in the zero-shot setting
>
> Due to the space constraints, we have included the zero-shot results on multimodal models in the Appendix (LayoutLMv1 in **Table 5**, p22 and Spatial-RoBERTa in **Table 6**). As shown in **Figure 2(a)**, Multimodal models are primarily categorized into single-stream and two-stream architectures. In our case, we opted for the single-stream variant (LayoutLMv1/v3) due to their incorporation of the CLS feature. However, when considering two-stream multimodal models, it's worth noting that during pretraining, there exist separate heads for visual and textual features. Consequently, such structure poses challenges for zero-shot scenarios as it prevents the use of feature-based and logit-based OOD detection methods. Instead, the feasible approach is to employ these methods after completing fine-tuning and multimodal feature fusion for two-stream models.

---

### Meta-Review · Area_Chair_6Ni2 · 2023-09-11

**Recommendation:** 4

**Metareview:**

This paper presents an extensive and in-depth investigation into the influence of pre-training, fine-tuning, model-modality, and OOD scoring functions across a wide range of document OOD detection tasks. To enhance OOD detection performance, the authors introduce a spatial-aware adapter. The paper's well-structured presentation effectively communicates the central concepts, and the experiments conducted are comprehensive.

All reviewers agree that this paper offers a thorough and extensive empirical analysis of document OOD detection, yielding valuable insights substantiated by robust experimental results. In light of this, the reviewers find the paper sound and express enthusiasm for the practical implications of the proposed add-on spatial-aware adapter and the depth of the analysis.

As a result, the final decision for the paper is acceptance.

---

### Decision · Program_Chairs · 2023-10-07

**Decision:**

Accept-Findings

**Comment:**

This paper presents an extensive and in-depth investigation into the influence of pre-training, fine-tuning, model-modality, and OOD scoring functions across a wide range of document OOD detection tasks. To enhance OOD detection performance, the authors introduce a spatial-aware adapter. The paper's well-structured presentation effectively communicates the central concepts, and the experiments conducted are comprehensive.

All reviewers agree that this paper offers a thorough and extensive empirical analysis of document OOD detection, yielding valuable insights substantiated by robust experimental results. In light of this, the reviewers find the paper sound and express enthusiasm for the practical implications of the proposed add-on spatial-aware adapter and the depth of the analysis.

As a result, the final decision for the paper is acceptance.